# Hydrological tracers for assessing transport and dissipation processes of pesticides in a model constructed wetland system

Elena Fernández-Pascual [1], Marcus Bork [1, 2], Birte Hensen [3], Jens Lange [1]

[1] Hydrology, Faculty of Environment and Natural Resources, University of Freiburg, Freiburg, Germany
[2] Soil Ecology, Faculty of Environment and Natural Resources, University of Freiburg, Freiburg, Germany
[3] Institute of Sustainable and Environmental Chemistry, Leuphana University Lüneburg, Lüneburg, Germany

*Correspondence to:* Elena Fernández-Pascual (elena.fernandez@hydrology.uni-freiburg.de)

**Abstract.** Studies that have used hydrological tracers to investigate the fate and transport of pesticides in constructed wetlands have often considered such systems as a "black box". Consequently, internal temporal and spatial mechanisms that dominate pesticides transport and dissipation (e.g. sorption, transformation, plant uptake) are still not fully understood. Here we present a novel approach that combines the use of tracers with different sorptive and reactive properties (i.e., bromide ($Br^-$), uranine (UR) and sulforhodamine B (SRB)) with high vertical-resolution sampling and monitoring to evaluate transport and dissipation processes of three selected pesticides (boscalid, penconazole and metazachlor) inside a model constructed wetland system on a long term basis and detailed spatial scale. Moreover, the influence of vegetation and alternating different hydrologic conditions on transport and dissipation processes has been evaluated by comparing a vegetated with a non-vegetated section and by alternating periods of saturation and drying. Breakthrough curves obtained at different sampling depths pointed out that the solutes were not equally distributed within the constructed wetland. Data revealed that a higher mass of solutes was transported to the vegetated part of the uppermost layer, which was associated with possible lateral transport at or near the surface and/or a shortcut effect produced by the roots. In contrast, the middle layers showed retardation, most likely due to the presence of water filled pores before the injections and low pore connectivity in the vicinity of the sampling ports. The strong temporal and spatial correlation found between $Br^-$, UR and metazachlor indicated that these solutes experienced less retention than SRB, boscalid and penconazole, which most likely underwent sorption, as evidenced by their absence in the middle layers, rapid decrease in their concentrations after the injections and gradual increase in accumulated mass recovery at the outlet. The overall tracer mass balance allowed us to identify three dissipation pathways: sorption, transformation and plant uptake. The detection of metazachlor transformation products (TPs) confirmed the contribution of transformation to metazachlor dissipation, whereas no TPs for boscalid and penconazole were detected. Yet, their transformation could not be ruled out in the present study. Hot spots of sorption and transformation were found in the uppermost layer, whereas hot moments were detected at the beginning of the experiment for sorption and after promoting aerated conditions for transformation. The use of hydrological tracers coupled with high vertical-resolution sampling and monitoring proved to provide valuable information about transport vectors and dissipation processes of pesticides inside a constructed wetland. This study represents a first approximation, and further experiments need to be done under field conditions together with modeling.

## 1 Introduction

Pesticides are widely used to protect crops and increase their yields around the world. It is well known that their use might result in ecotoxicological effects in non-target environments (Stehle and Schulz, 2015). Chemical analysis performed in the waters of European countries revealed that pesticides are often detected in surface waters (Müller et al., 2002; von der Ohe et al., 2011; Casado et al., 2019). This problem becomes even more severe if the incomplete degradation of pesticides leads to a large number of transformation products (TPs), whose behavior is often unknown, and whose toxicity or persistence may be greater than that of the parent compounds. In fact, the presence of TPs in water bodies has already been reported in numerous studies (Kolpin et al., 2004; Eurostat, 2012; Reemtsma et al., 2013).

Buffer zones emerged as a measure for controlling water pollution. Constructed wetlands are one example of buffer zones, where the removal of pesticides takes place. Constructed wetlands are designed to simulate and take advantage of processes that occur in natural wetlands (Vymazal et al., 2005), such as sedimentation, photolysis, hydrolysis, adsorption, microbial degradation and plant uptake (Vymazal et al., 2015). In these systems vegetation plays an essential role promoting

sedimentation by reducing the current velocities of the water (Petticrew and Kalff, 1992), providing a substrate for microorganism in the roots and rhizomes (Hofmann, 1986) and creating oxidized conditions in the roots that stimulate aerobic decomposition (Brix, 1997). Removal processes in constructed wetlands may also be promoted through intermittent water flows by enhancing aeration and by providing different redox conditions suitable for the growth of different microbiological communities (Ong et al., 2010; Maillard et al., 2011; Fan et al., 2013).

The mitigation capacities of buffer zones have recently been studied by using hydrological tracers as a low-cost approach. In this context, fluorescent tracers (e.g. uranine (UR), sulforhodamine B (SRB)) have often been chosen to study transport and fate of pesticides because they are organic molecules, non-toxic and easy to be analysed. For instance, some authors have used them in wetlands (Passeport et al., 2010; Lange et al., 2011; Durst et al., 2013; Maillard et al., 2016) and farm ditches (Dollinger et al., 2017). However, in most cases where this approach has been applied, the system under study has been treated as a "black

box" where the time scales were typically limited to the time spans of the tracers breakthroughs at the system outlet. Hence, internal temporal and spatial mechanisms that dominate pesticides transport and dissipation (e.g. sorption, transformation, plant uptake) are still not fully clear. Moreover, information on the fate and, particularly, transformation of pesticides inside wetland sediments is still limited.

Therefore, the objectives of this study are i) to apply a multi-tracer approach together with high vertical-resolution sampling

and monitoring to identify transport patterns and dissipation processes of three pesticides selected as test substances inside a model constructed wetland system; ii) to compare the temporal and spatial behavior of the applied tracers with the pesticides and evaluate their main dissipation pathways; and iii) to assess the influence of vegetation and alternating different hydrologic conditions (saturated and unsaturated) on transport and dissipation processes.

Our study is one of the first to look at the solutes behavior inside a constructed wetland on a long-term basis and detailed spatial

scale. With this experiment we expect to provide new insights about the potential of hydrological tracers to evaluate transport and dissipation processes of pesticides. Likewise, we seek to extend the knowledge on the mitigation capacities of constructed wetlands with our approach.

The experiment was conducted in a constructed wetland with one half planted with two common wetland plants and the other half unplanted. The constructed wetland was equipped with a system designed to perform high vertical-resolution sampling

and monitoring on a long-term basis. Three hydrological tracers were chosen as reference substances according to their reactive nature: bromide ($Br^-$) as a non-adsorbing tracer (Whitmer et al., 2000), UR as a photosensitive tracer (Gutowski et al., 2015) that can undergo processes of (bio-) chemical transformation (Lange et al., 2018) and SRB as a highly sorptive tracer (Kasnavia et al., 1999). Three pesticides were selected as test substances according to their different physicochemical properties and frequent detection in a field-based constructed wetland where other studies within the same project were carried out: boscalid

(2-chloro-N-(4'-chlorobiphenyl-2-yl) nicotinamide), penconazole ((RS)-1-[2-(2,4-dichlorophenyl) pentyl]-1H-1,2,4-triazole) and metazachlor (2-chloro-N-(pyrazol-1-ylmethyl) acet-2',6'-xylidide).

## 2  Materials and Methods

### 2.1  Chemicals

The physio-chemical properties of tracers and pesticides are summarized in Table 1. UR was purchased from Simon & Werner

GmbH (CAS-no. 518-47-8), SRB from Waldeck GmbH & Co KG (CAS-no. 3520-42-1) and $Br^-$ was obtained as sodium bromide from Carl Roth GmbH & Co KG. Boscalid (99.8%), penconazole (99%) and metazachlor (99.7%) already dissolved

in acetonitrile (99.9%) were purchased from Neochema (Bodenheim, Germany). The analytical standards of boscalid (99.9 %), penconazol (99.1 %), metazachlor (99.6 %) and p-Chlorobenzoic acid (99%) were purchased from Sigma-Aldrich Chemie GmbH (Steinheim, Germany). 1,2,4-Triazole (99.5%) was purchased from LGC Standards (Wesel, Germany). Metazachlor-ESA (95 %) and metazachlor-OA (98.8 %), hereunder named as met-ESA and met-OA, respectively, and the internal standard Terbutryn-D5 (98.5 %) already dissolved in acetonitrile (100 μg mL-1) were received from Neochema (Bodenheim, Germany).

The target injection masses of tracers and pesticides for an injection volume of 40 L were calculated according to Durst et al. (2013). Standard stock solutions of 1 g L$^{-1}$ for UR and SRB and of 10 g L$^{-1}$ for Br$^-$ were prepared in MilliQ water. Pesticides (0.1 g L$^{-1}$) dissolved in acetonitrile were directly mixed with the injection solution. The concentration of tracers and pesticides in the injection solution was 100 mg L$^{-1}$, 50 μg L$^{-1}$ and 100 μg L$^{-1}$, for sodium bromide, UR and SRB, respectively; and 50 μg L$^{-1}$ for boscalid, penconazole and metazachlor.

## 2.2 Design of the model constructed wetland system

The model constructed wetland system consisted of a glass tank 177.4 cm long, 47.6 cm wide and 56.8 cm deep (Fig. 1). The system was divided into three parts, two of them inlet and outlet reservoirs located at both ends and separated by two glass walls and a third part located in the middle consisting of the main bed of sediments. The bottom was filled with 10 cm of gravel (grain size 4-8 mm) and topped with 32 cm of sand (grain size 0.01-2 mm). The characteristics of the system are given in Table 2A. One half of the system was left unplanted, while the other half was planted with two species of widespread and ubiquitous wetland plants (*Typha latifolia* and *Phragmites australis*) with an average initial height of 79.8 cm ± 18.6 and 76.9 cm ± 10.1, respectively. They were purchased from a local garden center. The whole experiment was carried out indoors in a laboratory assembly. Therefore, 64 OSRAM SSL 3W light-emitting diode lamps for plant growth (Purple Alien 2.0, LED Grow Shop, Germany) were installed with daily photoperiods of 11 hours.

The inlet and outlet were intended to create vertical water flows. This was achieved through the installation of two pairs of peristaltic pumps coupled to Plexiglas pipes (15 mm diameter) that were connected to the bottom of the system; and another two pairs of peristaltic pumps coupled to Plexiglas pipes that channelled directly into the inlet and outlet reservoirs. A tank with a capacity of 350 L that was connected to the tap water, served as external inlet reservoir, while a second tank with a capacity of 1000 L received the waste water. In order to monitor the water level, three pairs of PVC observation pipes (DN: 35-40 mm, STÜWA Konrad Stükerjürgen GmbH, Germany) with a length of 50 cm, were arranged symmetrically on both sides at the center line of the system. One half of the pipes were located at the gravel layer and the other half in the sand.

In order to obtain pore water samples with high vertical resolution, a multi-level pipe with a sampling resolution of 12 cm was designed. This resulted in a total of four sampling depths that ranged from the gravel to the uppermost layer of the sand. Two multi-level pipes were installed in the sediment bed: one at the non-vegetated and the other one at the vegetated half of the system. Small glassfilters (12.5 mm diameter, porosity 2, ROBU, Germany) were installed in both multi-level pipes at each sampling depth. The filters were connected to a multichannel peristaltic pump (Pulse-free flow peristaltic pump, Gilson, France) via capillaries made of stainless steel (1/16" inner diameter, Swagelok, Germany) that were directly inserted into TYGON tubes (ID: 1.02 mm, Proliquid, Germany). In addition, 5TE sensors (Decagon Em50 serie, Campbell Scientific) and redox probes (Paleo terra, Amsterdam, The Netherlands) were installed at the same depths in both multi-level pipes. Glassfilters, sensors and probes were separated from each other at an angle of 90 degrees at each sampling depth (Fig. 1B). Furthermore, a reference electrode (Ag:AgCl) connected to the redox probes, was inserted in the sediment between the multi-level pipes. All sensors and probes were connected to a datalogger (CR1000, Campbell Scientific).

## 2.3 Operation of the model constructed wetland system

The model constructed wetland system was designed to alternate saturated and unsaturated conditions (long periods of aeration). A total of three phases were created (Fig. 2): 1) saturation with target substances (one week), 2) drying by

evapotranspiration (three weeks) and 3) saturation with tap water (one month). During the first saturation phase, only one injection was made, whereas in the second saturation phase, the system was kept saturated by constant injections of tap water. The operation of the constructed wetland is summarized in Table 2B. The solution of tracers and pesticides was prepared immediately before the injection. To check possible interactions between substances, the concentrations in the solution were

5 measured on the day of the injection and a couple of days after. Prior to the injection, the system was drained until field capacity was reached. The whole experiment lasted seven months (from March 2017 to October 2017), during which two identical experimental runs were performed. The first run (from March 9 to May 9, 2017) was followed by a resting period of about three months (from May 9 to August 1, 2017), during which occasional water additions to maintain the vegetation were carried out. After this, the second run (from August 1 to October 3, 2017) was conducted.

The execution of the injections is shown in Fig. 3. Three injections took place in each run of the experiment: (i) initial surface injection of tracers and pesticides, (ii) injection of tap water from the bottom of the system and (iii) flushing of the sediments with tap water from the bottom. The surface injection (i) was performed after having drained the system. The solution was constantly pumped into the inlet reservoir. Then, it overflowed the reservoir and enter the sediments bed. Due to the low flow rate the solution moved first downward near the inlet and then upward as the system was filling up. The inflow was held

constant for about two hours until the system became saturated and the upward flow formed a surface ponding of approximately two centimeters. In this way, possible entrapment of air in the system was avoided. The second injection (ii) was performed at the end of the drying phase by pumping tap water from the bottom. The water flowed evenly through the sediment in vertical upward direction. Again, the inflow was maintained until the system became saturated and a surface ponding of approximately two centimeters was formed. This injection was repeated throughout the second saturation phase in order to keep the system

constantly saturated. The flushing of the system (iii) was performed at the end of the second saturation phase and it was intended to recover all mobile fractions of the target compounds. To do this, tap water was injected from the bottom and was allowed to flow into the system continuously. Water overflowed the main bed and exited towards the outlet reservoir, from where it was pumped to the waste tank.

## 2.4 Sampling and monitoring

Pore water samples were collected from different depths twice a week during the experimental runs. The sampling of pore water was performed simultaneously in order to prevent the mixing of waters. A volume of 60 mL of pore water was transferred to 100 mL brown glass bottles and stored at 4°C for major ions and tracers analysis. Previously, a volume of 10 mL was transferred to 15 mL polypropylene tubes and stored at -20°C for the subsequent pesticide and TPs analysis. Polypropylene was chosen instead of glass because the samples had to be frozen immediately after their collection. Such material has already

been used to store pesticides in other studies (e.g. Joseph, 2015). Additional pore water samples were taken before and after the initial injections of tracers and pesticides to account for the background. During the flushing of the system, surface water samples were collected at the outlet and transferred to 100 mL brown glass bottles. Following this, the samples were stored at 4°C for the subsequent analysis of major ions and tracers and -20°C for the analysis of pesticides and TPs.

At the end of the experiment the sediment bed was emptied of its gravitational water. Then, 16 sediment cores (four per

35 longitudinal and four per lateral transect) were collected by inserting plastic pipes into the sediment. Sediment cores were divided into four fractions, each representing a different sampling depth (0-8 cm, 9-20 cm, 21-32 cm, 33-42 cm). The sediment samples were dried at room temperature for 24 h and stored in the dark for subsequent measurements of tracers, total organic carbon and iron oxides. Following this, the plants were removed from the vegetated zone and separated into aerial parts (stems and leaves) and roots. Immediately after they were oven dried at 60°C for approximately 24 hours and stored in the dark for

subsequent measurements of tracers. Biomass was determined on a dry matter basis.

Temperature, soil moisture, conductivity and redox potential were constantly monitored by means of the datalogger with an interval of two minutes throughout the entire experiment. Redox potential was calculated by adding the potential from the

reference electrode (Ag/AgCl) to the measured potential (Vorenhout et al., 2011). The final result was corrected for differences in temperature according to Bard et al. (1985).

## 2.5 Laboratory analysis

### 2.5.1 Major ions and tracers in the pore- and outlet-water

Pore- and outlet-water samples were measured for major ions ($Na^+$, $NH_4^+$, $K^+$, $Ca^{2+}$, $Mg^{2+}$, $Br^-$, $SO_4^{2-}$, $Cl^-$, $NO_3^-$ and $NO_2^-$) by ion chromatography (Dionex ICS-1100, Thermo Scientific, USA). All samples were previously filtered with a 0.45 µm filter.

Concentrations of the tracers UR and SRB in pore and outlet water samples were measured by fluorescence spectrometry (Perkin Elmer LS 50 B) as previously described (Leibundgut et al., 2009). Briefly, a synchronous scan method was applied with an excitation/emission wavelength difference of 25 nm and target wavelengths of 488 nm and 561 nm for UR and SRB, respectively. Detection limits were 0.05 µg $L^{-1}$ for UR and 0.1 µg $L^{-1}$ for SRB. The entire fluorescent spectrum (from 350 to 600 nm) was analyzed in order to identify different background fluorescent levels and subtract them.

### 2.5.2 Pesticides and TPs in the pore- and outlet- water

Pore- and outlet-water samples were analyzed for the pesticides boscalid, penconazole, metazachlor and their known TPs (metazachlor-ESA and -OA, p-Chlorobenzoic acid (boscalid), and 1,2,4-Triazole (penconazole)). Acetonitrile (LC-MS grade; VWR International GmbH, Darmstadt, Germany) was used as organic mobile phase in chromatography and for the preparation of stock solutions. Aqueous mobile phase was prepared with ultrapure water (Membra Pure, Germany; Q1:16.6 MΩ and Q2: 18.2 MΩ). Samples were filtered using syringe filter units (CHROMAFIL® Xtra RC-20/25; Macharey-Nagel, GmbH & Co. KG, Germany). Each sample (990 µL) was spiked with 10 µL Terbutryn-D5 as internal standard. Analysis of 5 µL of each sample was done by LC-MS/MS (Agilent Technologies, 1200 Infinity LC-System and 6430 Triple Quad, Waldbronn, Germany). Mobile phases were 0.01 % formic acid (A) and acetonitrile (B) with a flow of 0.4 mL $min^{-1}$. Gradient was as follows: 0-1 min (10% B), 1-11 min (10-50% B), 11-18 min (50-85% B), 18-21 min (85-90% B), 21-24 min (90% B), 24-26 min (90-10% B) and 26-30 (10% B). A NUCLEODUR® RP-C18 (125/2; 100-3 µm C18 ec) column (Macherey Nagel, Düren, Germany) was used as stationary phase and oven temperature was set to 30°C. Limits of detection (LOD) and quantitation (LOQ) were calculated with DINTEST (2003) according to DIN 32645. LOQ/LOD values for pesticides and transformation products are provided in Table 3.

### 2.5.3 Extraction and measurement of tracers in the sediments and plants

UR and SRB in the sediment (sand) and plants were extracted as described by Wernli (2009). In brief, two grams of the dried material were mixed with 10 mL of ammonia-ethanol solution (40:60, v/v). Dried stems, leaves and roots were previously grinded with a vibratory disc mill (Siebtechnik GmbH, Germany). All samples were shaken on an IKA HS 250 reciprocating shaker for 30 minutes at 240 rpm and stored at 4°C in the refrigerator for at least 24 hours. Afterwards, supernatant was collected, filtered (< 0.45 µm) and measured for the tracers. The resulting curves were corrected through interpolation and subtraction of the background signal from the peak intensity as described by Leibundgut et al. (2009).

A different methodology based on McMahon et al. (2003) was used to measure $Br^-$ in the sediment (sand) and plants to avoid the interference of the ammonia-ethanol solution with the ion chromatograph. Samples were prepared in the same way as previously described, but they were mixed with 20 mL of deionized water instead. Following this, they were shaken on an IKA HS 250 reciprocating shaker for 1 hour at 240 rpm and later centrifuged at 3000 rpm for 30 minutes (Megafuge 1.0R; Heraeus Instruments). Supernatant was then taken, filtered (< 0.45 µm) and measured by ion chromatography (Dionex ICS-1100, Thermo Scientific, USA).

### 2.5.4 TOC and iron oxides in the sediment

Total organic carbon (TOC) was measured in the sediment (sand) with a CNS-analyser (Vario El Cube, Elementar, Germany) after grinding the dried samples with a vibratory disc mill (Siebtechnik GmbH, Germany). Dithionite-extractable Fe ($Fe_d$) in the sediment (sand) was extracted according to Mehra and Jackson (1960) and measured using inductively coupled plasma - optical emission spectrometry (Spectro Ciros CCD, Spectro Analytical Instruments GmbH, Germany).

## 2.6 Data Analysis

Spatial and temporal dynamics of transport processes in the pore water were investigated by analyzing soil moisture data and Br⁻ breakthrough curves. Here, Br⁻ was chosen as reference due to its most conservative character. The occurrence and role of retardation was studied by performing cross-correlations between Br⁻ time series. The predominance of transport processes among the solutes was examined by looking at the relationship between Br⁻ and the other solutes via correlation matrices of the measured concentration time series. Here, it was assumed that a weak correlation would be due to the prevalence of sorption and transformation rather than transport. This was based on the premise that the solutes would experience retardation due to sorption or attenuation due to transformation. Transformation processes were examined through the detection of TPs. The calculated correlation matrices were also used to analyze the general behavior of the solutes and their relationship in the pore water. Spearman rank correlation coefficients ($rho_s$) were applied since the data did not fit a normal distribution. Correlations were calculated individually for the vegetated and the non-vegetated zone and the different depths. The influence of vegetation and hydrologic conditions on transport and dissipation processes was evaluated through the comparison of the results of the vegetated with the non-vegetated zone and the different phases. In addition, the performance of the two experimental runs was assessed by correlations between Br⁻ breakthrough curves of the first and second run.

Further comparisons between tracers, pesticides and their TPs were made by analyzing their cumulative recovery curves obtained at the outlet after the flushing phases. The fate of the tracers and main dissipation pathways were examined with a final overall mass balance that accounted for five different compartments (pore water, outlet water, sediments, stems + leaves and roots). The mass of tracers and pesticides recovered in the pore water was calculated as the sum of the weekly dissolved concentrations multiplied by the volume sampled. The mass of tracers and pesticides recovered in the outlet water was calculated based on the recovery curves obtained during the flushing phases. The mass of tracers and pesticides recovered in the sediments and plants was determined as the concentrations measured in their corresponding compartments multiplied by the total amount of sediments and plants in the system, respectively.

In the present study pesticides and their TPs could not be measured in the sediment and plants because a quantitative method was lacking. This issue pointed to the advantage of using tracers instead of pesticides because they are generally easier to measure. Statements on the behavior of pesticides in the compartments, where they could not be measured, were made according to their physicochemical properties, the results of the breakthrough and recovery curves, and their comparison with the tracers.

## 3 Results and discussion

### 3.1 Transport processes in the pore water according to Br⁻ behavior

The relative concentrations of Br⁻ measured at different depths (Table 4) indicated that similar values were reached in the lower- and uppermost layers right after the injection. In contrast, no Br⁻ was detected in the middle sections. These results were attributed to the conditions previous to the injection (i.e., system at field capacity). In such context, the presence of water-filled pores may have caused heterogeneities resulting in an uneven distribution of solutes within the system. Indeed, soil moisture values (Fig. 4) measured prior to the injections (indicated with a red circle) were three to two times higher in the middle sandy layers (15 and 27 cm) compared to the lower- and uppermost layers (39 and 3 cm, respectively). Moreover, such values barely changed over the experiment, thus suggesting that the water holding capacity of the middle sections was higher compared to

the other layers (at least in the vegetated zone, since there is no data from the non-vegetated). These results were associated with a possible lack of connectivity, as already suggested by Nimmo (2012). Consequently, water flow through these sections was most likely inhibited and/or delayed, as evidenced by the initial absence of Br⁻ (Table 4) and subsequent delay of the breakthrough peaks (see Fig. 5).

On the other hand, given that Br⁻ was detected in the uppermost layer (see Table 4), it was assumed that the heterogeneities due to the presence of poorly connected pores in the middle layers were only located in some areas, including the surroundings of the sampling ports. In the rest of the system probably matrix flow dominated. According to Fig. 5, the uppermost layer displayed a delayed breakthrough peak with relative concentrations of Br⁻ about three times higher than the maximum detected in the bottom (see also Table 4). In addition, the maximum values reached in the vegetated zone of the uppermost layer were twice

as high as those of the non-vegetated, although these differences were not that pronounced in the second run. Hence, it can be speculated that lateral transport at or near the surface may have occurred during the injections causing augmented transport of solutes towards the vegetated surface. However, other possible explanations could not be ruled out. These include the likely influence of the plants by means of water uptake and the possible contribution of the roots to the formation of channels through which preferential flow took place. Other mechanisms, not necessarily related to plants (e.g. fingering), may have been involved

in the transport of solutes to the vegetated area, too.

Lag correlations performed to the Br⁻ breakthrough curves (Fig. 6) confirmed the delayed arrival of solutes to the middle and uppermost layers. A significant lag correlation could only be observed in the non-vegetated zone. Specifically, the delays obtained in the middle layers (at time $t = -7$ and $t = -3$ in the first and the second run, respectively) were greater than those obtained in the uppermost layer (at time $t = -5$ and $t = -2$ in the first and the second run, respectively). The delayed peak of Br-

20 observed at 15 cm depth in the non-vegetated area for the first run was also detected by the conductivity probe located at the same depth. The complete breakthrough curve could not be capture by the sensor, but a strong correlation (Spearman's rho=0.83 and p-value<0.001) between Br⁻ and the conductivity values was found (see Table S2 of the supplementary material). Similar results could not be observed in the non-vegetated layer at 27 cm depth due to the failure of the sensor. Overall, the delayed peak of solutes in the middle layers coincided in time with the end of the drying phase and beginning of the second saturation

phase. Hence, it was related to the likely migration of solutes during the drop and subsequent rise of the water table. In contrast, the delayed peak of the uppermost layer was associated with possible low pore connectivity in the vicinity of the sampling ports. Here the solutes probably arrived earlier, but we could not measure them at the time of the arrival presumably because of the presence of water-filled pores in the surroundings of the sampling ports.

If we compare the performance of the vegetated and the non-vegetated zone by means of correlations (Table 5), we observe

stronger correlations in the lower layers (at 27 and 39 cm depth) than in the upper layers (at 15 and 3 cm depth), especially in the uppermost during the first run. These results suggested a greater influence of the plants and/or other causes (e.g. transport along the surface) on solute transport in the upper layers. These findings also indicated that the system did not behave the same way in each run. In fact, if we evaluate its performance over time, differences were observed (Table 6). With the exception of the uppermost layer, the non-vegetated zone showed strong correlation between the two runs regardless of the layer, whereas

in the vegetated zone some layers did not show any correlation (layers at 3 and 27 cm depth) or displayed correlations that were statistically non-significant (layers at 15 and 39 cm depth). These results indicated that the overall performance of the non-vegetated zone was similar in both runs, whereas the vegetated behaved differently. This was not applicable to the uppermost layer because it was probably influenced by the surface water and/or the presence of plants. Lag correlations between the first and second run were also analyzed (Fig. S3 of the supplementary material). A significant value (at time t=-3) was

found in the vegetated zone at 15 cm depth. In general, these results were related to possible changes in root density and/or spatial distribution in the system over time. This assumption was supported by visual observations of the sediment (Fig. 7).

However, other causes, such as the influence of flushing between runs and Br⁻ uptake by the plants (Xu et al., 2004) could not be ruled out.

### 3.2 Relationship between solutes and their behavior in the pore water throughout the experiment

Overall, the injected solutes have followed the same trend as Br⁻ in the lower- and uppermost layers (Fig. 5). Conversely, in the middle sections, only UR and metazachlor behaved similar to Br⁻, although this was only observed in the non-vegetated part. In fact, the vegetated zone only displayed observable amounts of Br⁻. Here in the middle layers, dominant processes most likely differed between zones, as was also evidenced by the different redox potentials measured in the vegetated and non-vegetated zone (See Fig. 5, second y-axis).

Two of the major metabolites of metazachlor, namely met-ESA and met-OA, were measured in the uppermost layer. It should be noted that other transformation products may have been formed in our system. However, such compounds were most likely below the limit of quantification ($\leq 9.29$ and $\leq 10.28$ μg L$^{-1}$ for p-Chlorobenzoic acid and 1,2,4-Triazole, respectively), and therefore, they could not be identified. Met-ESA peaked first (day 6) in the vegetated zone. Five days later it appeared in the non-vegetated zone with half of the relative concentration. Yet, during the same phase of the second run no TPs were found. A second peak of both, met-ESA and met-OA with about double the relative concentration measured before, was observed in the vegetated uppermost layer 32 days after the initial injection during the first run. However, in the same period of the second run only met-ESA peaked displaying residual amounts.

According to the correlations performed to the solute time series (Fig. 8), two spots exhibited the strongest relationships: the non-vegetated part of the lowermost layer and the vegetated part of the uppermost layer. These spots coincided with the beginning and end of the transport regime through the system. Boscalid and penconazole did not correlate with Br⁻ in the vegetated part of the lowermost layer and the non-vegetated of the uppermost layer. These results pointed out that probably other processes besides transport (e.g. sorption or plant uptake) dominated the dissipation of these compounds during the experiment. In the middle layers only UR and metazachlor exhibited significant correlations with Br⁻, although metazachlor did not show any correlation in the vegetated zone. These findings suggested that transport was the dominant process for metazachlor in the absence of vegetation, whereas under the influence of plants, it most likely experienced other processes (i.e., plant uptake, mineralization in the roots, sorption and/or transformation). In contrast, UR correlated with Br⁻ in all layers regardless of the presence of vegetation. Therefore, transport was almost certainly a dominant process for this tracer in the pore water. This was not the case for SRB, which only displayed strong positive correlation with Br⁻ in the lower- and uppermost layers. The former was explained by possible high conductivity and low sorption in the gravel. The results from the uppermost layer were associated with the likely promotion of transport towards the vegetated surface, given the strong sorptive character of SRB. Met-ESA also displayed strong positive correlation with the tracers and pesticides in the vegetated zone of the uppermost layer. On the contrary, met-OA did not show any statistically significant correlation. Yet, no conclusion could be drawn for the TPs, given their overall lower amounts detected in the present study.

### 3.3 Spatial and temporal dynamics of transport and dissipation processes: role of vegetation and hydrologic conditions

Spatial and temporal variability of transport and dissipation processes were associated with the conditions prior to injection, the way the solutes entered the system, the presence of plants and the promotion of aeration during the drying phase. Most of the solutes went to the lower- and uppermost layers. The highest concentrations were recorded in the vegetated part of the uppermost layer soon after the injection. This suggested that transport of solutes was favored towards the vegetated surface, what has already been observed in other studies (Holden, 2005; Durst et al., 2013). On the other hand, we speculated that transport was retarded in the middle layers because of the presence of water filled pores before the injection and low pore connectivity.

Metazachlor TPs were only found in the uppermost layer and their maximum relative concentrations were measured in the vegetated part after the promotion of aerated conditions. It should be noted, however, that the process of transformation may have been a function of time, and transport over that time ended in the vegetated part of the uppermost layer. Hence, the uppermost layer (possibly the vegetated part) and the end of the drying phase may have constituted hot spots and hot moments for transformation processes, respectively. Higher sorption activity was attributed to the same layer since a migration of the most sorptive solutes (SRB, boscalid and penconazole) was not observed. In contrast, the most mobile ones (Br⁻, UR and metazachlor) were transported to the non-vegetated part of the middle layers during the drop and subsequent rise of the water table. In the vegetated part of the middle layers only Br⁻ displayed observable amounts (see Fig. 5). This pointed out that besides plant uptake and transformation, retention processes may have also played a major role in the vegetated zone. This hypothesis was in agreement with the amount of SRB found in the sediment (see Fig. 11). It was assumed that sorption velocity was highest at the beginning of the experiment given the rapid decrease in relative concentrations of SRB, boscalid and penconazole. In the later phases, sorption probably decreased due to a decline in the number of free sorption places.

The results of our study underlined the importance of plants in promoting dissipation processes in constructed wetlands. Indeed, plants have already been attributed the ability to facilitate elimination, degradation and retention of pesticides in wetland systems (Liu et al., 2018). However, our findings also suggested that plant roots may be involved in the formation of preferential flow paths, which could result in a rapid transport of contaminants and decrease in the interactions between solutes and sediments (Durst et al., 2013). Plant roots have been related to the creation of discontinuities in the soil profile, greater presence of macropores and occurrence of bypass flow (Ghestem et al., 2011). Therefore, the beneficial impact of plants in terms of elimination, degradation and retention may be reduced by the occurrence of preferential flows.

Our results also indicated that the promotion of aeration facilitated the degradation of some substances. This was in agreement with recent studies that have demonstrated that intermittent flow regimes support aerobic microbial populations and boost degradation rates of pesticides (e.g. Karpuzcu et al., 2013; Maillard et al., 2016). Other authors also found that by alternating drainage with no drainage periods in constructed wetlands, these systems are capable of reducing non-point pollution (Vallée et al., 2015). Hence, the mitigation capacities of constructed wetlands might be improved if aerated conditions in the system are fostered.

### 3.4 Recoveries of solutes at the outlet

The results of the cumulative recovery curves obtained at the outlet are displayed in Fig. 9. During the first flushings, we observed a rapid increase in accumulated mass recovery for Br⁻, whereas the rest of the solutes displayed comparatively slower increases. In the following flushings, the accumulated mass recovery of SRB, penconazole and boscalid gradually increased, while for Br⁻, UR and metazachlor it stabilized. These results suggested that the retained fractions of SRB, boscalid and penconazole in the soil were greater than for Br⁻, UR and metazachlor. Indeed, the sorptive character of SRB, boscalid and penconazole has already been reported in the literature (Long et al., 2005; Vallée et al., 2014; Dollinger et al., 2017). In the fourth flushing, SRB, boscalid and penconazole still exhibited increases in their accumulated mass recoveries (Fig. 9-A1 and B1). Yet, SRB showed higher mass recovery than boscalid and penconazole (Table 7). These findings were associated with the different physico-chemical properties of boscalid and penconazole compared to SRB. According to this, different mechanisms are expected to be involved in their sorption, which will ultimately affect their fate in the environment. SRB has, besides a non-polar region, both charged groups (cationic and anionic). Hence, it is susceptible to sorption onto positive and negative charged mineral sites, OH-groups of hydroxides, clay minerals and organic matter (Kasnavia et al., 1999; Polat et al., 2011). This particular characteristic of SRB must be taken into account when using the tracer to investigate sorption processes of pesticides inside wetland systems.

Cumulative recovery curves of metazachlor TPs were also detected at the outlet during the flushings (Fig. 9). Even though small amounts of TPs were obtained, it was an indication that they were not further degraded or retained in the system, which

was in agreement with the findings of other studies (Mamy et al., 2005; European Food Safety Authority (EFSA), 2008). Higher amounts of met-ESA were recovered compared to met-OA.

According to the total amount of tracers and pesticides recovered at the outlet after the flushings (Table 7), the solutes were classified as follows (from highest to lowest recovery rate): Br⁻ >> SRB >> UR >> Boscalid >> Penconazole >> Metazachlor. Several processes, mostly adsorption by substrates, transformation and plant uptake, were responsible for the removal of tracers and pesticides from the water in our system. Here, it should be noted that the adsorption of the solutes by the substrates could also involve their "temporary removal" from the water, since some of them may be released back into the soil solution. The physico-chemical properties of the compounds have most likely been a determining factor for their dissipation. Vallée et al. (2014) found that a greater retention of pesticides in the soil was related to higher hydrophobic properties (low solubilities and high $K_{oc}$ values). Based on this assumption, we would have expected higher recoveries for metazachlor than for penconazole and boscalid, given its less hydrophobic character. However, unlike boscalid and penconazole, transformation processes played an important role in the dissipation of metazachlor, as demonstrated by the detection of its TPs. On the other hand, given the greater persistence of boscalid and penconazole in the soil (DT$_{50}$ values of 200 and 117 days, respectively), we would have anticipated higher recoveries of these solutes after the flushings. However, only 26.4 and 19% of boscalid and penconazole, respectively, were recovered. Hence, we have hypothesized that the cause of the low recoveries of boscalid and penconazole could have been their high sorption potential and possible plant uptake.

As for the tracers, Br⁻ recovery was the largest, which was in agreement with its most conservative character. The recovery of SRB was also high compared to the other solutes, presumably because this tracer was mostly subject to sorption processes and was more resistant to degradation and plant uptake. This behavior has already been reported in a recent study (Fernández-Pascual et al., 2018). The lowest recovery among the tracers was for UR. In this case, it was assumed that both retention and especially degradation processes were involved in its dissipation. These results were in agreement with field studies performed in wetland systems where recoveries for UR were lower than for SRB and were explained by a higher incidence of degradation processes (i.e., photodegradation) on UR dissipation compared to SRB (Passeport., 2010; Lange et al., 2011; Schuetz et al., 2012).

## 3.5  Final mass balance

The overall mass balance performed at the end of the experiment is shown in Fig. 10. According to the tracer results, Br⁻ was recovered almost fully (98.3 %), while SRB and UR displayed lower recoveries (76 % and 32.4 %, respectively). These findings were similar to those of Maillard et al. (2016) who recovered 97.2, 43.3 and 24.3 % of Br⁻, SRB and UR, respectively in their wetland experiment under batch conditions. Discrepancies with SRB were associated with the large uncertainty in the measurements performed in the sediment of our study. These results were caused by the heterogeneous distribution of SRB in the system, as can be seen in Fig. 11. Indeed, almost 99% of the SRB measured in the vegetated part was located in the uppermost layer.

According to the different compartments, Br⁻ and UR showed the highest recoveries in the outlet water which highlighted their higher mobility. In contrast, SRB was mainly found in the sediment. These results pointed out the different behavior of the tracers when it comes to dissipation. As already evidenced by the breakthrough and cumulative recovery curves, Br⁻ displayed the most conservative character, although some dissipation was observed via plant uptake (16.76% of the total recovered). The main dissipation pathway of SRB was sorption, which was in agreement with the results of other studies (Lange et al., 2011; Durst et al., 2013; Maillard et al.,2016). UR, on the other hand, displayed comparatively lower recoveries, and based on the small amounts found in the sediments, sorption processes were not relevant for its dissipation. Thus, photodegradation and, to a lesser extent, (bio-)chemical transformation were most likely the major dissipation pathways for UR. Indeed, the contribution of (bio-)chemical transformation to UR dissipation has already been reported in other long-term studies (Maillard et al., 2016; Fernández-Pascual et al., 2018; Lange et al., 2018). Due to the likely adaptation of microorganisms to UR degradation after

being exposed in the first run, we would have expected lower recovery rates in the second run (Käss 1998). However, the final recovery values of UR were similar in both runs (31.71 and 29.82% for the first and second run, respectively). Hence, we hypothesized that other substrates for microbial degradation were present in the system and were preferentially utilized limiting the degradation of alternative substrates such as UR. Photodegradation of UR was evidenced by the decrease in its concentration

during saturation in the vegetated part (exposed to light) of the uppermost layer (see Fig. 5). In contrast, no decrease was observed in the part of the same layer that was not exposed to light. Differences were also observed between runs. While in the first run a decrease in the concentration of UR from 17.44 to 12.26 µg/l was detected, in the second run the values decreased from 26.45 to 9.62 µg/l. Assuming first-order decay, we obtained degradation coefficients of 0.05 and 0.17 days$^{-1}$, and half-life times of 13 and 4 days for the first and second run, respectively. These values were comparatively lower than the half-life times

reported in the literature, that are in the range of 11 hours (Leibundgut et al., 2009). However, natural light conditions could not be achieved in the laboratory and this could have limited UR photodegradation. The differences between the first and second run were more difficult to explain, since according to literature the decay rate is inversely correlated to the tracer concentration (Leibundgut et al., 2009). The co-occurrence of photodegradation and microbial degradation could be a possible explanation of the higher decay rate obtained in the second run. However, other factors could not be excluded.

As for the pesticides, while the measurement of metazachlor TPs confirmed that biochemical transformation contributed to its dissipation, the transformation of boscalid and penconazole could not be proven. Yet, it could not be ruled out since the concentration of their metabolites may have been below the limit of quantification. However, considering the duration of the experiment (62.5 ± 2.12 days each run) and the $DT_{50}$ values of boscalid and penconazole reported in the literature (Table 1), probably their transformation was minimal. Plant uptake, on the other hand, could not be determined for the pesticides in the

present study. Yet, it was assumed that it took place given the results of Br$^-$ and the findings of other investigations. For instance, Papaevangelou et al. (2017) demonstrated that high amounts of boscalid accumulated in the tissue of *Phragmites australis* in constructed wetlands, although adsorption accounted as a main process as well. The same plant species was shown to take up, translocate and metabolize tebuconazole (Lv et al., 2017). Traces of metazachlor and its metabolites were also detected in the roots and stems of *Glyceria maxima* in wetland mesocosms experiments (Chen et al., 2017), although plant uptake was reported

to play a negligible role in their removal. Other dissipation pathways, such as mineralisation of the compounds to $CO_2$ or volatilisation from aqueous systems/soil water could not be ruled out. However, they were considered to be minimal according to literature (EFSA, 2008). With this in mind, it can be concluded that retention has played a fundamental role in our study (at least for boscalid and penconazole). Therefore, special attention should be given to retention processes when assessing the mitigation capacities of strongly sorbing pesticides, such as boscalid and penconazole, in wetlands. In these systems the

depletion of the sorption capacities will depend on both, the concentration of the adsorbing substances and the number of sorption places.

The contribution of vegetation to dissipation has also been evaluated in the final mass balance (Fig. 10). Lower amounts of UR and Br$^-$ were recovered from the pore water of the vegetated compared to the non-vegetated zone (Fig. 10-A), however, the differences were not significant. Contrary to expectations, the largest amounts of pesticides and SRB were recovered from the

pore water of the vegetated zone. Yet, these results supported the hypothesis of the promotion of transport towards the vegetated surface. Most of SRB was found sorbed in the sediment of the vegetated zone, where the highest concentration of organic carbon was located (Fig. 11). This may be explained by the susceptibility of SRB to sorption on nonpolar sorption sites of organic matter (Polat et al., 2011). Moreover, it has been recently demonstrated that SRB has high sorption affinity for litters in wetlands (Dollinger et al., 2017). Thus, probably the presence of dead leaves and decaying plant residues in the uppermost

layer enhanced sorption of SRB.

### 3.6  Implications for pesticide mitigation in wetland systems

The replicated conditions in our experiment may resemble those of a groundwater-fed wetland that undergo wet-dry cycles. In principle, we can expect to obtain analogous results in natural systems if similar conditions to those of our study are met. If we compare the characteristics of our experiment (see Table 2) with those of other wetland investigations (e.g. Catallo, 1999; Seybold et al., 2002; Maillard et al., 2011; Gardiner et al., 2012; Passeport et al., 2013; Vallée et al., 2016; Gikas et al., 2018)

we find similar values in terms of sediment texture (values ranging from 4 to 89.5/6.2 to 55/3.8 to 44 for % Sand/Silt/Clay), sediment pH (values ranging from 6 to 8), conductivity (values ranging from 0.45 to 0.9 dS/m) and redox potential (values ranging from -500 to +500 mV). However, there are some discrepancies regarding organic carbon content (values ranging from 2.6 to 32.7 %) and mean residence time (values ranging from 0.5 to 8 days). In this case, the values of our experiment were either below (for the organic carbon content) or above average (mean residence time). Yet, the overall removal rates obtained

in our experiment (see Table 7) were within the same range as those of the wetland studies. For instance, Vallée et al. (2015) found that the removal rates of boscalid in two pilot-scale wetlands ranged from 38 to 67%, whereas Gikas et al. (2018) obtained removal rates for S-metolachlor (pesticide from the same group as metazachlor) that reached up to 92.6% in a constructed wetland planted with Phragmites australis. Other authors have reported removal rates of 45%–90% for tebuconazole (a triazole fungicide similar to penconazole) in wetland systems (Passeport et al., 2013; Tournebize et al., 2013).

A possible explanation for the high removal rates obtained in our experiment could be the fact of having promoted the contact of solutes with the medium through a long period of stagnation (i.e., about two months in each run). In this regard, a recent study performed by Gaullier et al. (2019) has reported almost total mitigation of pesticides and their TPs during stagnation (over 50 days) in constructed wetlands. According to the findings, promoting solute contact with the medium through long periods of stagnation should be taken into account when constructing engineered systems designed to remove contaminants

from the water.

Overall, these observations highlight the importance of certain factors in the removal of pesticides, namely the presence of adequate vegetation, suitable matrix materials, long residence times, low flow rates, intermittent flow conditions, among others (Vymazal et al., 2015; Liu et al., 2018). When these factors are promoted, the removal rates tend to increase, and therefore, the mitigation capacities of constructed wetlands. However, as stated before, the physico-chemical properties of the compounds

will be a decisive factor in their elimination.

### 3.7 Potential of hydrological tracers to evaluate transport and dissipation processes of pesticides in constructed wetlands

In view of the results obtained in the present study, some conclusions could be drawn regarding the use of Br⁻, SRB and UR to evaluate transport and dissipation processes of pesticides in constructed wetlands. In particular, we have corroborated that Br⁻

can be used to elucidate non-reactive transport of solutes, as already reported in the literature (Lin et al., 2003; Małoszewski et al., 2006). But it can also be applied to identify plant uptake (Xu et al., 2004), although to a lesser extent. SRB, has been frequently used to identify sorption processes of pesticides in wetland systems (Passeport et al., 2010; Lange et al., 2011; Schuetz et al., 2012). Yet, its special sorptive character makes it difficult to be compared with a specific type of pesticide. Some authors (e.g. Dollinger et al., 2017) have stated that SRB could be used as a good proxy for hydrophilic and strongly

sorbing pesticides, while others (e.g. Lange et al., 2018) proved that the same tracer closely mimicked the gradual recession of a moderately hydrophobic pesticide in the topsoil of an agricultural field. Our results demonstrated that SRB can serve to approximate the behavior of the pesticides boscalid and penconazole (moderately and highly hydrophobic, respectively) in terms of retention and retardation in the pore water and in the water at the outlet when the constructed wetland is repeatedly flushed. However, it may not be suitable to evaluate overall recoveries of such pesticides at the outlet. In this context, lower

amounts of boscalid and penconazole may be recovered compared to SRB due to the greatest sorption potential of the pesticides and susceptibility to be taken up by the plants. Regarding UR, in terms of transport our results suggested that it may illustrate well the behavior of mobile and non-persistent pesticides, such as metazachlor. This is in agreement with the findings of other

studies (Durst et al., 2013; Maillard et al., 2016; Torrentó et al., 2018). At the same time, our results have underlined that UR may experience not only photodegradation, but also (bio-)chemical transformation, which is consistent with the results of recent investigations (Maillard et al., 2016; Lange et al., 2018, Fernández-Pascual et al., 2018). Yet, UR biodegradation might be limited in the presence of preferred substrates for microorganisms. In any case, it should be noted that the conclusions presented here are only valid if these tracers are used in studies under similar conditions as those of our experiment. That is, constructed wetlands that undergo long periods of stagnation (> 2 months), with drying periods in between, sorbing material with low organic carbon content, similar vegetation and subject to analogous dominant processes.

## 4 Conclusions

The present study introduces a new approach that combines the use of hydrological tracers with different sorptive and reactive properties and high vertical-resolution sampling and monitoring to explore transport and dissipation processes of reactive compounds (i.e., pesticides) inside a wetland system. By comparing tracers with selected pesticides, valuable hints about dominant transport vectors and main dissipation pathways have been collected.

Breakthrough curves obtained at different sampling depths suggested that solute transport was favored towards the vegetated uppermost layer, probably due to lateral transport at or near the surface and/or a possible shortcut effect produced by the roots. Yet, other mechanisms not necessarily related to plants (e.g. fingering) could not be ruled out. Conversely, solute transport was retarded in the middle layers most likely due to the presence of water filled pores before the injection and low pore connectivity in the vicinity of the sampling ports. Spatial and temporal variability of transport and dissipation processes were associated with the conditions prior to injection, the way the solutes entered the system, the presence of plants and the promotion of aeration during the drying phase. The strong positive correlation found between $Br^-$, UR and metazachlor highlighted the predominance of transport processes for these compounds. By contrast, SRB, boscalid and penconazole most likely experienced sorption as evidenced by their absence in the middle layers (in the case of boscalid and penconazole), rapid decrease in their concentrations after the injection and gradual increase in accumulated mass recovery during flushings. Yet, the lower final recoveries of boscalid and penconazole compared to SRB suggested a greater contribution of retention and possibly plant uptake in their dissipation.

The overall tracer mass balance allowed us to identify three dissipation pathways: sorption, transformation and plant uptake. While $Br^-$ was almost fully recovered (98.3 %), SRB and UR displayed lower recovery rates (76 % and 32.4 %, respectively). These results were explained by the greater occurrence of sorptive processes in SRB and transformation in UR. The detection of metazachlor TPs, namely met-ESA and met-OA demonstrated that biochemical transformation played an important role in metazachlor dissipation, whereas no TPs for boscalid and penconazole were found. Yet, their transformation could not be ruled out in the present study. Likewise, plant uptake of pesticides could not be confirmed but it was assumed that it took place throughout the experiment.

Our findings pointed out that the presence of plants and the alternation of different hydrological conditions (saturation and drying periods) may favor dissipation processes. The combination of these factors together with others (e.g. suitable matrix materials, long residence times) could increase the mitigation capacities of wetland systems. Yet, plants might also be involved in the creation of preferential flow paths with the consequent risk of rapid transport of contaminants.

Overall, the complexity of the processes that take place inside constructed wetlands and the lack of sufficient data on a temporal and spatial scale highlights the need to adopt new methods to fully understand the behavior of pollutants in these systems. The application of a multi-tracer approach coupled with high vertical-resolution sampling and monitoring may assist in unveiling

internal mechanisms that dominate transport vectors and dissipation of contaminants. Yet, further experiments need to be done, especially under field conditions combined with modeling.

**Acknowledgements**

This research has been carried out in the framework of the MUTReWa-project (02WRM1366B) funded by the German Federal Ministry for Education and Research. The authors wish to acknowledge Emil Blattmann and Britta Kattenstroth for their help with the construction of the experimental setup as well as Barbara Herbstritt, Jens Robertson, Felix Zimmermann and Maria Martin Pérez for sampling and analytical support. We would also like to thank Marit van Tiel and Sunanth Venkateshwaran for constructive criticism of the manuscript, and Christine Stumpp for her valuable advice and contribution to the design of the sampling setup.

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

## Tables and Figures

35  **Table 1:** Physico-chemical properties of the applied tracers and pesticides (20°C-25°C).

| Property | Unit | UR | SRB | $Br^-$ | Boscalid | Penconazole | Metazachlor |
|---|---|---|---|---|---|---|---|
| Chemical formula | | $C_{20}H_{10}O_5Na_2$ | $C_{27}H_{29}N_2NaO_7S_2$ | NaBr | $C_{18}H_{12}Cl_2N_2O$ | $C_{13}H_{15}Cl_2N_3$ | $C_{14}H_{16}ClN_3O$ |
| Chemical family | | Xanthene dye | Xanthene dye | Inorganic salt | Carboxamide | Triazole | Chloroacetamide |
| Molecular mass [a] | g mol$^{-1}$ | 376.3 | 580.7 | 102.89 | 342.033 | 283.064 | 277.098 |
| Aqueous solubility | g L$^{-1}$ | 25.0 [b] | 70.0 [b] | 850 [c] | 0.0046 [d] | 0.073 [d] | 0.450 [d] |
| Aqueous diffusion coefficient | cm$^2$ s | 3.5 x 10$^{-6}$ [c] | 4.7 x 10$^{-6}$ [c] | - | 4.4 x 10$^{-12}$ [g] | - | - |
| Soil degradation | DT$_{50}$ days | - | - | stable [c] | 200 [d] (persistent) | 117 [d] (persistent) | 8.6 [d] (non-persistent) |
| Dissipation rate on plant matrix | RL$_{50}$ days | - | - | - | 6.9 [d] | 65.6 [d] | - |
| Photolytic stability | DT$_{50}$ days | 0.5 [c] | 34 [c] | stable [c] | 30 [d] (stable) | 4 [d] (moderately fast) | stable [d] |
| Hydrolytic stability | DT$_{50}$ days | stable [c] | stable [c] | stable [c] | stable [d] | stable [d] | stable [d] |
| Water-sediment | DT$_{50}$ days | stable [i] | stable [i] | - | - | 853 [d] (stable) | 20.6 [d] (fast) |
| Organic carbon - water partitioning | Koc L kg$^{-1}$ | 0-62 [b] | 147-498 [b] | - | 772.0 [f] | 2205 [f] | 134.0 [f] |
| Octanol - water partitioning (at pH 7) | Log Kow | 1.26-3.56 [h] | 0.21-4.77 [h] | - | 2.96 [d] | 3.72 [d] | 2.49 [d] |

[-] Information not available
[a] From ChemID database (2017)
[b] From Sabatini (2000)
[c] From Leibundgut et al. (2009)
[d] From Pesticide Properties DataBase, University of Hertfordshire.
[e] From Merck Millipore (http://www.merckmillipore.de)
[f] From PAN Pesticides Database (.pesticideinfo.org/Search_Chemicals.jsp)
[g] From Martin et al. (2017)
[h] From EPA Chemistry Dashboard
[i] From Smart and Laidlaw (1977)

**Table 2:** (A) characteristics and (B) operation of the model constructed wetland system.

A

| Compartment | Parameter | Unit | sub-compartment | Value |
|---|---|---|---|---|
| Sediments | Texture* | % Sand/Silt/Clay | Sand | 97.8/2.3/0.1 |
| | Mean initial organic carbon content * | % | Sand | 0.2 ± 0.02 |
| | Mean final organic carbon content ** | % | Sand | 0.8 ± 1.4 |
| | Mean initial dithionite-extractable Fe (Fed)* | g Kg$^{-1}$ | Sand | 1.0 ± 0.01 |
| | Mean final dithionite-extractable Fe (Fed)** | g Kg$^{-1}$ | Sand | 1.1 ± 0.2 |
| | pH (H$_2$O) | - | Sand | 9.1 |
| | pH (CaCl$_2$) | - | Sand | 8.1 |
| | Diameter* | mm | Sand | 0-2 |
| | | | Gravel | 4-8 |
| | Bulk density* | Kg L$^{-1}$ | Sand | 1.5 |
| | | | Gravel | 1.6 |
| | Porosity | % | Sand | 42 |
| | | | Gravel | 45 |
| | Height | cm | Sand | 32 |
| | | | Gravel | 10 |
| | Surface area | m$^2$ | - | 0.7 |
| | Mass | Kg | Sand | 430.8 |
| | | | Gravel | 124.0 |
| | Redox potential*** | mV | Sand and gravel | -328 ± 10.7 to +740 ± 25.6 |
| | Conductivity*** | dS/m | Sand and gravel | 0 to 0.4 ± 0.1 |

| Plants | Density | | N° | *Typha latifolia* | 4 |
|---|---|---|---|---|---|
| | | | | *Phragmites australis* | 7 |
| | | | Plants m$^{-2}$ | *Typha latifolia* | 10.8 |
| | | | | *Phragmites australis* | 18.9 |
| | Mean initial height | | cm | *Typha latifolia* | 79.8 ± 18.6 |
| | | | | *Phragmites australis* | 76.9 ± 10.1 |

*\* Determined prior to planting*
*\*\* Determined at the end of the experiment as the mean of all the values measured at the different depths*
*\*\*\* Range of values (min. to max) measured in the sediment during the experiment*

**B**

| Parameter | Unit | Value |
|---|---|---|
| Inlet/outlet pumping rate | L h$^{-1}$ | 21.6 |
| Peristaltic pumping rate | L h$^{-1}$ | 0.1 |
| Volume of tracers and pesticides injected | L | 40 |
| Volume of clean water injected at the end of the drying phase | L | 34.1 ± 3.1 |
| Volume of total clean water injected in the flushings | L | 355.1 ± 20.5 |
| Hydraulic retention time | Days | 62.5 ± 2.12 |

*Values represent means ± standard deviation.*

**Table 3:** LOQ/LOD values for the pesticides and TPs.

| Substance | LOD [ng L$^{-1}$] | LOQ [ng L$^{-1}$] |
|---|---|---|
| Boscalid | 0.35 | 1.27 |
| Penconazole | 0.35 | 1.29 |
| Metazachlor | 0.35 | 1.27 |
| Metazachlor-ESA | 2.78 | 10.35 |
| Metazachlor-OA | 0.54 | 1.90 |

**Table 4:** Selected relative concentrations of Br$^-$ measured during the: 1) first and 2) second run for the different zones, phases (saturation and drying) and depths.

| 1) | Depth (cm) | Saturation | | | Drying | | | | | Saturation | | | | |
|---|---|---|---|---|---|---|---|---|---|---|---|---|---|---|
| | | 09/03 | 13/03 | 16/03 | 20/03 | 21/03 | 23/03 | 27/03 | 04/04 | 10/04 | 12/04 | 02/05 | 04/05 | 09/05 |
| Non-vegetated | 3 | 0.08 | 0.17 | 0.19 | 0.34 | 0.41 | - | - | - | 0.33 | 0.35 | 0.21 | 0.17 | 0.03 |
| | 15 | 0 | 0 | 0 | 0.01 | 0.02 | 0.04 | 0.09 | 0.23 | 0.15 | 0.18 | 0.06 | 0.04 | 0.01 |
| | 27 | 0 | 0 | 0.02 | 0.03 | 0.04 | 0.04 | 0.05 | 0.08 | 0.08 | 0.07 | 0.01 | 0.01 | 0.05 |
| | 39 | 0.30 | 0.21 | 0.15 | 0.09 | 0.11 | 0.11 | 0.12 | 0.11 | 0.01 | 0.01 | 0.01 | 0.02 | 0.01 |
| Vegetated | 3 | 0.20 | 0.70 | 0.79 | 0.76 | 0.79 | - | - | - | 0.48 | 0.55 | 0.45 | 0.39 | 0.08 |
| | 15 | 0 | 0 | 0.01 | 0.03 | 0.04 | 0.06 | 0.09 | 0.06 | 0.02 | 0.00 | 0.06 | 0.05 | 0.03 |
| | 27 | 0 | 0 | 0 | 0 | 0 | 0 | 0 | 0.01 | 0.02 | 0.02 | 0.02 | 0.02 | 0.08 |
| | 39 | 0.10 | 0.02 | 0.02 | 0.03 | 0.04 | 0.05 | 0.06 | 0.06 | 0.01 | 0.02 | 0.04 | 0.05 | 0.00 |

| 2) | Depth (cm) | Saturation | | | Drying | | | | | Saturation | | | | |
|---|---|---|---|---|---|---|---|---|---|---|---|---|---|---|
| | | 01/08 | 04/08 | 08/08 | 10/08 | 14/08 | 18/08 | 22/08 | 25/08 | 29/08 | 05/09 | 13/09 | 27/09 | 03/10 |
| Non-vegetated | 3 | 0.07 | 0.20 | 0.46 | - | - | - | - | - | 0.43 | 0.53 | 0.62 | 0.35 | 0.03 |
| | 15 | 0.01 | 0.01 | 0.02 | 0.06 | 0.18 | 0.31 | 0.42 | - | 0.39 | 0.41 | 0.30 | 0.15 | 0.02 |
| | 27 | 0.01 | 0.01 | 0.03 | 0.02 | 0.04 | 0.07 | 0.11 | 0.15 | 0.14 | 0.10 | 0.07 | 0.05 | 0.00 |
| | 39 | 0.29 | 0.19 | 0.19 | 0.19 | 0.20 | 0.21 | 0.21 | 0.20 | 0.02 | 0.02 | 0.02 | 0.02 | 0.00 |
| Vegetated | 3 | 0.18 | 0.59 | 0.53 | - | - | - | - | - | 0.09 | 0.16 | 0.18 | 0.08 | 0.01 |
| | 15 | 0.00 | 0.00 | 0.00 | 0.01 | 0.01 | 0.01 | 0.00 | 0.00 | 0.01 | 0.03 | 0.04 | 0.04 | 0.02 |
| | 27 | 0.00 | 0.00 | 0.00 | 0.00 | 0.00 | 0.01 | 0.01 | 0.03 | 0.06 | 0.04 | 0.03 | 0.02 | 0.00 |
| | 39 | 0.02 | 0.01 | 0.01 | 0.03 | 0.06 | 0.10 | 0.10 | 0.09 | 0.01 | 0.02 | 0.02 | 0.02 | 0.00 |

**Table 5.** Spearman rank correlation of the breakthrough curves between the vegetated and non-vegetrated zones for the different solutes and the first and second run.

| Depth (cm) | | First run | | Second run | |
|---|---|---|---|---|---|
| | | rho | p-value | rho | p-value |
| 3 | **Br⁻ₙᵥ: Br⁻ᵥ** | 0.43 | 0.09 | 0.53 | 0.05* |
| | **URₙᵥ: URᵥ** | 0.30 | 0.26 | 0.33 | 0.25 |
| | **SRBₙᵥ: SRBᵥ** | 0.79 | 0.26 | 0.32 | 0.26 |
| | **Bosₙᵥ: Bosᵥ** | 0.79 | 0.06 | 0.76 | 0.03* |
| | **Penₙᵥ: Penᵥ** | 0.46 | 0.35 | - | - |
| | **Metₙᵥ: Metᵥ** | 0.59 | 0.22 | 0.58 | 0.13 |
| | **Met-ESAₙᵥ: Met-ESAᵥ** | 0.40 | 0.43 | - | - |
| | **Met-OAₙᵥ: Met-OAᵥ** | - | - | - | - |
| 15 | **Br⁻ₙᵥ: Br⁻ᵥ** | 0.49 | 0.06 | 0.63 | 0.02* |
| | **URₙᵥ: URᵥ** | 0.53 | 0.04* | 0.05 | 0.86 |
| | **SRBₙᵥ: SRBᵥ** | 0.51 | 0.04* | -0.01 | 0.98 |
| | **Bosₙᵥ: Bosᵥ** | - | - | - | - |
| | **Penₙᵥ: Penᵥ** | - | - | - | - |
| | **Metₙᵥ: Metᵥ** | - | - | - | - |
| | **Met-ESAₙᵥ: Met-ESAᵥ** | - | - | - | - |
| | **Met-OAₙᵥ: Met-OAᵥ** | - | - | - | - |
| 27 | **Br⁻ₙᵥ: Br⁻ᵥ** | 0.58 | 0.02* | 0.97 | <0.001*** |
| | **URₙᵥ: URᵥ** | 0.85 | <0.001*** | 0.78 | <0.001*** |
| | **SRBₙᵥ: SRBᵥ** | 0.67 | 0.004** | 0.64 | 0.01** |
| | **Bosₙᵥ: Bosᵥ** | - | - | - | - |
| | **Penₙᵥ: Penᵥ** | - | - | - | - |
| | **Metₙᵥ: Metᵥ** | - | - | 0.76 | 0.03* |
| | **Met-ESAₙᵥ: Met-ESAᵥ** | - | - | - | - |
| | **Met-OAₙᵥ: Met-OAᵥ** | - | - | - | - |
| 39 | **Br⁻ₙᵥ: Br⁻ᵥ** | 0.53 | 0.03* | 0.27 | 0.35 |
| | **URₙᵥ: URᵥ** | 0.84 | <0.001*** | 0.95 | <0.001*** |
| | **SRBₙᵥ: SRBᵥ** | -0.06 | 0.83 | 0.73 | 0.003** |
| | **Bosₙᵥ: Bosᵥ** | - | - | - | - |
| | **Penₙᵥ: Penᵥ** | - | - | - | - |
| | **Metₙᵥ: Metᵥ** | 0.40 | 0.44 | 0.76 | 0.03* |
| | **Met-ESAₙᵥ: Met-ESAᵥ** | - | - | - | - |
| | **Met-OAₙᵥ: Met-OAᵥ** | - | - | - | - |

Signif. Codes: 0.001 '***'; 0.01 '**'; 0.05 '*'
nv = non-vegetated; v = vegetated

**Table 6:** Spearman rank correlation between the breakthrough curves of Br⁻ of the first and the second experimental run and the different depths and zones.

| Depth (cm) | Zones | rho | p-value |
|---|---|---|---|
| 3 | Non-vegetated | 0.29 | 0.56 |
| | Vegetated | 0.14 | 0.80 |
| 15 | Non-vegetated | 0.84 | <0.01 ** |
| | Vegetated | 0.55 | 0.17 |
| 27 | Non-vegetated | 0.77 | 0.03 * |
| | Vegetated | 0.26 | 0.53 |
| 39 | Non-vegetated | 0.73 | 0.04 * |
| | Vegetated | 0.55 | 0.15 |

Signif. Codes: 0.001 '***'; 0.01 '**'; 0.05 '*'

**Table 7:** Percentage of tracers, pesticides and their TPs recovered from the outlet water after the flushing of the system and the removal for the first and second run.

| | Recovery water outlet (%) | | Mean recovery (%) | Removal (%) | | Mean removal (%) |
|---|---|---|---|---|---|---|
| | First run | Second run | | First run | Second run | |
| Br⁻ | 76.5 | 76.7 | 76.6 ± 0.1 | 23.5 | 23.3 | 23.4 ± 0.1 |
| UR | 30.3 | 28.8 | 29.6 ± 1.1 | 69.7 | 71.2 | 70.5 ± 1.1 |
| SRB | 36.4 | 38.0 | 37.2 ± 1.1 | 63.6 | 62.0 | 62.8 ± 1.1 |
| Boscalid | 27.9 | 24.9 | 26.4 ± 2.1 | 72.1 | 75.1 | 73.6 ± 2.1 |
| Penconazole | 17.3 | 20.7 | 19.0 ± 2.4 | 82.7 | 79.3 | 81.0 ± 2.4 |
| Metazachlor | 7.5 | 7.4 | 7.5 ± 0.1 | 92.5 | 92.6 | 92.6 ± 0.1 |
| Met-ESA | 6.1 | 5.4 | 5.8 ± 0.5 | - | - | - |
| Met-OA | 0 | 0.8 | 0.4 ± 0.6 | - | - | - |

Mean recovery and removal represent means ± standard deviation.

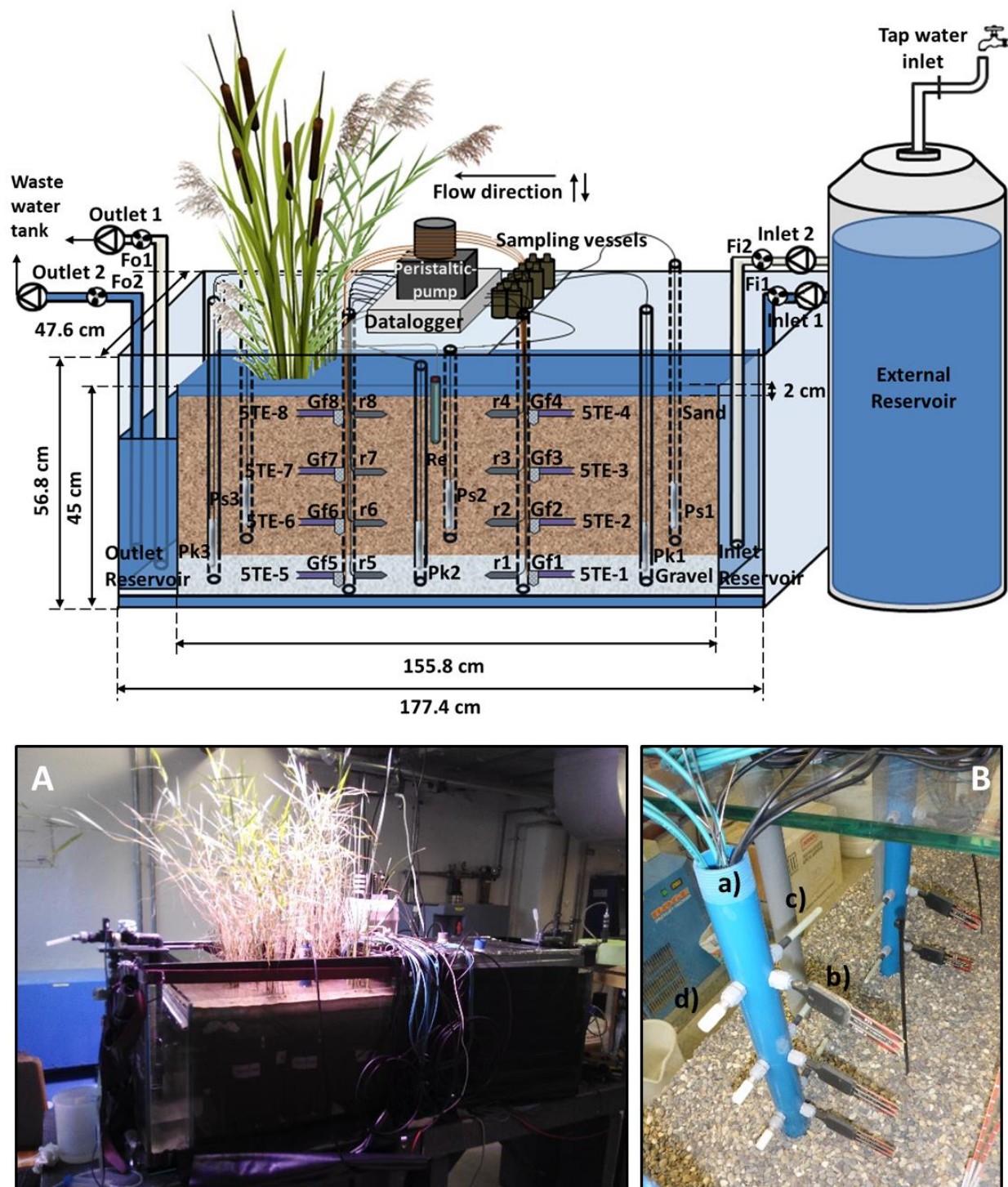

**Figure 1:** Schematic representation of the model constructed wetland system (not to scale). Fi1 and Fi2 indicate the flowmeters at the inlet; Fo1 and Fo2, the flowmeters at the outlet; Ps(n), piezometer in the sand; Pk(n), piezometer in the gravel; 5TE-(n), soil moisture, temperature and electrical conductivity sensor; r(n), platinum redox electrode; Re, reference electrode (Ag:AgCl); Gf(n), glassfilter. For the piezometers, n indicates the position with respect to the inlet; n=1, close to the inlet; n=2, in the middle of the sediment bed and n=3, close to the outlet. For the sensors installed in the multi-level pipes, n indicates the zone and the depths where they are located; n = 1, 2, 3 and 4, non-vegetated zone at a depth of 39, 27, 15 and 3 cm, respectively; n = 5, 6, 7 and 8, vegetated zone at a depth of 39, 27, 15 and 3 cm, respectively. (A) front view photograph of the model constructed wetland system; (B) detail of the multi-level pipes: (a) multi-level pipe at the vegetated zone, (b) 5TE sensor, (c) redox electrode and (d) glass filter.

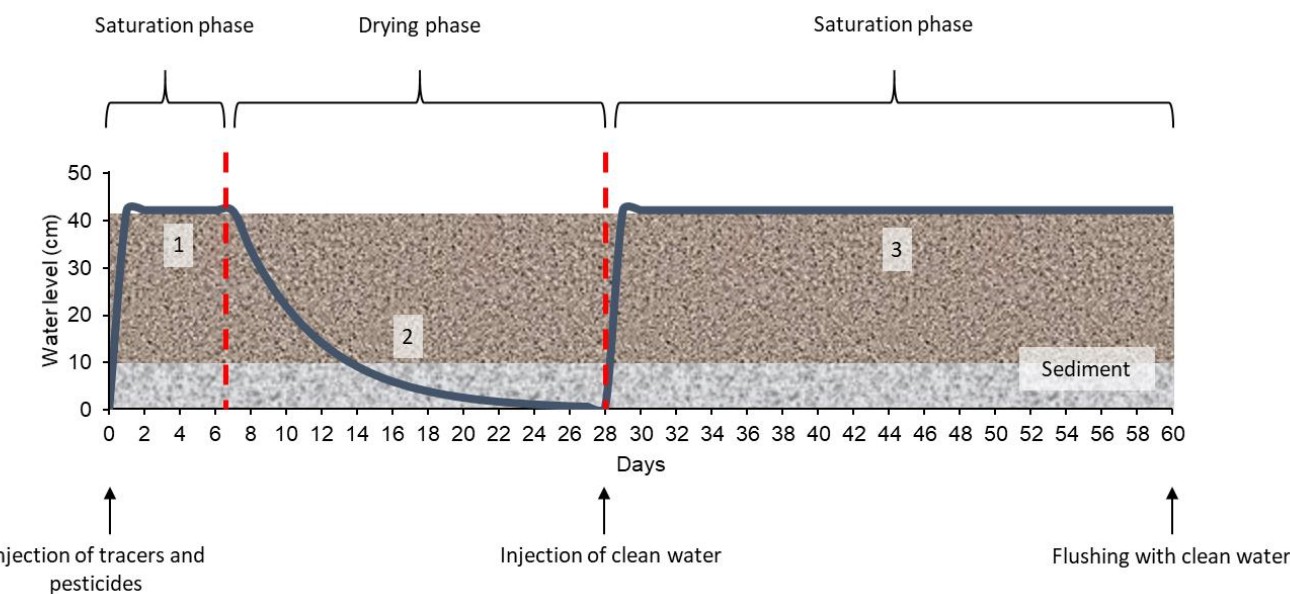

**Figure 2:** Experimental protocol with the different phases and injections performed during the experiment. The x-axis indicates the duration of the experiment and the y-axis the variation of the water level during the different phases.

Note Fig. 2: The water level curve (blue) is only schematic and does not correspond to real water level measurements.

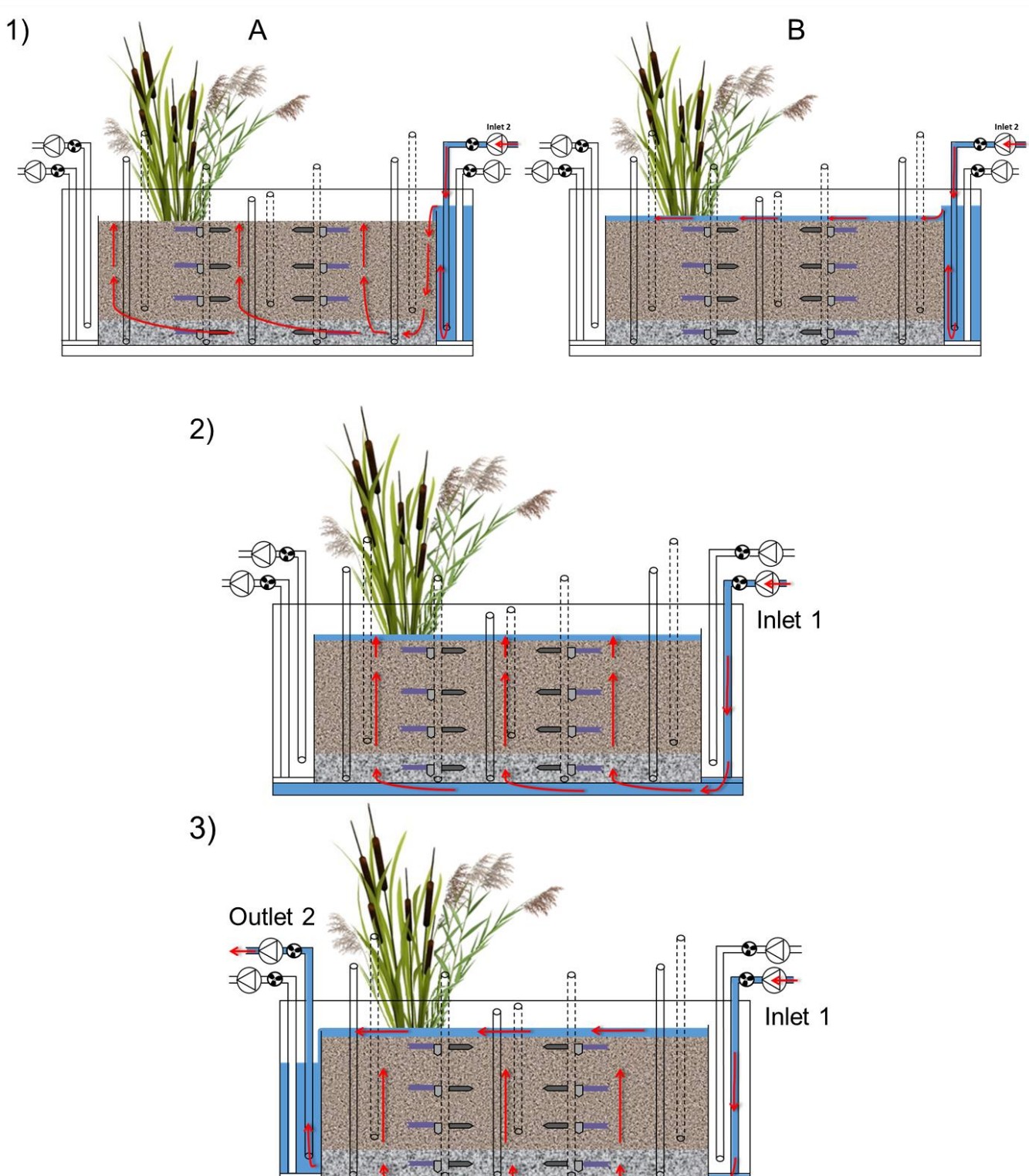

**Figure 3:** Front view of the model constructed wetland system showing the execution of the injections (red arrows indicate the direction of the water flow): (1) surface injection of tracers and pesticides (A) corresponds to the upward vertical filling and B) to the flow on the surface while the ponding was forming), (2) injection of clean water (tap water) from the bottom, (3) flushing of the system with clean water (tap water) from the bottom.

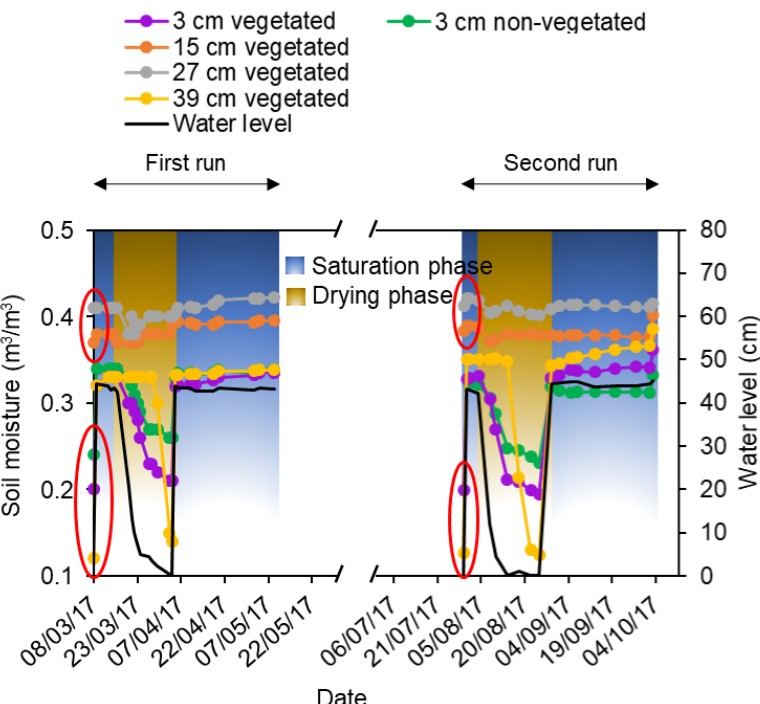

**Figure 4:** Soil moisture values measured in the pore water during the first and the second run for the different zones, phases (saturation and drying) and depths. Water level is displayed in the second y-axis. The missing data from the sampling depths at 15, 27 and 39 cm in the non-vegetated zone is due to failures in the sensors. Red circles indicate the values previous to the injection.

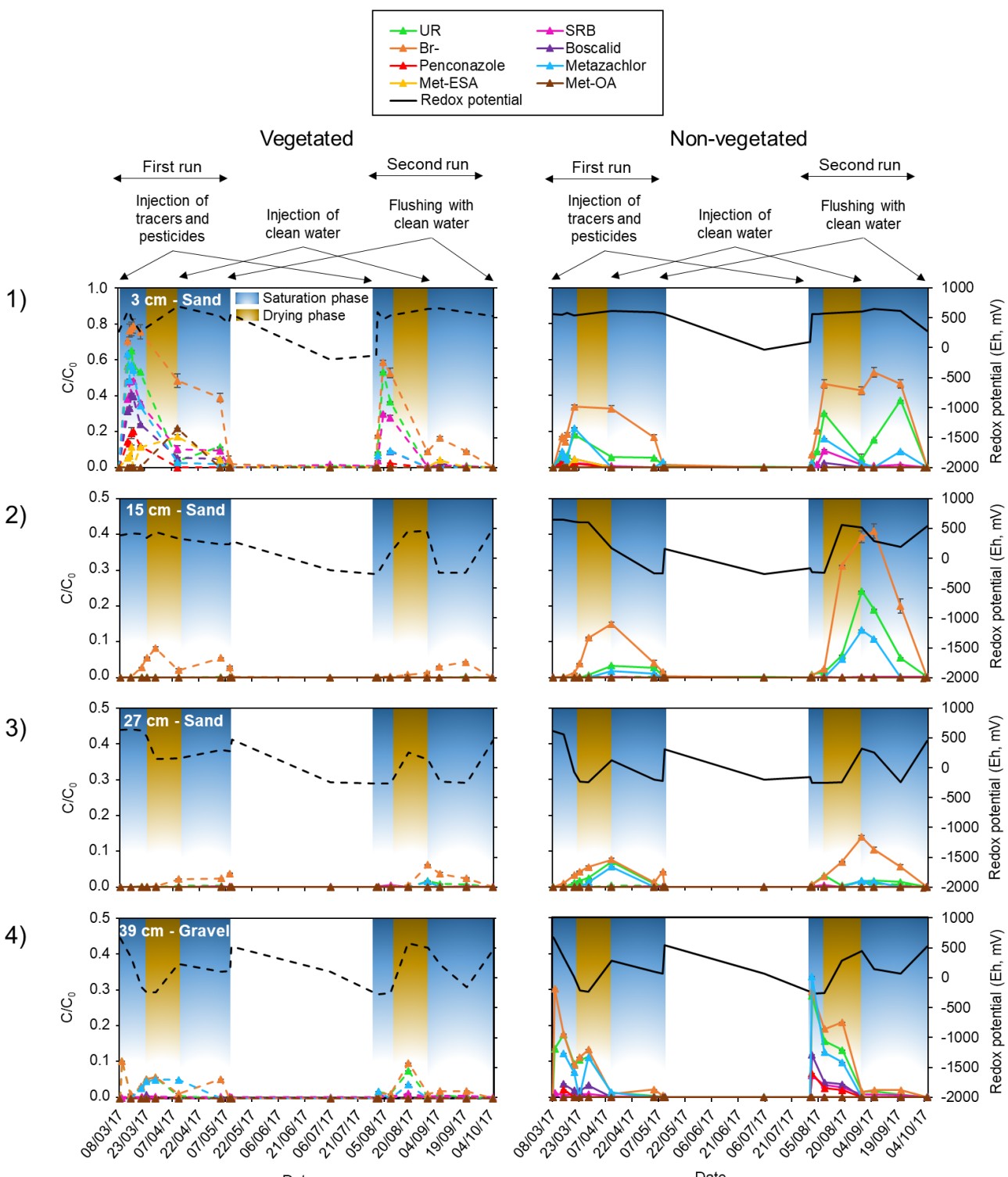

**Figure 5:** Breakthrough curves of the different tracers, pesticides and their TPs in terms of relative concentrations (C/C$_0$) (obtained by scaling with the input concentrations) measured in the pore water during the first and the second run for the different zones, phases (saturation and drying) and depths: 1) 3cm; 2) 15cm; 3) 27cm and 4) 39cm. Changes in redox potential are displayed in the second y-axis (Eh in mV). The different injections performed during each run are displayed on top of the figure. Note that the scale of the relative concentrations corresponding to the sampling depth of 3 cm is extended.

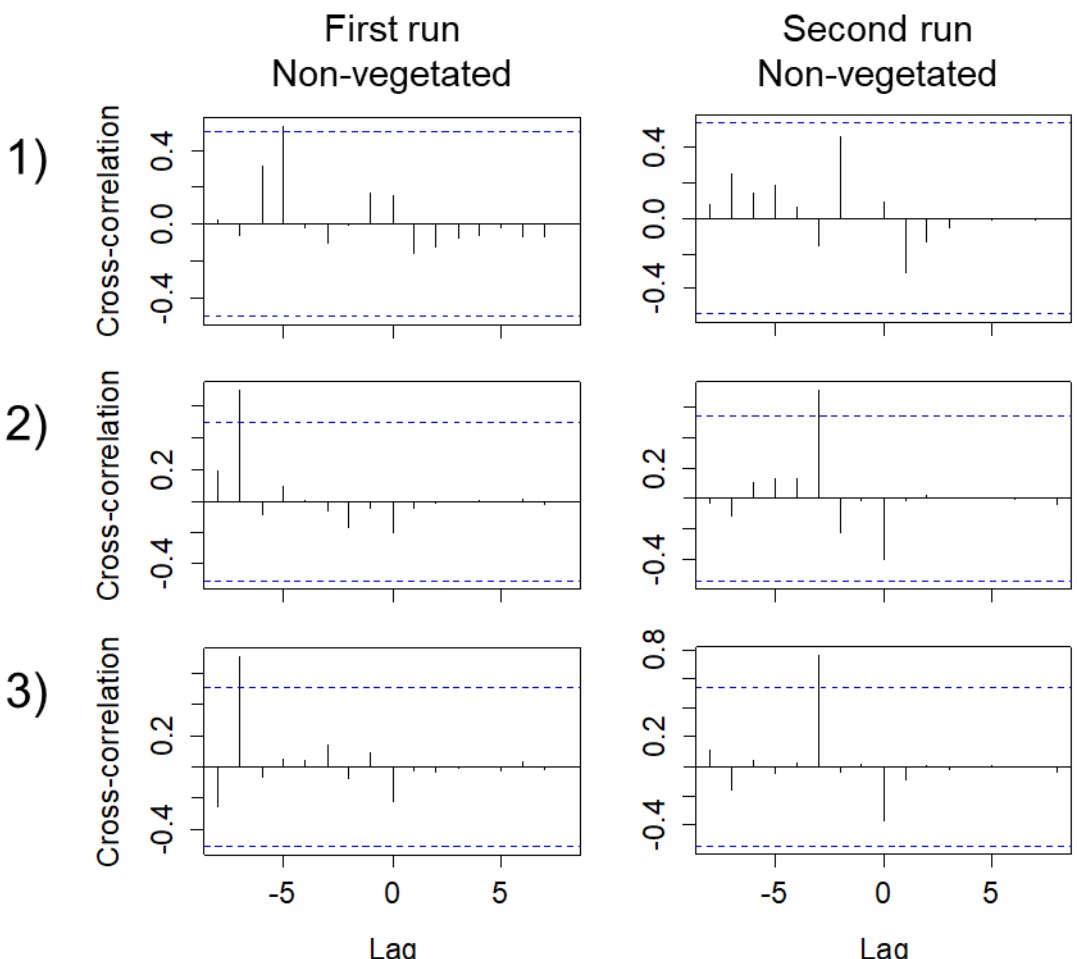

**Figure 6:** Lag analysis performed to the Br⁻ breakthrough curves for the first and second run between the sampling depths: 1) 39cm and 3cm; 2) 39cm and 15cm; and 3) 39cm and 27cm. Only significant lag correlations are displayed. Lag units are given in days (corresponding to the dates when the observations were made). The blue lines represent the approximate 95% confidence intervals.

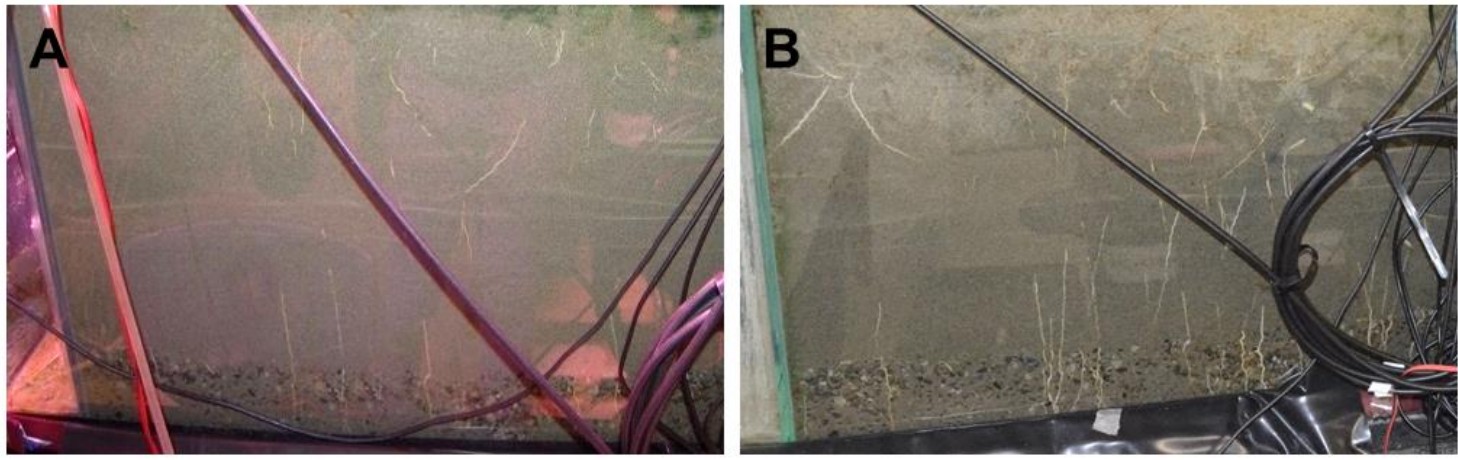

**Figure 7:** Front view photograph of the root system in the vegetated part of the model constructed wetland for: A) before the first run and B) at the end of the second run.

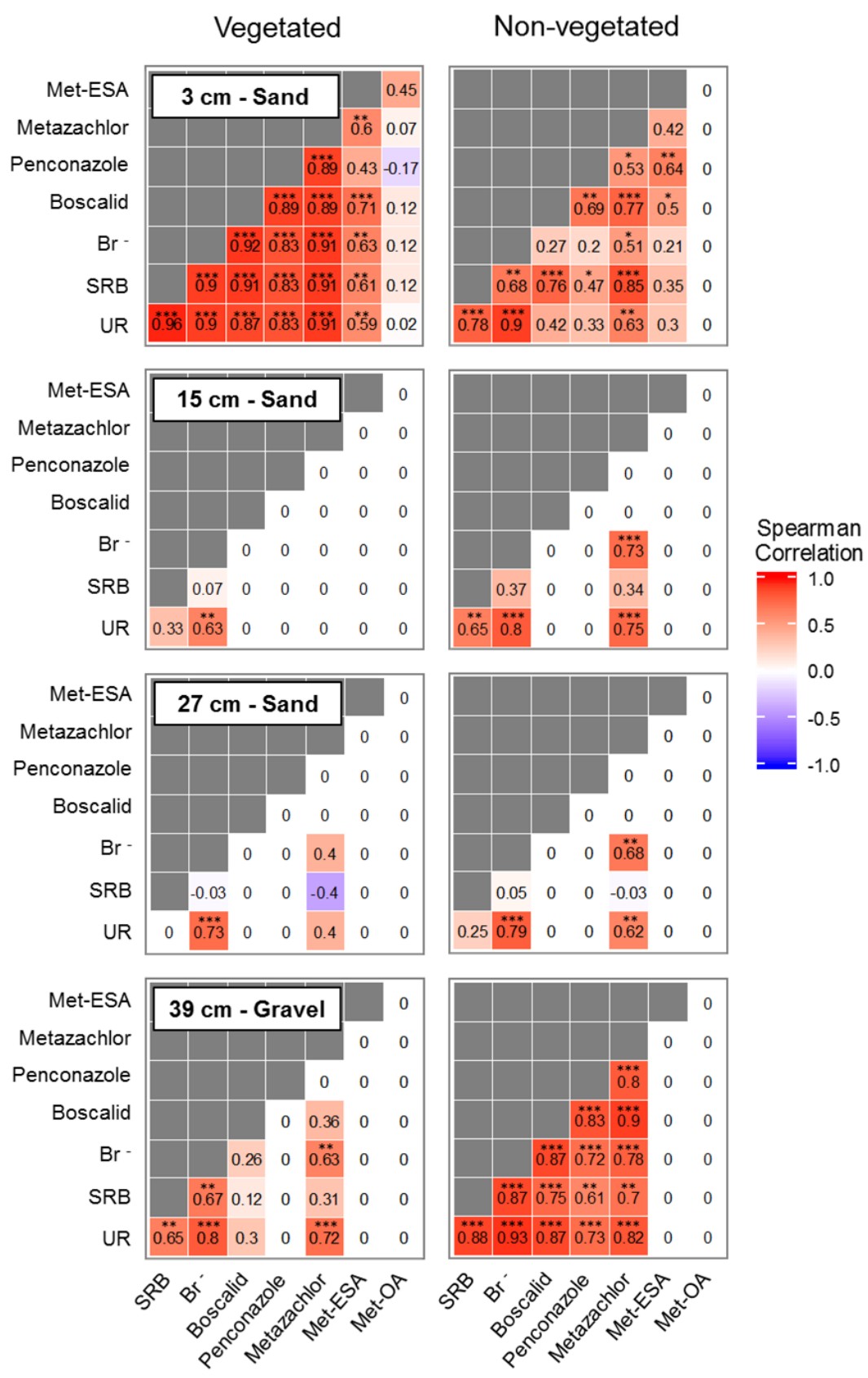

**Figure 8:** Spearman correlation matrices between the relative concentration of tracers, pesticides and their TPs in the pore water during the whole experiment, distinguishing between the different depths and zones.

Signif. Codes:   0.001 '***'; 0.01 '**'; 0.05 '*'

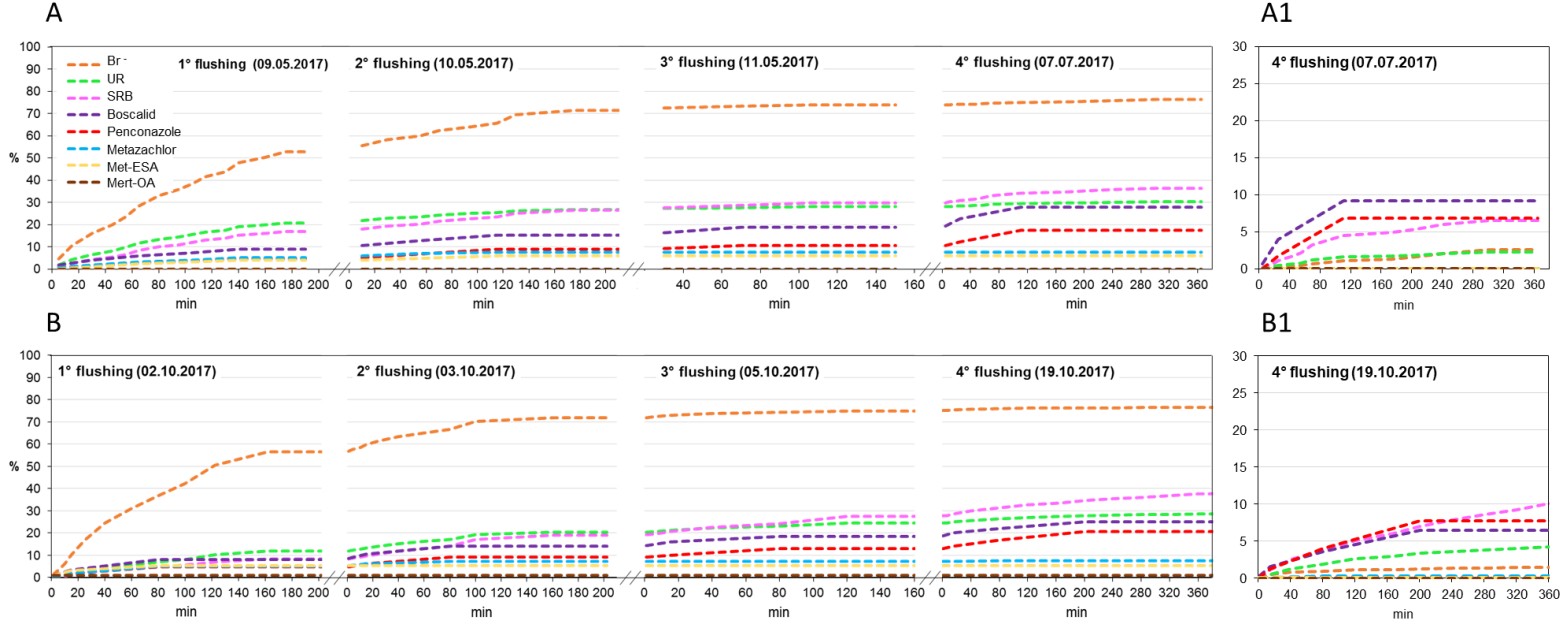

**Figure 9:** Cumulative recovery curves of tracers, pesticides and their TPs during the four flushings for: A) first and B) second run. Recovery curves for the fourth flushing are detailed in: A1 and B1 for the first and the second run, respectively.

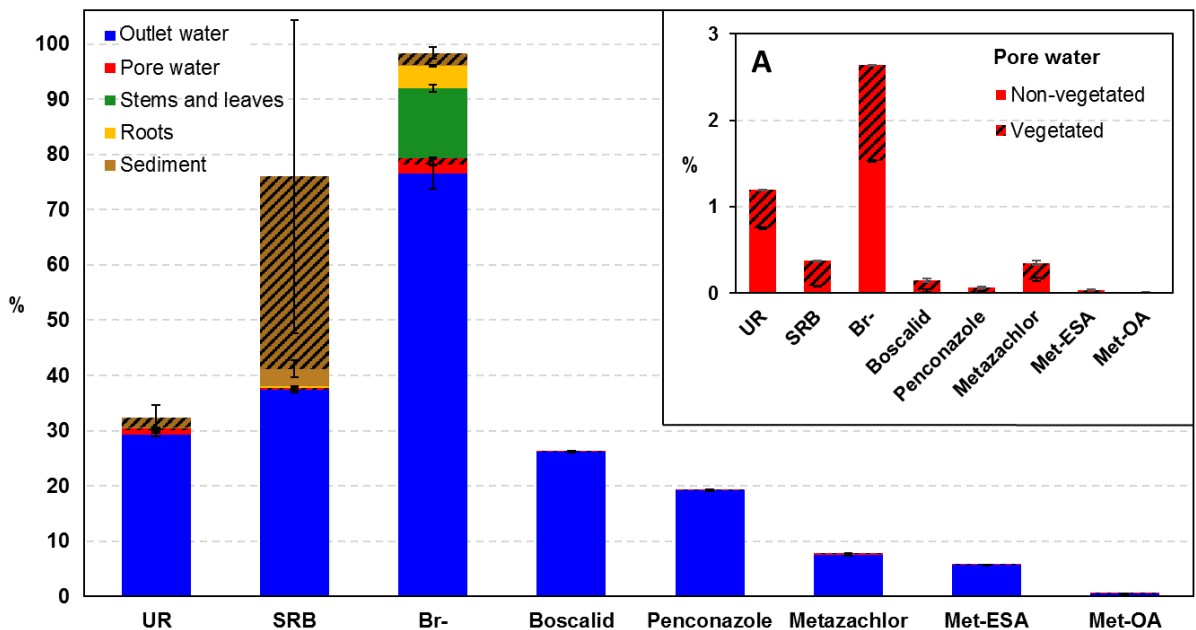

10 **Figure 10:** Final mass balance conducted at the end of the experiment in the different compartments.

Note that the pesticides and their TPs could only be measured in the outlet and pore water compartments. The mass balance for the TPs was calculated according to the total amount of parent compound injected. The shaded area represents the percentage measured in the vegetated zone. The mass balance for the pore water compartment is detailed in the upper right portion of the graph (A).

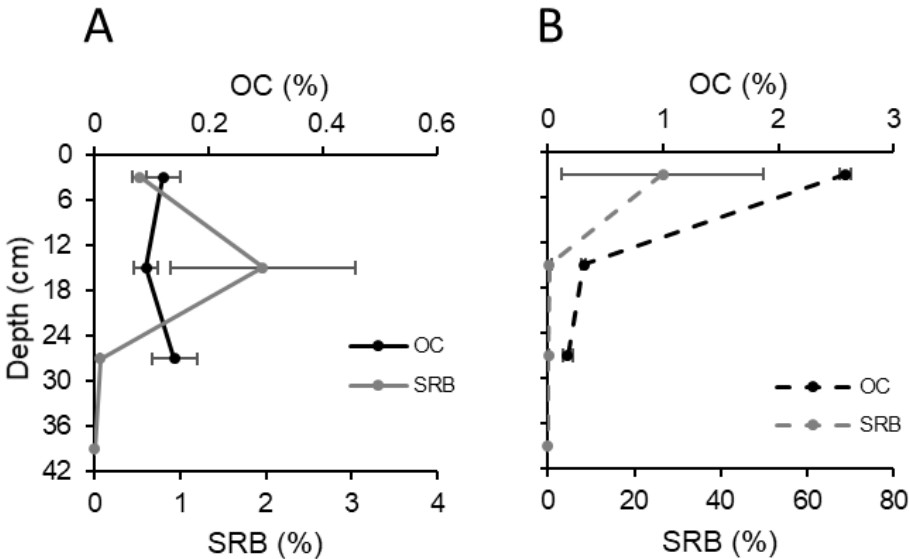

**Figure 11:** Selected vertical gradients of percentage of organic carbon content (OC) and SRB measured in the sediment at the end of the experiment for A) non-vegetated and B) vegetated zone. Values represent means of duplicates ± standard deviation.