# Peer review of "Hydrological tracers for assessing transport and dissipation processes of pesticides in a model constructed wetland system"

_Hydrology and Earth System Sciences, 2019_

## Referee Comment (RC1) · Anonymous Referee #1 · 20 Feb 2019

**Scientific significance**:

The manuscript aims at improving the understanding of the fate of pesticides in constructed wetlands, which are implemented to mitigate pesticide pollution of surface water bodies. To that end, the authors describe in quite some detail findings from a complex laboratory experiment simulating the fate of different (organic) chemicals and $Br^-$ as a conservative tracer (except for plant uptake) in a constructed wetland. To improve with regard to previous studies, the authors have put a lot of effort in obtaining spatial and temporal resolution of the concentrations of their model compounds in the experimental wetland.

Despite the fact that constructed wetlands have some practical relevance as mitigation measures, the scientific relevance of the manuscript seems to be limited. On p. 2, L. 5 – 9, the authors describe their objectives. However, in the current form they are very specific to the experimental design and it remains unclear (also subsequently in the manuscript, see also comments below) how answers to the posed question can be generalised:

- How to gain general insights if one knows in detail the spatial and temporal patterns of pesticide fate processes in this particular wetland at the lab scale (refers to objective i))?

- How to generalise the findings related to the different behaviour of the model compounds (refers to objective ii))?

- How to generalise the results regarding vegetation and hydrologic conditions (refers to objective iii))?

My statement does not imply that no such general insight could be gained from the experiment. However, in order to do so, one would need to ask first general scientific questions and subsequently demonstrate how the experiment can provide such generalizable answers. Such questions however are missing. The sentence on p. 2, L. 3 – 4, is too vague in this respect. This limitation is subsequently reflected in the Conclusion section. There is a lack of novelty and the statements are either very general or too speculative.

One way how the generality could be improved would for example be to put the characteristics of the study wetland (texture, organic carbon content, water residence time, redox conditions etc.) into the context of real-world wetlands, to reflect – based on scientific theory – what follows for pesticide retention in such wetland and to demonstrate respective insights that go beyond prior knowledge. I missed such information in the manuscript.

**Scientific quality:**

Overall, the manuscript indicates that the experiments were carefully planned and executed. There are few technical questions that are listed below.

However, there are conceptual limitations that also relate to the comments on the scientific significance above. A major issue is the lack of replication. There is only one vegetated and one non-vegetated chamber of the experimental tank. I am aware of the effort needed to carry out such experiments and to build such experimental facilities. Nevertheless, the results and conclusions hinge solely on single realisations of two experimental treatments. Especially in the context of preferential flow phenomena, this may be very critical because a single connected flow paths may exert a strong effect on the overall outcome. Without replication, it is very difficult to judge the robustness of the differences observed between the two treatments.

Another limitation is the lack of quantitative analyses that could link the different pieces of information. The authors report for example $K_{oc}$-values for the different compounds from the literature but do not provide quantitative analyses how transport and concentrations levels were expected based on this information. I also missed key features such as expected hydraulic residence time in the system etc. One could probably calculate such things from the information in the text and Tab. 2 but it would be useful for readers to directly get such information.

**Presentation quality:**

In general, the paper is clearly written and the findings are carefully presented in the figures and tables.

**Detailed comments:**

*Title:* Use of tracers: Why do you distinguish between tracers and pesticides? Uranine and sulforhodamine B are organic chemicals as are the three pesticides used in the study. Of course, there is a difference in the use of the compounds, but why is this

distinction relevant for elucidating the fate of the pesticides (given the fact that also these tracers undergo sorption plant uptake and degradation)?

*Abstract:*

p. 1, L. 10: What are spatial and temporal mechanisms? p. 1, L. 13: What was the rationale behind the selection of these compounds? p. 1, L. 16 – 17: What do you mean by the statement that transport dominated for some compounds? p. 1, L. 17 – 18: What other dissipation processes could be expected? This statement is not very informative.

*Introduction:*

p. 1, L. 27: The reference is not very recent. Many others are available representing more current findings. p. 1, L. 28 – 29: Generally, transformation products are less toxic. There are exceptions but the wording may be misleading. p. 1, L. 40: This is an important aspect. Unfortunately, this manuscript does not really elaborate any further on this topic. It would be interesting to learn how the results reported here relate to other studies and what the results imply for mitigation capacities. p. 2, L. 13 – 18: The critical question about the compound selection is what insight can be gained. In the result section (p. 9, L. 1 – 20), the results about the compound-specific differences are summarised. The reported findings basically reflect the knowledge already used for making the compound selection. Hence, the authors miss to derive more general insight that goes beyond the prior knowledge.

*Methods:*

p. 2, L. 28: How reliable is terbutryn as an internal standard for the other pesticides? p. 4, L. 8: Generally, glass bottles are used for storing pesticide samples. p. 4, L. 30: What about possible inferences with the fluorescence of the background matrix? p. 5., L. 4: Where are LOQ/LOD provided? p. 5, L. 12: How can an independent background be determined? p. 5, L. 35 – 36: This sentence sounds strange because transport

processes affect all compounds irrespective whether or not they are sorbed or not (or degraded or not).

*Results and discussion:*

p. 6, L.10: What means an early breakthrough? Early compared to what? p. 6, L.10 – 30: These paragraphs list different findings without a clear structure and logic. p. 6, L. 37: Where can one see these redox conditions? p. 7, L. 6: I assume that sorption takes place all the time and not only during the initial phase. p. 7, L. 18: Where can one see this correlation? p. 7, L. 20 – 30: These sentences are confusing. p. 7, L. 30 – 33: This paragraph is not well linked into the structure. p. 8, L. 11 – 13: Please be aware that different transformation products may have different source terms because they are generally formed at different rates and possibly in different parts of the subsurface. p. 8, L. 22 – 24: This is very qualitative. What were the expected compound-specific differences solely based on the $K_{oc}$-values? p. 8, L. 25 – 26: Again, this statement appears rather isolated in the text. p. 9, L. 31: Here you contradict yourself: above you have argued that SRB is expected to be strongly sorbed because of its $K_{oc}$-value (p. 8, L. 23)

*Figures:* Fig. 4: - It is difficult to distinguish all the different lines. - What were the hypotheses how the breakthrough would differ between the different depths and the different compounds? Fig. 7: Is there no differentiation between vegetated and non-vegetated treatments? Fig. 8: You might consider comparing the two treatment with separate bars. Fig. 9: Is the sorption consistent with $K_{oc}$-values known for SRP?

---

## Referee Comment (RC2) · Anonymous Referee #2 · 8 Mar 2019

The authors present an interesting and well written paper on transport and dissipation processes of different substances in a constructed wetland at the lab scale. This is a highly relevant topic that is within the scope of HESS and of interest for a broader audience. The experiments and results are highly interesting and are largely presented clearly. A few points in the analysis and interpretation, however, should be revised to be less speculative and more supported by the results. This requires mainly further elaboration of the discussion, as detailed in the specific comments below.

**1 General comments**

1. Recovery

   A main concern is the low recovery of most of the substances. Except for bromide and SRB, less than a third of the applied masses were detected in the investigated compartments (the data on SRB suggests a recovery between 48 and 105 % - Fig. 8). Because it is ambiguous to judge the parts that have not been observed, care needs to be taken in drawing conclusions on transport and dissipation from the data. The authors often did well in this regard, and addressed possible pathways of the substances' fates by deduction and use of available literature. Sometimes they overachieved a bit, and some aspects deserve further clarification.

   I would appreciate if the authors discussed possible reasons for low recovery in more detail. What about the formation of other transformation products – is this likely, are other TP known that might be formed under the given circumstances? If sorption is a major pathway, why have the substances not been detected in sediment/plants? Which other pathways are possible, especially for the substances that are not likely to be adsorbed or degraded? Can the expected degradation/mineralisation be quantified using literature values, and contrasted with the measurements?

   With regard to the transformation/degradation of UR: How much of the degradation was possibly due to photolysis in the inlet container or at the system's surface? Could you estimate photolysis rates quantitatively? Microbial degradation of UR seems not to be enhanced after the first application, as illustrated by the similar recovery rate in the second part. If the system has not been exposed to UR before, and microbial decay was a major pathway, the first application of tracer probably would have fostered the microbial community capable of degrading the dye (Käss 1998), which would have enhanced microbial decay in the second run. Can alteration of fluorescent properties be a reason for apparent loss of UR (pH values are given in Table 2, but apparently only one measurement)?

The different substances were mixed in one solution for application, but possible interactions of the applied substances are not discussed. Can interactions of the different organic compounds be ruled out? Testing a reference sample of the injection solution repeatedly over time could give further hints on interactions or degradation rates of the mixture.

2. Flow paths and preferential flow

The flow paths during the tracer application are not completely clear to me. From Fig. 3 it appears as if the tracer solution was applied to the surface near the inlet container by letting the inlet container overflow. The arrows indicate vertical movement downwards near the inlet, and vertical movement upwards elsewhere. Is the ponding on the surface from surface flow from the inlet, or from upward flow through the soil? In case of the former: Has air been entrapped in the system?

A great part of the transport in the experiment has been attributed to preferential flow in the upper and lowermost layer. Given the coarse texture of the soil, hydraulic conductivity will be high and lead to fast regular flow rates already. Is the observed breakthrough still considered preferential when compared with expected flow rates using conductivity and hydraulic gradient? If preferential flow is an issue, how would that influence the spatial distribution of substances in the sediment, and in turn the recovery of substances from sampling sediment?

3. Correlation analysis

Much of the interpretation is based on a correlation analysis. Please describe in bit more detail what was correlated – I expect you used measured concentration time series? Sorption is significant for some of the substances. How would retardation affect the results of the correlation analysis?

**2  Specific comments**

P6, LL 33-35: This part is unclear. How does the design of the inlet cause preferential flow towards the bottom? And what is meant with "plants channel flow to the surface" – flow from lower layer to the soil surface? Or do you mean enhanced infiltration from the surface?

P6, LL39-40: Consider rewriting sentence. Br$^-$ had almost complete recovery and was found in plants and roots, so you may delete "possibly", and refer to Fig. 8 and not only the lack of measured Br$^-$ in pore water.

P7, L4: "Early Breakthrough" – compare with expected flow velocity (see comment above)

P7 L12: "absence of BTC in middle layer" and "early BTC in uppermost layer" "confirmed the influence of plants" is too strong as a statement. Other explanations are possible for these observations – preferential flow without the influence of plants (fingering), bias in the observations, etc.

P7 L 14 "evidenced" – too strong as well. It might be a hint, but could also be that the degradation is just a function of time, and transport over that time ended in the vegetated part, opposite from the inlet.

P7 LL23-27/Table 4: How significant are the differences in recovery of Br$^-$ given in Table 4, which is the basis for your argumentation here? The differences do not appear large enough to justify the conclusion.

P8, L 36: Please explain "low leaching potential" as a property of a substance – does that mean high sorption?

P9, L 38: "could be identified" – an unambiguous identification was unfortunately not possible in the experiment, but valuable hints / indications were collected

P10, L 5: "biochemical transformation had a major contribution" – only <10 % of the parent substance were found as TP, so it is not possible to say which was a

major contribution

Fig 4: How do you explain the obvious differences in Br$^-$ breakthrough between the first and second run? Fig 7: Recovery of TP in % - how can the total amount be known?

Fig. 8: Please comment on the large error bar for SRB, which indicates recovery to be between 48 and 105 %. Would more sediment samples have reduced this uncertainty? How does this uncertainty influence your interpretation?

**3 Technical comments**

P2, L32: 100 mg L -> 100 mg L$^{-1}$

P2, L35: Please give the dimensions of the constructed wetland system also without inlet/outlet.

P5, L11: resulted curves -> resulting curves

P5, L13, and elsewhere throughout the text: Br -> Br$^-$

P7, L26: "was most likely" -> "were most likely"

P8, L28: "were classified": classified for what (recovery rate, I presume?)

Fig 4: Consider duplicating the figure and display vegetated and non-vegetated parts separately, which would make distinguishing these parts a lot easier

**4 References**

Käss, W. (1998): Tracing technique in geohydrology, 581 pp., Balkema, Rotterdam, The Netherlands.

---

## Author Response (AR1)

**Response to the review of the manuscript: Hydrological tracers for assessing transport and dissipation processes of pesticides in a model constructed wetland system**

Elena Fernández-Pascual [a] *, Marcus Bork [a, b], Birte Hensen [c], Jens Lange [a]*

Hydrology, Faculty of Environment and Natural Resources, University of Freiburg, Freiburg, Germany [a]
Soil Ecology, Faculty of Environment and Natural Resources, University of Freiburg, Freiburg, Germany [b]
Institute of Sustainable and Environmental Chemistry, Leuphana University Lüneburg, Lüneburg, Germany [c]

* corresponding author, elena.fernandez@hydrology.uni-freiburg.de

**Answers to editor**

[Figure]

Dear Prof. Zehe,

We are very grateful to have been given the opportunity to revise and resubmit our manuscript. We greatly appreciate your comments and suggestions. All of them have been taken into account and we now trust that the quality of the manuscript has improved considerably. We would also like to acknowledge the two reviewers for their constructive and helpful comments.

Below we indicate your evaluation of the manuscript and a point by point explanation (in blue) of how we have addressed your comments and those of the reviewers. For the latter ones, we have specified in some of the comments the corrections performed and where in the revised manuscript (in the not marked version) they have been made, detailing the corresponding page and line number where applicable. Additionally, we have indicated the new literature included in the revised version. The marked-up manuscript version is provided at the end of this document.

We hope you find these revisions rise to your expectations. Thank you once again for taking the time and energy to help us improve the paper.

On behalf of all the authors,

Elena Fernández-Pascual

**Editor**

**-------------------------------------**

**Comment:** After reading your study again, I am, in line with both reviewers, impressed by the high quality of the underlying experiments. Nevertheless, both reviewers came up with a long list of critical and constructive comments which need to be thoroughly addressed in a round of major revisions.

1. Reviewer 1 recommended rejection – her/his main critique is that it is not clear how the findings of your experiments can be generalized and the lack of replication. In your reply you better explained that the overall objective of your study is to introduce the approach and more specifically to show the potential of fluorescent tracers to explore the interplay of transport and transformation of reactive compounds. Seems like a good idea, however, this implies that the story line of the manuscript needs to be strongly revised. Moreover, I think that it is very much worth to consider her/his suggestion of "Putting the characteristics of the artificial wetlands (texture, organic carbon content, water residence time, redox conditions etc.) into the context of real-world wetlands, to reflect – based on scientific theory – what follows for pesticide retention in such wetland and to demonstrate respective insights that go beyond prior knowledge."

2. I see that the lack of replicates is a weakness and that the abundance/absence of a single macropore will at this scale drastically change outcome of the experiment (you operate below the REV!). A proper design would need at least three replicates. But I do not think that it is a killing argument, if the story line will be revised as you proposed. An alternative is the combination with modelling to overcome this limitation. Solute transport has a theory, means that we are not thrown to a pure statistical learning paradigm.

3. While reviewer 2 recommended minor revision, her/his main critique points deserve much attention as well. A mass balance closure of 30% leaves us with much uncertainty about the entire concert of transport and transformation. In a field study this would be a nice achievement, but for a controlled environment it is poor, and this puts the value of these experiments into question (also with respect to the issue of being representative and working below the REV). This needs to be discussed.

4. Given coarse texture of the soil, a rapid reaction does not necessarily speak for preferential flow! This is surely related to fuzzy definition of preferential flow. Is transport in the "near field" preferential flow (transport times < than lateral mixing times) or is it a non-Gaussian residence time distribution? I think the manuscript will benefit from being precise in this respect, maybe a straight forward model exercise with could help to quantify the degree of non-normality of the breakthrough curves. Last but not least I think it is worth to discuss the role of retardation (which introduces a time lag) when doing a correlation analysis of breakthrough curves. One could thus also infer on the retardation using lag correlations?"

**Response:**

1. As suggested, the story line of our manuscript has been thoroughly revised. Some subsection have been completely rewritten (3.2 and 3.3) and new ones have been

created (3.1 and 3.5). Likewise, the results have been compared with similar investigations and the characteristics of our experimental system have been put into the context of real-world wetlands. In this regard, we have detailed under which circumstances our results would be valid and what would be the implications for the mitigation capacities of real-world wetlands. All this is detailed below in the responses to the reviewers' comments.

2.  We still believe that the realization of two identical runs provides equally valid results. Moreover, our system was not the typical laboratory experiment, but rather a system halfway between controlled laboratory and field conditions. In addition, performing more replicates of such a complex experiment was beyond our financial possibilities.

3.  We agree that the recoveries of some of the tracers were small. But we don't think it was due to uncertainty but rather due to the reactive character of such tracers and the type of experiment (long duration and stagnant conditions) since the most conservative tracer (bromide) was recovered almost entirely (98.3%). In fact, and as detailed in the revised version, recoveries of the same tracers in other studies were similar. Therefore, we believe in the validity of our results. We have widely discussed this point and the possible reasons for such recoveries in the revised version.

4.  We also concur with the view of the editor in his opinion about the occurrence of preferential flow in our system. Indeed, considering the type of sediment used in our experiment, it is very unlikely that preferential flow towards the bottom during the injection will occur. That is why we have eliminated such statement. However, we still believe that transport of solutes may have been favored towards the vegetated surface since there are strong indications that support it. This point has also been discussed extensively in the revised version. In addition, after carefully reviewing the data, we have concluded that the conditions prior to injection (i.e., system at field capacity) have influenced the transport of solutes as well. Specifically, we believe that the delay in the middle layers has been due to the existence of water-filled pores, which has been associated with a possible lack of connectivity. These statements have been discussed in the revised version and are supported by additional information (new Table 4 and new Fig. 4, see below), including a lag correlation analysis, as suggested (new Fig. 6, see below).

    Finally, we would like to point out that we have not performed the classical tracer test under steady state conditions but rather a more complex experiment with different phases of variable duration, stagnant conditions and separate measurements over time. We believe that a simple modeling exercise cannot be applied but rather a more specific model capable of explaining the complexity of our system. Therefore, and as already stated in the response to the reviewers comments, although we are aware of the importance of modelling (it is intended to be carried out in the future), we think that including it in the manuscript would enlarge it too much and would go beyond the scope of the study.

**Table 4.** Selected relative concentrations of Br⁻ measured during the: 1) first and 2) second run for the different zones, phases (saturation and drying) and depths.

| 1) | Depth (cm) | Saturation | | | Drying | | | | | Saturation | | | | |
|---|---|---|---|---|---|---|---|---|---|---|---|---|---|---|
| | | 09/03 | 13/03 | 16/03 | 20/03 | 21/03 | 23/03 | 27/03 | 04/04 | 10/04 | 12/04 | 02/05 | 04/05 | 09/05 |
| Non-vegetated | 3 | 0.08 | 0.17 | 0.19 | 0.34 | 0.41 | - | - | - | 0.33 | 0.35 | 0.21 | 0.17 | 0.03 |
| | 15 | 0 | 0 | 0 | 0.01 | 0.02 | 0.04 | 0.09 | 0.23 | 0.15 | 0.18 | 0.06 | 0.04 | 0.01 |
| | 27 | 0 | 0 | 0.02 | 0.03 | 0.04 | 0.04 | 0.05 | 0.08 | 0.08 | 0.07 | 0.01 | 0.01 | 0.05 |
| | 39 | 0.30 | 0.21 | 0.15 | 0.09 | 0.11 | 0.11 | 0.12 | 0.11 | 0.01 | 0.01 | 0.01 | 0.02 | 0.01 |
| Vegetated | 3 | 0.20 | 0.70 | 0.79 | 0.76 | 0.79 | - | - | - | 0.48 | 0.55 | 0.45 | 0.39 | 0.08 |
| | 15 | 0 | 0 | 0.01 | 0.03 | 0.04 | 0.06 | 0.09 | 0.06 | 0.02 | 0.00 | 0.06 | 0.05 | 0.03 |
| | 27 | 0 | 0 | 0 | 0 | 0 | 0 | 0 | 0.01 | 0.02 | 0.02 | 0.02 | 0.02 | 0.08 |
| | 39 | 0.10 | 0.02 | 0.02 | 0.03 | 0.04 | 0.05 | 0.06 | 0.06 | 0.01 | 0.02 | 0.04 | 0.05 | 0.00 |

| 2) | Depth (cm) | Saturation | | | Drying | | | | | Saturation | | | | |
|---|---|---|---|---|---|---|---|---|---|---|---|---|---|---|
| | | 01/08 | 04/08 | 08/08 | 10/08 | 14/08 | 18/08 | 22/08 | 25/08 | 29/08 | 05/09 | 13/09 | 27/09 | 03/10 |
| Non-vegetated | 3 | 0.07 | 0.20 | 0.46 | - | - | - | - | - | 0.43 | 0.53 | 0.62 | 0.35 | 0.03 |
| | 15 | 0.01 | 0.01 | 0.02 | 0.06 | 0.18 | 0.31 | 0.42 | - | 0.39 | 0.41 | 0.30 | 0.15 | 0.02 |
| | 27 | 0.01 | 0.01 | 0.03 | 0.02 | 0.04 | 0.07 | 0.11 | 0.15 | 0.14 | 0.10 | 0.07 | 0.05 | 0.00 |
| | 39 | 0.29 | 0.19 | 0.19 | 0.19 | 0.20 | 0.21 | 0.21 | 0.20 | 0.02 | 0.02 | 0.02 | 0.02 | 0.00 |
| Vegetated | 3 | 0.18 | 0.59 | 0.53 | - | - | - | - | - | 0.09 | 0.16 | 0.18 | 0.08 | 0.01 |
| | 15 | 0.00 | 0.00 | 0.00 | 0.01 | 0.01 | 0.01 | 0.00 | 0.00 | 0.01 | 0.03 | 0.04 | 0.04 | 0.02 |
| | 27 | 0.00 | 0.00 | 0.00 | 0.00 | 0.00 | 0.01 | 0.01 | 0.03 | 0.06 | 0.04 | 0.03 | 0.02 | 0.00 |
| | 39 | 0.02 | 0.01 | 0.01 | 0.03 | 0.06 | 0.10 | 0.10 | 0.09 | 0.01 | 0.02 | 0.02 | 0.02 | 0.00 |

[Figure]

**Figure 4:** Soil moisture values measured in the pore water during the first and the second run for the different zones, phases (saturation and drying) and depths. Water level is displayed in the second y-axis. The missing data from the sampling depths at 15, 27 and 39 cm in the non-vegetated zone is due to failures in the sensors. Red circles indicate the values previous to the injection.

[Figure]

**Figure 6:** Lag analysis performed to the Br- breakthrough curves for the first and second run between the sampling depths: 1) 39cm and 3cm; 2) 39cm and 15cm; and 3) 39cm and 27cm. Only significant lag correlations are displayed.

**Answers to referee #1**
* * *
We wish to acknowledge the constructive and thoughtful comments of the reviewer. The following explains point by point how we will address the reviewer comments (in italics). We appreciate the efforts of the reviewer and the valuable suggestions that we will consider when revising our manuscript. Some long comments have been subdivided into several comments.

**Scientific significance**

**Comment 1:** *The manuscript aims at improving the understanding of the fate of pesticides in constructed wetlands, which are implemented to mitigate pesticide pollution of surface water bodies. To that end, the authors describe in quite some detail findings from a complex laboratory experiment simulating the fate of different (organic) chemicals and Br- as a conservative tracer (except for plant uptake) in a constructed wetland. To improve with regard to previous studies, the authors have put a lot of effort in obtaining spatial and temporal resolution of the concentrations of their model compounds in the experimental wetland.*
*Despite the fact that constructed wetlands have some practical relevance as mitigation measures, the scientific relevance of the manuscript seems to be limited. On p. 2, L. 5 – 9, the authors describe their objectives. However, in the current form they are very specific to the experimental design and it remains unclear (also subsequently in the manuscript, see also comments below) how answers to the posed question can be generalised:*

**Response 1:** We thank the referee for pointing this out. Indeed, we have not clearly stated in the manuscript how the specific findings of this experiment can be generalised. Primarily, we wanted to highlight the usefulness of the experimental method, namely that fluorescent tracers (which are organic molecules, non-toxic and easy to be analysed) can be used to highlight the fate of pesticides inside wetland systems (mostly considered as black boxes so far). We apologize for the lack of clarity in this regard. While we think that a generalization of the results of our study to real-world wetlands cannot be made without validation in the field by additional experiments, it is true that the generality could be improved in the manuscript. In the revised version we will be clearer about this question in order not to limit the scientific relevance of the study (see also the responses to the comments below).

Notes on response 1: This comment has been addressed in the responses below (2, 3, 4, 5 and 6).

**Comment 2:** *How to gain general insights if one knows in detail the spatial and temporal patterns of pesticide fate processes in this particular wetland at the lab scale (refers to objective i))?*

**Response 2:** The first objective of the study (objective i)) was to find out whether the use of a multi-tracer approach together with high vertical-resolution sampling and monitoring would allow to identify spatial and temporal patterns of pesticide fate processes. Our experiment aimed at providing a new methodology to better understand the behavior of pesticides in constructed wetlands. The level of detail of the data obtained made it possible to link more accurately the response of the target compounds with the different variables. If we know these relationships, we can extrapolate the results of our particular lab-scale experiment to real-world systems, provided that the same conditions take place. Furthermore, we found important state variables that should be monitored in field experiments.
A better explanation about how to generalise our particular results will be provided in the revised version. This will include a comparison of our system with real-world wetlands.

Notes on response 2: We have addressed this point on p. 9 L. 12-20 as follows:

"Although our experiment has been carried out in a laboratory environment, the replicated conditions may resemble those of a groundwater-fed wetland that undergo wet-dry cycles and that intercepts pesticide-contaminated water during groundwater discharge. Similar systems have already been investigated with the same multi-tracer approach under laboratory (Durst et al., 2013) and field conditions (Maillard et al., 2016) and the results were analogous to our findings. For instance, Durst et al. (2013) found that preferential flow along the roots took place in the vegetated part of the wetland resulting in greater solutes recoveries, whereas Maillard et al. (2016) demonstrated that the alternation of oxic-anoxic conditions enhanced the dissipation of solutes. Other field studies in wetland systems have pointed out that the presence of vegetation greatly increases contact time and surface area for adsorption (Moore et al., 2006; Liu et al., 2018), which may also be enhanced when organic matter content is high (Passeport et al., 2011)."

**Comment 3:** *How to generalise the findings related to the different behavior of the model compounds (refers to objective ii))? Tracers-versus pesticides*

**Response 3:** The second objective of the study (objective ii)) was to compare the temporal and spatial behavior of the selected pesticides with reference tracers. In this case, a generalization could be made by comparing our results with those of other similar studies where the same or comparable tracers and pesticides have been used in wetland/buffer systems. One example is the study of Maillard et al., 2016. This information will be included in the revised version.

Notes on response 3: This point has been discussed in the new subsection "3.5 Potential of hydrological tracers to evaluate transport and dissipation processes of pesticides in constructed wetlands" on p. 12 L. 18-39 as follows:

"In view of the results obtained in the present study, some conclusions could be drawn regarding the use of Br-, SRB and UR to evaluate transport and dissipation processes of pesticides in constructed wetlands. In particular, we have corroborated that Br- can be used to elucidate non-reactive transport of solutes in constructed wetlands, as already reported in the literature (Lin et al., 2003; Małoszewski et al., 2006). But it can also be applied to identify plant uptake, although to a lesser extent. As for SRB, despite the fact that it has been extensively used to identify sorption processes in wetland systems (Passeport et al., 2010; Lange et al., 2011; Schuetz et al., 2012), its special sorptive character makes it difficult to be compared with a certain type of pesticide. In this regard, while Dollinger et al. (2017) stated that SRB could be used as a good proxy for hydrophilic and strongly sorbing pesticides, Lange et al., 2018 demonstrated that the same tracer closely mimicked the gradual recession of a moderately hydrophobic pesticide in the top soil of an agricultural field. As for our results, we found that SRB could describe well the behavior of the pesticides boscalid and penconazole (moderately and highly hydrophobic, respectively) in terms of retention and retardation in the pore water and in the water at the outlet when the constructed wetland is repeatedly flushed. However, it may not be suitable to evaluate overall recoveries of boscalid and penconazole at the outlet given that greater amounts of SRB may be recovered compared to such pesticides, possibly due to its greater leachability and/or lower susceptibility to be taken up by the plants. Regarding UR, in terms of transport our results suggested that it may illustrate well the behavior of mobile and non-persistent pesticides, such as metazachlor, which is in agreement with the findings of other studies (Durst et al., 2013; Maillard et al., 2016; Torrentó et al., 2018). At the same time, our results have underlined that UR may experience not only photodegradation, but also (bio-)chemical transformation, which is consistent with the results of recent investigations (Maillard et al., 2016; Lange et al., 2018, Fernández-Pascual et al., 2018). Yet, UR biodegradation might be limited in the presence of preferred substrates for microorganisms. In any case, it should be noted that the conclusions presented here are only valid if these tracers are used in studies under similar conditions as those of our experiment. That is, constructed wetlands that undergo long periods of stagnation (> 2 months), with drying periods in between, sorbing material with low organic carbon content, similar vegetation and subject to analogous dominant processes."

**Comment 4:** *How to generalise the results regarding vegetation and hydrologic conditions (refers to objective iii))?*

**Response 4:** The third objective of the study (objective iii)) was to assess the influence of vegetation and the alternation of different hydrologic conditions on pesticide transport and dissipation processes. The results of our study regarding vegetation and hydrologic conditions can be generalised by establishing parallels between the conditions simulated in the laboratory and those that occur in real wetlands. In particular we will discuss effects of temporary flooding and different kinds of groundwater surface water interactions. These questions will be addressed in the revised version.

Notes on response 4: This point has been addressed on p. 8, L. 40-41 and p. 9, L. 1-11 as follows:

"The results of our study underlined the importance of plants in promoting dissipation processes in constructed wetlands. Indeed, plants have already been attributed the ability to facilitate elimination, degradation and retention of pesticides in wetland systems (Liu et al., 2018). However, our findings also suggested that plant roots may be involved in the formation of preferential flow paths, which could result in a rapid transport of contaminants and decrease in the interactions between solutes and sediments (Durst et al., 2013). In fact, plant roots have been related to the creation of discontinuities in the soil profile, greater presence of macropores and occurrence of bypass flow (Ghestem et al., 2011). Therefore, the beneficial impact of plants in terms of elimination, degradation and retention may be reduced by the occurrence of preferential flows.
Our results have also indicated that the promotion of aeration has facilitated the degradation of some substances. This was in agreement with recent studies that have demonstrated that intermittent flow regimes support aerobic microbial populations and boost degradation rates of pesticides (e.g. Karpuzcu et al., 2013; Maillard et al., 2016). Other authors also found that by alternating drainage with no drainage periods in constructed wetlands, these systems are capable of reducing non-point pollution (Vallée et al., 2015). Hence, it could be generalized that the mitigation capacities of constructed wetlands might be improved if aerated conditions in the system are fostered."

**Comment 5:** *My statement does not imply that no such general insight could be gained from the experiment. However, in order to do so, one would need to ask first general scientific questions and subsequently demonstrate how the experiment can provide such generalizable answers. Such questions however are missing. The sentence on p. 2, L. 3 – 4, is too vague in this respect. This limitation is subsequently reflected in the Conclusion section. There is a lack of novelty and the statements are either very general or too speculative.*

**Response 5:** We apologize for the overall lack of clarity and agree that our general scientific questions should be better defined in order not to limit the conclusions, and we are grateful to the reviewer for pointing this out. As stated above, we will address this point in the revised version. We will follow two main lines: (i) we will compare existing (black-box) field results with our findings and (ii) we will further emphasize which conditions in natural wetland systems were actually mimicked in our

experiments. This way, our experiment will provide original and relevant data that can help improve the understanding of complex phenomena related to transport and dissipation of pesticides observed in real-world systems.

Notes on response 5: In order to better reflect the objectives of our study and how it can provide generalizable answers, we have modified the introduction on p. 2, L. 10-27 as follows:

"The mitigation capacities of buffer zones have recently been studied by using hydrological tracers as a low-cost approach. In this context, fluorescent tracers (e.g. uranine (UR), sulforhodamine B (SRB)) have often been chosen to study transport and fate of pesticides because they are organic molecules, non-toxic and easy to be analysed. For instance, some authors have used them in wetlands (Passeport et al., 2010; Lange et al., 2011; Durst et al., 2013; Maillard et al., 2016) and farm ditches (Dollinger et al., 2017). Yet, in most cases where this approach has been applied the system under study has been treated as a "black box" where the time scales were typically limited to the time spans of the tracers breakthroughs at the systems outlet. Hence, internal temporal and spatial mechanisms that dominate pesticides transport and dissipation (e.g. sorption, transformation, plant uptake) are still not fully clear. Moreover, information on the fate and, particularly, transformation of pesticides inside wetland sediments is still limited.
Therefore, the objectives of this study are i) to apply a multi-tracer approach together with high vertical-resolution sampling and monitoring to identify transport patterns and dissipation processes of three pesticides selected as test substances inside a model constructed wetland system; ii) to compare the temporal and spatial behavior of the applied tracers with the pesticides and evaluate their main dissipation pathways; and iii) to assess the influence of vegetation and alternating different hydrologic conditions (saturated and unsaturated) on transport and dissipation processes.
Our study is one of the first to look at the solutes behavior inside a constructed wetland on a long-term basis and detailed spatial scale. With this experiment we expect to provide new insights about the potential of hydrological tracers to evaluate transport and dissipation processes of pesticides. Likewise, we seek to extend the knowledge on the mitigation capacities of constructed wetlands with our approach."

**Comment 6:** *One way how the generality could be improved would for example be to put the characteristics of the study wetland (texture, organic carbon content, water residence time, redox conditions etc.) into the context of real-world wetlands, to reflect – based on scientific theory – what follows for pesticide retention in such wetland and to demonstrate respective insights that go beyond prior knowledge. I missed such information in the manuscript.*

**Response 6:** We appreciate this comment and we agree that we have to improve the explanation about the insights we have gained from of our study. As stated in Responses 2, 3 and 4, the characteristics of our lab-scale constructed wetland will be better addressed and put into the context of real-world wetlands.

Notes on response 6: This point has been addressed on p. 10, L. 24-36 as follows:

"In principle, we expect to obtain analogous results in wetland systems if similar conditions are met. In this regard, if we compare the characteristics of our experiment

(see Table 2) with those of other wetland studies (e.g. Catallo, 1999; Seybold et al., 2002; Maillard et al., 2011; Gardiner et al., 2012; Passeport et al., 2013; Vallée et al., 2016; Gikas et al., 2018) we find similar values in terms of sediment texture (values ranging from 4 to 89.5/6.2 to 55/3.8 to 44 for % Sand/Silt/Clay, respectively), sediment pH (values ranging from 6 to 8), conductivity (values ranging from 0.45 to 0.9 dS/m) and redox potential (values ranging from -500 to +500 mV). However, there are some discrepancies regarding organic carbon content (values ranging from 2.6 to 32.7 %) and mean residence time (values ranging from 0.5 to 8 days). In this case, the values of our experiment were either below (for the organic carbon content) or above average (mean residence time). Yet, the overall removal rates obtained in our experiment (see Table 6) were within the same range of those of the wetland studies. For instance, Vallée et al. (2015) found that the removal rates of boscalid in two pilot-scale wetlands ranged from 38 to 67%, whereas Gikas et al. (2018) obtained removal rates for S-metolachlor (pesticide from the same group as metazachlor) that reached up to 92.6% in a constructed wetland planted with Phragmites australis. Other authors have reported removal rates of 45%–90% for tebuconazole (a triazole fungicide similar to penconazole) in wetland systems (Passeport et al., 2013; Tournebize et al., 2013)."

**Scientific quality**

**Comment 7:** *Overall, the manuscript indicates that the experiments were carefully planned and executed. There are few technical questions that are listed below.*
*However, there are conceptual limitations that also relate to the comments on the scientific significance above. A major issue is the lack of replication. There is only one vegetated and one non-vegetated chamber of the experimental tank. I am aware of the effort needed to carry out such experiments and to build such experimental facilities. Nevertheless, the results and conclusions hinge solely on single realisations of two experimental treatments. Especially in the context of preferential flow phenomena, this may be very critical because a single connected flow paths may exert a strong effect on the overall outcome. Without replication, it is very difficult to judge the robustness of the differences observed between the two treatments*

**Response 7:** We thank the reviewer for pointing this out. While it is true that we only had one experimental unit with one vegetated and one non-vegetated zone, the results and conclusions did not depend solely on one single experimental run. In fact, we performed two experimental runs. We think that two identical runs of a dynamic system (the vegetation with its root system was constantly developing and hence also modified preferential flowpaths) may be treated as a replication. To build replicates of such a complex experiment was beyond our financial possibilities.

**Comment 8:** *Another limitation is the lack of quantitative analyses that could link the different pieces of information.*
*(1) The authors report for example Koc-values for the different compounds from the literature but do not provide quantitative analyses how transport and concentrations levels were expected based on this information.*
*(2) I also missed key features such as expected hydraulic residence time in the system etc. One could probably calculate such things from the information in the text and Tab. 2, but it would be useful for readers to directly get such information.*

**Response 8-(1):** Thanks for raising this important point. The information regarding Koc-values has been given in the text primarily as a guiding reference to interpret the behavior of the different solutes in terms of persistence and mobility. The use of parameters such as Koc-values to do predictions on transport and concentrations levels may be possible by applying modeling approaches. We are aware of the importance of modelling and we plan to carry out modelling in the future. However, we believe that this would go beyond the scope of the present study and would enlarge the manuscript too much.

**Response 8-(2):** Our system has not worked like a conventional constructed wetland. That is, the solutes were injected in the system and principally remained there throughout the experiment. We only sampled very small fractions of pore water. That is why, the hydraulic residence time would largely be equivalent to the duration of the experiment. We understand that this has to be made clear in the revised version and we will take care of this point.

Notes on response 8-(2): The value of the equivalent hydraulic retention time has been specified in Table 2B (see below).

**B**

| Parameter | Unit | Value |
|---|---|---|
| Inlet/outlet pumping rate | L h$^{-1}$ | 21.6 |
| Peristaltic pumping rate | L h$^{-1}$ | 0.1 |
| Volume of tracers and pesticides injected | L | 40 |
| Volume of clean water injected at the end of the drying phase | L | $34.1 \pm 3.1$ |
| Volume of total clean water injected in the flushings | L | $355.1 \pm 20.5$ |
| Hydraulic retention time | Days | $62.5 \pm 2.12$ |

**Presentation quality**

**Comment:** In general, the paper is clearly written, and the findings are carefully presented in the figures and tables.

**Detailed comments**

**>> Title:**

**Comment 9:** *Use of tracers: Why do you distinguish between tracers and pesticides? Uranine and sulforhodamine B are organic chemicals as are the three pesticides used in the study. Of course, there is a difference in the use of the compounds, but why is this distinction relevant for elucidating the fate of the pesticides (given the fact that also these tracers undergo sorption plant uptake and degradation)?*

**Response 9:** We appreciate your comment. It is true that both the tracers (Uranine (UR) and sulforhodamine B (SRB)) and the pesticides are organic chemicals. We have made a distinction between them because the hydrological tracers are the instrument that we expect to be a reference to study pesticide transport and dissipation processes. The present study seeks to confirm the feasibility of these tracers to investigate processes

that dominate the behavior of pesticides in constructed wetlands. To do that we need to make comparisons between them, and therefore a distinction was made.

**>> Abstract:**

**Comment 10:** *p. 1, L. 10: What are spatial and temporal mechanisms?*

**Response 10:** Here we refer to those "processes" that may dominate pesticide transport and dissipation in constructed wetlands over time and space (e.g. sorption, transformation, plant uptake). We will make this clear in the revised version.

Notes on response 10: The term "spatial and temporal mechanisms" has been clarified throughout the text as follows:

"…internal temporal and spatial mechanisms that dominate pesticides transport and dissipation (e.g. sorption, transformation, plant uptake)…"

**Comment 11:** *p. 1, L. 13: What was the rationale behind the selection of these compounds?*

**Response 11:** We thank the referee for pointing this out. Boscalid, penconazole and metazachlor were selected because these pesticides were the most frequently detected in a field-based constructed wetland where other studies within the same project were carried out. We apologize for the omission of this information, which will be duly included in the revised version.

Notes on response 11: The information regarding the selection of pesticides (see below) has been included on p. 2, L. 33-36:

"Three pesticides were selected as test substances according to their different physicochemical properties and frequent detection in a field-based constructed wetland where other studies within the same project were carried out: boscalid (2-chloro-N-(4'-chlorobiphenyl-2-yl) nicotinamide), penconazole ((RS)-1-[2-(2,4-dichlorophenyl) pentyl]-1H-1,2,4-triazole) and metazachlor (2-chloro-N-(pyrazol-1-ylmethyl) acet-2',6'-xylidide)."

**Comment 12:** *p. 1, L. 16 – 17: What do you mean by the statement that transport dominated for some compounds?*

**Response 12:** Obviously, we did not make this point clear enough. Here, we mean that transport was more significant for Br, UR and metazachlor compared to SRB, boscalid and penconazole. That is, according to the results Br, UR and metazachlor experienced more transport than the other solutes during the experiment. This will be clarified in the revised version.

Notes on response 12: The statement "transport dominated for some compounds" has been changed by "The strong temporal and spatial correlation found between Br-, UR and metazachlor indicated that these solutes experienced more transport than SRB, boscalid and penconazole" on p. 1, L. 21-22.

**Comment 13:** *p. 1, L. 17 –18: What other dissipation processes could be expected? This statement is not very informative.*

**Response 13:** This is a very important remark. We agree that the statement may not be clear enough. Our intention was to show that the mass balance has allowed us to identify the processes of sorption, transformation and plant uptake. So, we still believe that this statement should be kept in the text, but it will be better explained.

Notes on response 13: The statement has been changed on p. 1, L. 24-25 and p. 13, L. 4 as follows:

"The overall tracer mass balance allowed us to identify three dissipation pathways: sorption, transformation and plant uptake."

**>> Introduction:**

**Comment 14:** *p. 1, L. 27: The reference is not very recent. Many others are available representing more current findings.*

**Response 14:** We agree, and the reference "Müller et al., 2002" will be replaced by more recent studies.

Notes on response 14: We have added the following references on p. 1, L. 36-37: von der Ohe et al., 2011 and Casado et al., 2019.

**Comment 15:** *p. 1, L. 28 – 29: Generally, transformation products are less toxic. There are exceptions but the wording may be misleading.*

**Response 15:** We thank the referee for this comment. The sentence "transformation products (TPs), whose toxicity or persistence is unknown." will be changed to "transformation products (TPs), which in some cases, could be more persistent and toxic than the parent compound"

Notes on response 15: The corresponding change has been made on p. 1, L. 37-39 as follows:

"…transformation products (TPs), whose behavior is unknown, and toxicity or persistence may be in some cases greater than the parent compounds."

**Comment 16:** *p. 1, L. 40: This is an important aspect. Unfortunately, this manuscript does not really elaborate any further on this topic. It would be interesting to learn how the results reported here relate to other studies and what the results imply for mitigation capacities.*

**Response 16:** Thanks for raising this important point. It is true that the possible implications of our results for the study of the mitigation capacities of constructed wetlands have not been discussed thoroughly enough. In this sense, we believe that our findings are relevant and make an important contribution for the evaluation of the mitigation capacities of buffer zones. Therefore, the revised version will provide a more

in-depth discussion on this topic. This also refers to the general comments above: we will provide comparisons to existing wetland field studies that have used the same or similar components.

Notes on response 16: The question regarding mitigation capacities of constructed wetlands has been addressed throughout the manuscript. For instance:

On p. 10, L. 41-43 and p. 11, L. 1-2: "These observations highlight the importance of certain factors in the elimination of pesticides, namely the presence of adequate vegetation, suitable matrix materials, long residence times, low flow rates, intermittent flow conditions, among others (Vymazal et al., 2015; Liu et al., 2018). When these factors are promoted, the elimination rates tend to increase, and therefore, the mitigation capacities of constructed wetlands."

On p. 13, L. 23-26: Our findings pointed out that the presence of plants and the alternation of different hydrological conditions (saturation and drying periods) may favor dissipation processes. The combination of these factors together with others (e.g. suitable matrix materials, long residence times, etc.) could increase the mitigation capacities of wetland systems. Yet, plants might also be involved in the creation of preferential flow paths with the consequent risk of rapid transport of contaminants."

**Comment 17:** *p. 2, L. 13 – 18: The critical question about the compound selection is what insight can be gained. In the result section (p. 9, L. 1 – 20), the results about the compound-specific differences are summarised. The reported findings basically reflect the knowledge already used for making the compound selection. Hence, the authors miss to derive more general insight that goes beyond the prior knowledge.*

**Response 17:** You raise a very valid point about the fact that we have not added enough information about the insight that we have gained from the selection of the hydrological tracers Br, UR and SRB. Our study is relevant because it has corroborated previous knowledge about these hydrological tracers with an experiment that had not been done before. We agree that more general statements about the use of these tracers for studying transport and dissipation processes of other pesticides can be made. To do this, a more exhaustive review of the bibliography on this topic will be included in the revised version.

Notes on response 17: As already answered in comment 3, a new subsection (3.5) has been created in order to discuss the insight that we have gained from the selection of the hydrological tracers Br⁻, UR and SRB.

**>> Methods:**

**Comment 18:** *p. 2, L. 28: How reliable is terbutryn as an internal standard for the other pesticides?*

**Response 18:** In fact, as stated in the manuscript, we used Terbutryn-D5 as an internal standard for the measurement of environmental water samples due to the possible occurrence of Terbutryn. Measurements of a variety of samples (about 1000 samples) determined that this internal standard was reliable for the detection of the substances in the water. Reliability was proved by the determination of recovery rates of

substances. Here, a certain concentration was spiked into the environmental water samples where matrix effects could suppress the signal of the substance. Recoveries were found to be about 100 % by the correction of the internal standard.

**Comment 19:** *p. 4, L. 8: Generally, glass bottles are used for storing pesticide samples.*

**Response 19:** We used polypropylene tubes instead of glass bottles to store the pesticides because the samples had to be frozen immediately after their collection in order to preserve them before their shipment to the laboratory. This type of material has already been used to store pesticides in other studies (e.g. Joseph, 2015).

Notes on response 19: This information has been clarified on p. 4, L. 27-30 as follows:

"Previously, a volume of 10 mL was transferred to 15 mL Polypropylene tubes and stored at -20°C for the subsequent pesticide and TPs analysis. Polypropylene was chosen instead of glass because the samples had to be frozen immediately after their collection. Such material has already been used to store pesticides in other studies (e.g. Joseph, 2015)"

**Comment 20:** *p. 4, L. 30: What about possible inferences with the fluorescence of the background matrix?*

**Response 20:** We always analysed the entire fluorescent spectrum from 350 to 600 nm. This way, we could identify different background fluorescent levels and were able to subtract them. We will state this detail in the revised version.

Notes on response 20: The following information has been added on p. 5, L. 12-13:

"The entire fluorescent spectrum (from 350 to 600 nm) was analyzed in order to identify different background fluorescent levels and subtract them."

**Comment 21:** *p. 5, L. 4: Where are LOQ/LOD provided?*

**Response 21:** Thanks for pointing this out. LOQ/LOD values (see below) for the pesticides and transformation products will be provided in Section 2.5.2 "Pesticides and TPs in the pore- and outlet- water" of the manuscript.

| Substance | LOD [ng L$^{-1}$] | LOQ [ng L$^{-1}$] |
|---|---|---|
| Boscalid | 0.35 | 1.27 |
| Penconazole | 0.35 | 1.29 |
| Metazachlor | 0.35 | 1.27 |
| Metazachlor-ESA | 2.78 | 10.35 |
| Metazachlor-OA | 0.54 | 1.90 |

Notes on response 21: The corresponding information has been included in Table 3

**Comment 22:** *p. 5, L. 12: How can an independent background be determined?*

**Response 22:** We extracted the background signal according to the method described by Leibundgut et al. (2009). Such method does not use an independent background.

Instead, it uses an equation that is based on the geometry of the curve from which the background is to be removed.

**Comment 23:** *p. 5, L. 35 – 36: This sentence sounds strange because transport processes affect all compounds irrespective whether or not they are sorbed or not (or degraded or not).*

**Response 23:** We agree with your statement. However, what we claim here is that if other processes such as sorption or transformation dominate, they will have an influence on the behavior of solutes in terms of transport (e.g. retardation by sorption, attenuation by degradation). Therefore, we have assumed that in those cases a strong correlation with Br (considered as the most conservative tracer) will not be observed. We will reformulate the sentence to make it more clear.

Notes on response 23: The sentence has been reformulated on p. 6, L. 11-15 as follows:

"The predominance of transport processes among the solutes was examined by looking at the relationship between Br- and the other solutes via correlation matrices of the measured concentration time series. Here, it was assumed that a weak correlation would be due to the prevalence of sorption and transformation rather than transport. This was based on the premise that the solutes would experience retardation due to sorption or attenuation due to transformation."

**>> Results and discussion:**

**Comment 24:** *p. 6, L.10: What means an early breakthrough? Early compared to what?*

**Response 24:** We thank the referee for this important comment. "Early breakthrough peaks" means that they were detected in the first place. This is a common expression that has been used in other studies (e.g. Torrentó et al., 2018). We will clarify this in the revised version.

Notes on response 24: Results and discussion section has been rewritten to present the results in a more understandable way. Here, it has been decided to eliminate the term "early breakthrough" because it can be misleading.

**Comment 25:** *p. 6, L.10– 30: These paragraphs list different findings without a clear structure and logic.*

**Response 25:** We apologize for the lack of clarity. Our intention in this subsection was to explain the arrival of the breakthrough peaks of the solutes (including the TPs) to the different zones and depths in chronological order. The structure of these paragraphs will be improved in the revised version so that the ideas are presented in a more clear way.

Notes on response 25: As indicated in notes on response 24, the results section has been rewritten and restructured in order to present the results in a clearer and more logical way. This has included the creation of a new subsection (3.1 Transport processes in the pore water according to Br- behavior).

**Comment 26:** *p. 6,L. 37: Where can one see these redox conditions?*

**Response 26:** The graphs of the redox conditions can be found in Fig. 4 (black line, second y-axis). We will better indicate this information in the revised version.

Notes on response 26: This information has been clarified on p. 7, L. 37-39 as follows:

"Here in the middle layers, dominant processes most likely differed between zones, as was also evidenced by the different redox potentials measured in the vegetated and the non-vegetated zone (See Fig. 5, second y-axis)."

**Comment 27:** *p. 7, L. 6: I assume that sorption takes place all the time and not only during the initial phase.*

**Response 27:** That is correct. What we have stated here is, that sorption velocity was most likely higher at the beginning of the experiment compared to later phases when it probably decreased, given that the number of free sorption places became smaller. This will be clarified in more detail in the revised version.

Notes on response 27: This information has been detailed on p. 8, L. 37-39 as follows:

"Given the rapid decrease in relative concentrations shortly after the injections, it was assumed that sorption velocity was higher at the beginning of the experiment compared to later phases, when it probably decreased due to a decline in the number of free sorption places."

**Comment 28:** *p. 7, L. 18: Where can one see this correlation?*

**Response 28:** The correlation between the breakthrough curves of Br is shown in Table 3. We will clarify this.

**Comment 29:** *p. 7, L. 20 – 30: These sentences are confusing.*

**Response 29:** We apologize for the confusion. These sentences provide evidence (through correlations) that the performance of the experiment in the vegetated zone during the first run was different compared to the second run. This information supports the idea that the plants possibly played an important role in our experiment and possibly modified flowpaths, etc. Therefore, it has been included in this subsection. The sentences will be rewritten to better express the main message in the revised version. This also has a bearing on why we regarded the second execution as a kind of replication (see general comments above).

Notes on response 29: As indicated in comment 25, the results have been rewritten to better illustrate the findings. Transport processes have been now discussed according to Br- behavior in subsection 3.1. Here, on p. 7, L. 26-33 we have inserted the information related to the role of plants in the possible modification of transport processes as follows:

"The different behavior of Br- in terms of transport observed in the first and second run was explained by a possible development of the root system. In fact, when correlating

the breakthrough curves of Br- of the first and the second run, differences between the vegetated and non-vegetated zones were observed (Table 5). With the exception of the uppermost layer, the non-vegetated zone showed strong correlation between the two runs regardless of the layer, whereas the vegetated did not show any correlation. This meant that the performance of the non-vegetated zone was similar in both runs, whereas the vegetated behaved differently. Hence, changes in root density and/or spatial distribution most likely occurred during the experiment. As a result, presumably both, transport processes and dissipation varied over time (Goss et al., 1993). This assumption was supported by visual observations of the sediment (Fig. 7)."

**Comment 30:** *p. 7, L. 30 – 33: This paragraph is not well linked into the structure.*

**Response 30:** We are grateful for this observation. The information provided in this paragraph is relevant because it justifies the role of the hydrologic conditions in transport and dissipation of pesticides. This paragraph will be better explained and integrated into the text in the revised version.

Notes on response 30: See notes on responses 25 and 29

**Comment 31:** *p. 8, L. 11 – 13: Please be aware that different transformation products may have different source terms because they are generally formed at different rates and possibly in different parts of the subsurface.*

**Response 31:** We thank the reviewer for this comment. We totally agree, and the sentences regarding the possible transport of metazachlor TPs based on their comparison with Br will be removed from the text, as we cannot accurately determine where and when they were formed.

**Comment 32:** *p. 8, L. 22 – 24: This is very qualitative. What were the expected compound-specific differences solely based on the Koc-values?*

**Response 32:** We thank the referee for raising this important issue. We have stated in the manuscript that, according to our results, SRB, boscalid and penconazole experienced more sorption than the other compounds (Br, UR ad metazachlor), which may be explained by their sorption properties. While it is true that the Koc-values may help interpreting these results from a qualitative point of view, the amount of compounds adsorbed and/or the type of interaction behind the adsorption cannot be explained only with Koc-values. Hence, a more detailed discussion based on substrate properties and additional parameters (e.g. Kd-values, aqueous solubility) will be done in the revised version.

Notes on response 32: This question has been discussed in the new subsection 3.5 on p. 12, L. 21-31 as follows:

"As for SRB, despite the fact that it has been extensively used to identify sorption processes of pesticides in wetland systems (Passeport et al., 2010; Lange et al., 2011; Schuetz et al., 2012), its special sorptive character makes it difficult to be compared with a certain type of pesticide. In this regard, while Dollinger et al. (2017) stated that SRB could be used as a good proxy for hydrophilic and strongly sorbing pesticides, Lange et al., 2018 demonstrated that the same tracer closely mimicked the gradual

recession of a moderately hydrophobic pesticide in the top soil of an agricultural field. As for our results, we found that SRB could describe well the behavior of the pesticides boscalid and penconazole (moderately and highly hydrophobic, respectively) in terms of retention and retardation in the pore water and in the water at the outlet of constructed wetlands when the system is repeatedly flushed. However, it may not be suitable to evaluate overall recoveries of boscalid and penconazole at the outlet given that greater amounts of SRB may be recovered compared to such pesticides, possibly due to its greater leachability and/or lower susceptibility to be taken up by the plants."

**Comment 33:** *p. 8, L. 25 – 26: Again, this statement appears rather isolated in the text.*

**Response 33:** We are sorry for the lack of clarity in this regard. Given that we consider that the observations on the recovery of metazachlor TPs at the outlet are an important finding of our study, they will be better integrated into the text to facilitate the reader's understanding.

Notes on response 33: The statement has been integrated into the text on p. 9, L. 39-41 as follows:

"Cumulative recovery curves of metazachlor TPs were also obtained at the outlet of the system during the flashings (Fig. 9), thereby evidencing their great mobility and persistence in the environment (Mamy et al., 2005; European Food Safety Authority (EFSA), 2008). In this case, higher amounts of met-ESA were recovered compared to met-OA."

**Comment 34:** *p. 9, L. 31: Here you contradict yourself: above you have argued that SRB is expected to be strongly sorbed because of its Koc-value (p. 8, L. 23)*

**Response 34:** We thank the referee for raising this important issue. The peculiarity of SRB is that it has both charged groups (cationic and anionic) and a non-polar region (Polat et al., 2011). This will make SRB susceptible to sorption on positive and negative charged mineral sites, OH-groups of hydroxides and clay minerals, but also on nonpolar sorption sites of organic matter. The latter would explain why we found large amounts of this tracer in the part of the sediment where the largest portion of organic carbon was observed. Considering the above, the use of Koc-values would probably not be appropriate to interpret the results of SRB as it may lead to misunderstandings. This will be taken into account and corrected in the revised version.

Notes on response 34: We have rewritten the statements on p. 12, L. 11-15 as follows:

"On the other hand, most of SRB was found sorbed in the sediment of the vegetated zone, where the highest concentration of organic carbon was located (Fig. 11). This may be explained by the susceptibility of SRB to sorption on nonpolar sorption sites of organic matter (Polat et al., 2011). Moreover, it has been recently demonstrated that SRB has high sorption affinity for litters in wetlands (Dollinger et al., 2017). Thus, probably the presence of dead leaves and decaying plant residues in the uppermost layer enhanced sorption of SRB."

>> **Figures:**

**Comment 35:** *Fig. 4: - It is difficult to distinguish all the different lines. - What were the hypotheses, how the breakthrough would differ between the different depths and the different compounds?*

**Response 35:** We are grateful for this comment. In order to facilitate a better distinction of the curves, Figure 4 will be split in two graphs, one for the vegetated and one for the non-vegetated zone. What we wanted to show in this figure was, on the one hand, the evolution of the temporal and spatial concentration of the solutes in the pore water, and on the other, how the pesticides behave compared to the tracers.

Notes on response 35: The new figure (Fig. 5) is shown below

[Figure]

**Figure 5:** Breakthrough curves of the different tracers, pesticides and their TPs in terms of relative concentrations ($C/C_0$) (obtained by scaling with the input concentrations) measured in the pore water during the first and the second run for the different zones, phases (saturation and drying) and depths: 1) 3cm; 2) 15cm; 3) 27cm and 4) 39cm. Changes in redox potential are displayed in the second y-axis (Eh in mV). The different injections performed during each run are displayed on top of the figure. Note that the scale of the relative concentrations corresponding to the sampling depth of 3 cm is extended.

**Comment 36:** *Fig. 7: Is there no differentiation between vegetated and non-vegetated treatments?*

**Response 36:** No, because the objective of this figure is to show how much solute in general is recovered at the outlet of the system after each flushing.

**Comment 37:** *Fig. 8: You might consider comparing the two treatment with separate bars.*

**Response 37:** We thank the referee for this suggestion. However, it is not possible to make a distinction between the treatments with two bars. Both zones (vegetated and non-vegetated) are part of the same unit and the percentages of recovery from each zone have been calculated with respect to the total amount of solutes injected. Therefore, the final percentage recovered is the sum of the percentages from the vegetated and the non-vegetated zone.

**Comment 38:** *Fig. 9: Is the sorption consistent with Koc-values known for SRB?*

**Response 38:** As stated in the responses to comments 32 and 34, the Koc-value for SRB itself would not explain the results obtained for this tracer. In this case, we have to look into its molecular structure and sorption properties in more detail to elucidate the performance of SRB in the sediment. Therefore, to avoid confusion, and as already mentioned in the previous comments, Koc-values will not be used in the revised version to interpret the behavior of SRB. We think that our findings provide some general insights into the ambivalent sorption behavior of the tracer SRB that has been reported in literature. We will discuss this in our revised version.

Notes on response 38: See notes on responses 32 and 34.

**Literature**

Joseph, G. (2015). Determination of sodium monofluoroacetate in dairy powders by liquid chromatography tandem mass spectrometry (LC-MS/MS): First Action 2015.02. Journal of AOAC International, 98(4), 1121-1126.

Leibundgut, C., Maloszewski, P., & Külls, C. (2009). Environmental tracers. Tracers in Hydrology, John Wiley&Sons, Ltd., Chichester, UK, 13-56.

Maillard, E., Lange, J., Schreiber, S., Dollinger, J., Herbstritt, B., Millet, M., & Imfeld, G. (2016). Dissipation of hydrological tracers and the herbicide S-metolachlor in batch and continuous-flow wetlands. Chemosphere, 144, 2489-2496.

Polat, B. E., Lin, S., Mendenhall, J. D., VanVeller, B., Langer, R., & Blankschtein, D. (2011). Experimental and molecular dynamics investigation into the amphiphilic nature of sulforhodamine B. The Journal of Physical Chemistry B, 115(6), 1394-1402.

Torrentó, C., Prasuhn, V., Spiess, E., Ponsin, V., Melsbach, A., Lihl, C., ... & Hunkeler, D. (2018). Adsorbing vs. nonadsorbing tracers for assessing pesticide transport in arable soils. Vadose Zone Journal, 17(1).

von der Ohe, P. C., Dulio, V., Slobodnik, J., De Deckere, E., Kühne, R., Ebert, R. U., ... & Brack, W. (2011). A new risk assessment approach for the prioritization of 500 classical and emerging organic microcontaminants as potential river basin specific pollutants under the European Water Framework Directive. Science of the Total Environment, 409(11), 2064-2077.

Casado, J., Brigden, K., Santillo, D., & Johnston, P. (2019). Screening of pesticides and veterinary drugs in small streams in the European Union by liquid chromatography high resolution mass spectrometry. Science of The Total Environment, 670, 1204-1225.

**Answers to referee #2**
* * *
**Comment 1:** *The authors present an interesting and well written paper on transport and dissipation processes of different substances in a constructed wetland at the lab scale. This is a highly relevant topic that is within the scope of HESS and of interest for a broader audience. The experiments and results are highly interesting and are largely presented clearly. A few points in the analysis and interpretation, however, should be revised to be less speculative and more supported by the results. This requires mainly further elaboration of the discussion, as detailed in the specific comments below.*

> **Response 1:** We appreciate the positive feedback, thoughtful comments and constructive suggestions from the reviewer that will help us improve the manuscript. We next detail the reviewer's comments (in italics) and our answers on how we will address the comments in the revised manuscript. Some long comments have been subdivided into several comments

**General comments**

**1. Recovery**

**Comment 2:** *A main concern is the low recovery of most of the substances. Except for bromide and SRB, less than a third of the applied masses were detected in the investigated compartments (the data on SRB suggests a recovery between 48 and 105% - Fig. 8). Because it is ambiguous to judge the parts that have not been observed, care needs to be taken in drawing conclusions on transport and dissipation from the data. The authors often did well in this regard and addressed possible pathways of the substances' fates by deduction and use of available literature. Sometimes they overachieved a bit, and some aspects deserve further clarification.*

> **Response 2:** We thank the reviewer for pointing this out. We are aware of the difficulty of drawing conclusions on transport and dissipation from the data when some observations in certain parts could not been made. That is why we have been very careful when interpreting the results. In any case, we agree with the comments of the reviewer and further clarification of the results in terms of transport and dissipation will be done in the revised version.

Notes on response 2: As already mentioned in the notes to the responses of referee 1, a new subsection has been created within the results (3.1) to address only transport processes. Regarding dissipation, a better clarification has been made based on the results obtained and similar studies found in the literature. This last point has been developed further and additional literature has been included in the revised manuscript.

**Comment 3:** *I would appreciate if the authors discussed possible reasons for low recovery in more detail.*

**Response 3:** Thank you for this suggestion. The low recoveries of the solutes is precisely a key point in our study and despite the limitations of the experiment, we have very detailed data that can help us better address this question. We agree with your assessment. As such, we will discuss it in greater depth in the revised version: in particular we will discuss recent scientific studies that have shown transformation of UR in contrast to SRB. Br as a salt can anyhow be treated to be chemically inert. Pesticides are known to be affected by biochemical degradation, however most knowledge stems from unsaturated soil and not from wetland sediments.

Notes on response 3: The discussion about the overall low recoveries of solutes at the outlet has been extended on p. 10, L. 16-23 as follows:

"As for the tracers, as expected Br- recovery was the highest, given its most conservative character. Following Br-, SRB showed the greatest recoveries, presumably because this tracer was mostly subject to sorption processes, as evidenced by its behavior in the pore water, and probably because it was more resistant to degradation, as already evidenced in a recent study (Fernández-Pascual et al., 2018). The lowest recovery among the tracers was for UR. In this case, it was assumed that both retention and especially degradation processes were involved in its dissipation. Overall, these results were in agreement with field studies performed in wetland systems where recoveries for UR were lower than for SRB and were explained by a higher incidence of degradation processes (i.e., photodegradation) on UR dissipation compared to SRB (Passeport., 2010; Lange et al., 2011; Schuetz et al., 2012)."

And on p. 10, L. 37-40 as follows:

"Overall, a possible explanation for the high elimination rates obtained in our experiment could be the fact of having promoted the contact of solutes with the medium through a long period of stagnation (i.e., about two months in each run). In this regard, a recent study performed by Gaullier et al. (2019) has reported almost total mitigation of pesticides and their TPs during stagnation (over 50 days) in constructed wetlands."

**Comment 4:** *What about the formation of other transformation products – is this likely, are other TP known that might be formed under the given circumstances?*

**Response 4:** We are grateful for this comment. Indeed, although met-ESA and met-OA are reported to be the major metabolites of metazachlor, it is possible that other transformation products formed in our system. However, such minor compounds were most likely below the limit of detection and therefore could not be identified. This information will be mentioned in the revised version.

Notes on response 4: This point has been clarified on p. 7, L. 40-41 and p. 8, L. 1-2 as follows:

"It should be noted that other transformation products may have been formed in our system. However, such compounds were most likely below the limit of quantification ($\leq 9.29$ and $\leq 10.28$ µg L-1 for p-Chlorobenzoic acid and 1,2,4-Triazole, respectively), and therefore, they could not be identified."

**Comment 5:** *If sorption is a major pathway, why have the substances not been detected in sediment/plants?*

**Response 5:** We are sorry for not making this point clear enough. The hydrological tracers were detected in the sediment/plants. Only the pesticides and their TPs could not be measured in this compartment because a quantitative method was lacking. We are aware of the importance of such information in unveiling the fate of the solutes. However, this again points to the advantage of using tracers instead of pesticides, as they are easier to be measured. The data provided in our study allowed us to build an overall view of the solutes behavior with great spatial and temporal detail. Moreover, we believe that our study represents a first approximation in this regard, and further experiments need to be done. We will enlarge the discussion in this point.

Notes on response 5: We have better explained this point on p. 6, L. 30-34 as follows:

"In the present study pesticides and their TPs could not be measured in the sediment and plants because a quantitative method was lacking. This issue pointed to the advantage of using tracers instead of pesticides because they are generally easier to be measure. Statements on the behavior of pesticides in the compartments where they could not be measured were made according to their physicochemical properties, the results of the breakthrough and recovery curves, their comparison with the tracers and the findings of similar studies."

**Comment 6:** *Which other pathways are possible, especially for the substances that are not likely to be adsorbed or degraded?*

**Response 6:** We thank the referee for this important comment. In our study, we have speculated that plant uptake could be an important dissipation pathway. Mineralisation of the compounds to $CO_2$ may be another possible pathway, although according to literature (EFSA, 2008) the mineralisation of the compounds is generally minimal and slow. As for possible volatilisation from aqueous systems/soil water, only limited losses can be expected, based on the same literature. This information will be discussed more in depth in the revised version.

Notes on response 6: The information has been included and discussed on p. 12, L. 4-6 as follows:

"Other dissipation pathways, such as mineralisation of the compounds to CO2 or volatilisation from aqueous systems/soil water were not ruled out. However, they were considered to be minimal according to literature (EFSA, 2008)."

**Comment 7:** *Can the expected degradation/mineralisation be quantified using literature values, and contrasted with the measurements?*

**Response 7:** We appreciate the observation, but an exact quantification would imply the application of modeling approaches. As already mentioned in the response to comment 8 of reviewer 1, modeling the data to do predictions is a distinct topic and not the purpose of the present manuscript. Therefore, it will be treated separately in a forthcoming study.

**Comment 8:** *With regard to the transformation/degradation of UR:*
*(1) How much of the degradation was possibly due to photolysis in the inlet container or at the system's surface?*
*(2) Could you estimate photolysis rates quantitatively?*
*(3) Microbial degradation of UR seems not to be enhanced after the first application, as illustrated by the similar recovery rate in the second part. If the system has not been exposed to UR before, and microbial decay was a major pathway, the first application of tracer probably would have fostered the microbial community capable of degrading the dye (Käss 1998), which would have enhanced microbial decay in the second run.*
*(4) Can alteration of fluorescent properties be a reason for apparent loss of UR (pH values are given in Table 2, but apparently only one measurement)?*

**Response 8-(1):** We thank the reviewer for raising this important point. Possible photolysis of the compounds in the inlet container was discarded, because this part of the system was covered to avoid exposure to light. As for the system's surface, we have assumed that photolysis of UR most likely took place. We will try to estimate a possible loss in the revised version.

**Response 8-(2):** One could estimate how much UR was lost during the first saturation phase according to the concentrations measured at the vegetated part of the uppermost layer. Light decay can be estimated assuming first order loss and half-lives from literature. This information will be included in the revised version.

Notes on response 8-(2): The estimated values are detailed and discussed on p. 11, L. 27-31 as follows:

"Assuming first-order decay, we obtained degradation coefficients of 0.05 and 0.17 days-1, and half-life times of 13 and 4 days for the first and second run, respectively. These values were comparatively lower than the half-life times reported in the literature, that are in the range of 11 hours (Leibundgut et al., 2009). However, natural light conditions could not be achieved in the laboratory and this could have limited UR photodegradation."

**Response 8-(3):** Regarding the possible microbial degradation of UR, we speculated that the missing percentage of the final mass balance was mostly due to abiotic degradation (i.e., photodegradation). Nevertheless, we have also hypothesized that possible microbiological degradation of UR took place, but to a lesser extent. The fact that it was not enhanced after the first application could be due to the probable existence of other preferred substrates for microbial degradation. These preferentially utilized compounds would have limited the degradation of alternative substrates such as UR. This is an important point that will be enlarged in the discussion of the revised version,

since we also expected a more intense biodegradation of UR during the second execution of our experiment.

Notes on response 8-(3): The discussion about microbial degradation has been addressed on p. 11, L. 15-23 as follows:

"UR, on the other hand, displayed comparatively lower recoveries, and based on the small amounts found in the sediments, sorption processes were not relevant for its dissipation. Thus, photodegradation and, to a lesser extent, (bio-)chemical transformation were most likely the major dissipation pathways for UR. Indeed, the contribution of (bio-)chemical transformation to UR dissipation has already been reported in other long-term studies (Maillard et al., 2016; Fernández-Pascual et al., 2018; Lange et al., 2018). Due to the likely adaptation of microorganisms to UR degradation after being exposed in the first run, we would have expected lower recovery rates in the second run (Käss 1998). However, the final recovery values of UR were similar in both runs (31.71 and 29.82% for the first and second run, respectively). Hence, we hypothesized that other substrates for microbial degradation were present in the system and were preferentially utilized limiting the degradation of alternative substrates such as UR."

**Response 8-(4):** The possible alteration of the fluorescent properties of UR is ruled out because the pH of the samples was always raised with buffer solution during the measurements. Unfortunately, the pH of the sand could only be measured at the end of the experiment when the sediment samples could be extracted. That is why we only provide one measurement.

**Comment 9:** *The different substances were mixed in one solution for application, but possible interactions of the applied substances are not discussed.*
   *(1) Can interactions of the different organic compounds be ruled out?*
   *(2) Testing a reference sample of the injection solution repeatedly over time could give further hints on interactions or degradation rates of the mixture.*

**Response 9-(1):** It is true that the possible interaction between the applied substances is not discussed in the manuscript and cannot be completely ruled out. The solution was prepared immediately prior to the injection. Tracer concentrations were measured inside the solution during the same day and a couple of days after and no significant changes were observed. On the other hand, the pesticides were mixed in one solution in the laboratory and no interaction of the substances was detected. We will discuss this point in the revised version.

Notes on response 9-(1): This point has been indicated on p. 4, L. 2-4 as follows:

"The solution of tracers and pesticides was prepared immediately before the injection. To control possible interactions between substances, the concentrations in the solution were measured on the day of the injection and a couple of days after."

**Response 9-(2):** Testing the injection solution over a long time was not considered. On the one hand because it was not possible to prevent the solution from being degraded by unknown microorganisms, and on the other hand, because the results of a possible

degradation/interaction in a bottle could not be transferred to a system with different conditions and greater complexity.

**2. Flow paths and preferential flow**

**Comment 10:** *The flow paths during the tracer application are not completely clear to me.*
- *(1) From Fig. 3 it appears as if the tracer solution was applied to the surface near the inlet container by letting the inlet container overflow.*
- *(2) The arrows indicate vertical movement downwards near the inlet, and vertical movement upwards elsewhere. Is the ponding on the surface from surface flow from the inlet, or from upward flow through the soil? In case of the former: Has air been entrapped in the system?*

**Response 10-(1):** We apologize for the confusion. In fact, the arrows on the surface of Fig. 3-1) only indicate the direction of the flow but do not represent the actual movement of the water during the injection and therefore can be misleading. These arrows will be eliminated in the revised version.

Notes on response 10-(1): The arrows in Fig. 3-1) have been corrected (see below).

[Figure]

**Figure 3:** Front view of the model constructed wetland system showing the execution of the injections (red arrows indicate the direction of the water flow): (1) surface injection of tracers and pesticides, (2) injection of clean water (tap water) from the bottom, (3) flushing of the system with clean water (tap water) from the bottom.

**Response 10-(2):** That is correct, the injection solution was applied to the surface near the inlet by letting the inlet container overflow. Due to the low flow rate the solution moved first downward near the inlet and then upward as the system was filling up. The ponding on the surface is coming from the upward flow. Therefore, possible entrapment of air in the system can be ruled out. We will clarify the flow paths by a more detailed description in the revised version.

Notes on response 10-(2): The description of the surface injection has been detailed on p. 4, L. 12-16 as follows:

"The surface injection (i) was performed after having drained the system. The solution was constantly pumped into the inlet reservoir. Then, it overflowed it and enter the sediments bed. Due to the low flow rate the solution moved first downward near the inlet and then upward as the system was filling up. The inflow was maintained until the system became saturated and the upward flow formed a surface ponding of approximately two centimeters height. In this way, possible entrapment of air in the system was avoided."

**Comment 11:** *A great part of the transport in the experiment has been attributed to preferential flow in the upper and lowermost layer. Given the coarse texture of the soil, hydraulic conductivity will be high and lead to fast regular flow rates already.*
   *(1) Is the observed breakthrough still considered preferential when compared with expected flow rates using conductivity and hydraulic gradient?*
   *(2) If preferential flow is an issue, how would that influence the spatial distribution of substances in the sediment, and in turn the recovery of substances from sampling sediment?*

**Response 11-(1):** We thank the referee for this important comment. Yes, we consider that the obtained breakthrough curves are due to non-uniform movement of water through the soil as a consequence of the system design and the presence of plants. This statement is based on the obvious differences in concentrations observed and the faster arrival of solutes to the lower and uppermost layers compared to the middle layers.
Mean flow velocity can be estimated if we assume uniform flow, but a comparison with expected velocities for each curve cannot be made without modeling the data. And, as already indicated in previous comments, this was not the purpose of the present manuscript.
**Response 11-(2):** Indeed, the observed distribution of substances in the sediment and their recovery is in agreement with the formation of preferential flow in the upper and lowermost layers.

Notes on response 11-(2): After carefully reviewing the data we have realized that certainly with the type of sediment we have used in our experiment it is very unlikely that preferential flow towards the bottom will occur. That is why we have eliminated such statement from the manuscript. However, we still believe that transport of solutes may have been favored towards the vegetated surface as there are strong indications that support it.

**3. Correlation analysis**

**Comment 12:** *Much of the interpretation is based on a correlation analysis.*

*(1) Please describe in bit more detail what was correlated – I expect you used measured concentration time series?*

*(2) Sorption is significant for some of the substances. How would retardation affect the results of the correlation analysis?*

**Response 12-(1):** We apologize for the lack of detail in this section and the revised version will include a better description. That is correct, we used measured concentration time series for the correlations.

**Response 12-(2):** We have hypothesized that the shape of the breakthrough curves will be affected by the retardation of the solutes resulting in non-significant or non-existent correlations particularly with Br. We will clarify this dependence in the revised version.

Notes on response 12-(2): This point has already been commented in the notes on response 23 of referee 1.

**Specific comments**

**Comment 13:** *P6, LL 33-35: This part is unclear.*

*(1) How does the design of the inlet cause preferential flow towards the bottom?*

*(2) And what is meant with "plants channel flow to the surface" – flow from lower layer to the soil surface? Or do you mean enhanced infiltration from the surface?*

**Response 13-(1):** We appreciate this comment. Obviously, we did not make this point clear enough and will clarify this in the revised version. The inflow system (by overflowing the inlet reservoir) and the low flow rate are believed to be the origin of preferential flow. Such design caused the injection solution to slide down the inlet glass wall channeling the water towards the bottom.

Notes on response 13-(1): See above notes on response 11-(2).

**Response 13-(2):** As for the plants, we hypothesized that they likely facilitated the transport of solutes along the root channels from the bottom to the surface layer. Besides this, the plants may have also introduced heterogeneities in the medium that have contributed to the formation of preferential flows. However, in agreement with comment 16 (see below), there could be other explanations for this phenomenon that are not necessarily related to the presence of plants. This will be addressed in the revised version.

Notes on response 13-(2): We have addressed this statement on p. 7, L. 11-18 as follows:

"Yet, according to Fig. 5, the uppermost layer displayed a delayed breakthrough peak with relative concentrations of Br- about three times higher than the maximum detected in the bottom (see also Table 4). In addition, the maximum values reached in the vegetated zone of the uppermost layer were twice as high as those of the non-vegetated, although these differences were not that pronounced in the second run. Hence, it was speculated that the plants, and more specifically the roots contributed to the formation of channels through which the transport of solutes was favored towards the vegetated surface. In this context, transport by preferential flow may have occurred along the

macropores formed by the root system. However, the results were inconclusive and other mechanisms, not necessarily related to plants (e.g. fingering), may have been involved too."

**Comment 14:** *P6, LL39-40: Consider rewriting sentence. Br- had almost complete recovery and was found in plants and roots, so you may delete "possibly", and refer to Fig. 8 and not only the lack of measured Br- in pore water.*

**Response 14:** Thank you for this suggestion. We agree with your assessment. As such, we will rewrite the sentence as proposed.

**Comment 15:** *P7, L4: "Early Breakthrough" – compare with expected flow velocity (see comment above)*

**Response 15:** We appreciate this remark. Yet, as stated in comment 11-(1), a comparison between the estimated mean flow velocity and the expected flow velocities for each curve is not possible without applying modeling approaches.

Notes on response 15: This point has already been commented in the notes on response 24 of referee 1. Here, we have clarified that the term "early breakthrough" has been removed from the text because it can be misleading.

**Comment 16:** *P7 L12: "absence of BTC in middle layer" and "early BTC in uppermost layer" "confirmed the influence of plants" is too strong as a statement. Other explanations are possible for these observations – preferential flow without the influence of plants (fingering), bias in the observations, etc.*

**Response 16:** We agree, and the statement will be corrected so that other possible explanations to the observations will be discussed.

Notes on response 16: See above notes on response 13-(2).

**Comment 17:** *P7 L 14 "evidenced" – too strong as well. It might be a hint but could also be that the degradation is just a function of time, and transport over that time ended in the vegetated part, opposite from the inlet.*

**Response 17:** We thank the reviewer for pointing this out. This sentence will be corrected too.

Notes on response 17: The sentence has been corrected on p. 8, L. 28-32 as follows:

"Metazachlor TPs were only found in the uppermost layer and their maximum relative concentrations were measured in the vegetated part after the promotion of aerated conditions. It should be noted, however, that the process of transformation may have been a function of time, and transport over that time ended in the vegetated part of the uppermost layer. Hence, the uppermost layer (possibly the vegetated part) and the end of the drying phase may have constituted hot spots and hot moments for transformation processes, respectively."

**Comment 18:** *P7 L23-27/Table 4: How significant are the differences in recovery of Br- given in Table 4, which is the basis for your argumentation here? The differences do not appear large enough to justify the conclusion.*

**Response 18:** We are grateful for this comment. The percentage of total Br recovered from the different depths was used to support the statement about the possible affectation of the system's performance due to changes in the density of the roots and/or spatial distribution. However, we agree that the differences in recovery between the first and second run are not big enough to justify such conclusion. Therefore, we will refrain from this conclusion in the revised version.

**Comment 19:** *P8, L 36: Please explain "low leaching potential" as a property of a substance – does that mean high sorption?*

**Response 19:** We appreciate this comment. "low leaching potential" means that the substance is less likely to move through the soil, but not only because of sorption, as this index is based on the chemical's adsorption (Koc) and persistence (DT50) in the soil. We will clarify this in the revised version.

Notes on response 19: This information has been clarified on p. 10, L. 11-15 as follows:

"Hence, we have hypothesized that the cause of the lower recoveries of boscalid and penconazole could have been their low leaching potential (USEPA, 2003; European Food Safety Authority (EFSA), 2008; Marín-Benito et al., 2015), which is based on their chemical's adsorption (Koc) and persistence in the soil (DT50). Yet, other causes, such as a higher incidence of plant uptake could not be ruled out."

**Comment 20:** *P9, L 38: "could be identified" – an unambiguous identification was unfortunately not possible in the experiment, but valuable hints / indications were collected*

**Response 20:** We thank the reviewer for this suggestion. The sentence in the revised version will be changed as proposed.

Notes on response 20: The corresponding change has been made on p. 13, L. 5-6.

**Comment 21:** *P10, L 5: "biochemical transformation had a major contribution" – only <10 % of the parent substance were found as TP, so it is not possible to say which was a major contribution*

**Response 21:** While it is true that only <10 % of metazachlor was found as TPs; the recoveries of this pesticide were the lowest among the solutes. This result, together with the physicochemical properties of metazachlor could be a hint that transformation/mineralisation might have played an important role in its dissipation. Nevertheless, it is true that we do not have enough information to justify such statement. Therefore, we will be more careful with this statement in the revised version.

Notes on response 21: This statement has been rewritten on p. 13, L. 18-20 as follows:

"The detection of metazachlor TPs, namely met-ESA and met-OA demonstrated that biochemical transformation played an important role in metazachlor dissipation,"

**Comment 22:** *Fig 4: How do you explain the obvious differences in Br- breakthrough between the first and second run?*

**Response 22:** We thank the referee for pointing this out. This difference has been attributed to possible changes in the density of the roots and/or spatial distribution over the experiment. This statement is based on the greatest development of the roots observed in the system at the end of the study (see Fig. 5 below). We will include this discussion in our revised version.

Notes on response 22: This point has already been commented in the notes on response 29 of referee 1.

[Figure]

**Figure. 5:** Front view of the root system in the vegetated part of the model constructed wetland before the first run A) and at the end of the second run B).

**Comment 23:** *Fig 7: Recovery of TP in % - how can the total amount be known?*

**Response 23**: We apologize for the lack of clarity in this case. The recovery of TPs has been calculated according to the total amount of parent compound injected. This will be defined in the revised paper.

Notes on response 23: This information has been clarified in Fig. 10 as follows:

"The mass balance for the TPs was calculated according to the total amount of parent compound injected."

**Comment 24:** *Fig. 8: Please comment on the large error bar for SRB:*
*(1) Which indicates recovery to be between 48 and 105 %.*
*(2) Would more sediment samples have reduced this uncertainty?*
*(3) How does this uncertainty influence your interpretation?*

**Response 24-(1):** We appreciate this comment. The recovery measured in the sediment of the vegetated part has a large error bar due to the heterogeneous distribution of the tracer. That is, almost 99% of the tracer measured in the vegetated part is located in the uppermost layer. This heterogeneous distribution indicates that the tracer was transported preferentially to this layer, as discussed in the manuscript.

**Response 24-(2):** We thank the referee for raising this important question. In fact, we collected a great number of sediment samples to reduce the uncertainty: A total of 16 sediment cores (four per longitudinal and four per lateral transect) that were divided into four fractions, each representing a different sampling depth (0-8 cm, 9-20 cm, 21-32 cm, 33-42 cm). This gives a total number of 64 sediment samples. We think that this number is adequate for the system.

**Response 24-(3):** In our case, we believe that it does not constitute a major factor of uncertainty. As indicated in the previous response, we have a measurement for each longitudinal/lateral transect and sampling depth, covering practically the whole sediment. This gives us a detailed picture of the distribution of the tracers in the system.

**Technical comments**

**Comment 25:** *P2, L32: 100 mg L -> 100 mg L$^{-1}$*

**Response 25:** The indicated change will be made in the revised version.

Notes on response 25: The corresponding change has been made on p. 3, L. 9.

**Comment 26:** *P2, L35: Please give the dimensions of the constructed wetland system also without inlet/outlet.*

**Response 26:** The dimensions will be provided in the revised version.

Notes on response 26: This information has been provided in Fig. 1 (see below).

[Figure]

**Figure 1:** Schematic representation of the model constructed wetland system (not to scale). Fi1 and Fi2 indicate the flowmeters at the inlet; Fo1 and Fo2, the flowmeters at the outlet; Ps(n), piezometer in the sand; Pk(n), piezometer in the gravel; 5TE-(n), soil moisture, temperature and electrical conductivity sensor; r(n), platinum redox electrode; Re, reference electrode (Ag:AgCl); Gf(n), glassfilter. For the piezometers, n indicates the position with respect to the inlet; n=1, close to the inlet; n=2, in the middle of the sediment bed and n=3, close to the outlet. For the sensors installed in the multi-level pipes, n indicates the zone and the depths where they are located; n = 1, 2, 3 and 4, non-vegetated zone at a depth of 39, 27, 15 and 3 cm, respectively; n = 5, 6, 7 and 8, vegetated zone at a depth of 39, 27, 15 and 3 cm, respectively.

(A) front view photograph of the model constructed wetland system; (B) detail of the multi-level pipes: (a) multi-level pipe at the vegetated zone,  (b) 5TE sensor, (c) redox electrode and (d) glass filter.

**Comment 27:** *P5, L11: resulted curves -> resulting curves*

**Response 27:** The indicated change will be made in the revised version.

Notes on response 27: The corresponding change has been made on p. 5, L. 33.

**Comment 28:** *P5, L13, and elsewhere throughout the text: Br -> Br$^-$*

**Response 28:** The indicated change will be made in the revised version.

Notes on response 28: The corresponding change has been made throughout the manuscript.

**Comment 29:** *P7, L26: "was most likely" -> "were most likely"*

**Response 29:** The indicated change will be made in the revised version.

Notes on response 29: The corresponding change has been made on p. 7, L. 41.

**Comment 30:** *P8, L28: "were classified": classified for what (recovery rate, I presume?)*

**Response 30:** Yes, that is right, the classification is for the recovery rate. We apologize for the confusion. The sentence will be improved.

Notes on response 30: The sentence has been improved on p. 10, L. 1-2 as follows:

"According to the total amount of tracers and pesticides recovered at the outlet after the flushings (Table 6), the solutes were classified as follows (from highest to lowest recovery rate): Br- >> SRB >> UR >> Boscalid >> Penconazole >> Metazachlor."

**Comment 31:** *Fig 4: Consider duplicating the figure and display vegetated and non-vegetated parts separately, which would make distinguishing these parts a lot easier*

**Response 31:** Thanks for the suggestion. We agree with your assessment, and as already indicated in the response to the comment 35 from reviewer 1, we will duplicate the figure to show separately the vegetated and non-vegetated parts.

**References**

**Comment 32:** Käss, W. (1998): Tracing technique in geohydrology, 581 pp., Balkema, Rotterdam, The Netherlands.

**Response 32:** The indicated change will be made in the revised version.

**Literature**

[revised manuscript text omitted]

---

## Author Response (AR2)

**Response to the review of the manuscript: Hydrological tracers for assessing transport and dissipation processes of pesticides in a model constructed wetland system**

Elena Fernández-Pascual1, Marcus Bork1,2, Birte Hensen3, Jens Lange1

1Hydrology, Faculty of Environment and Natural Resources, University of Freiburg, Freiburg, Germany

2 Soil Ecology, Faculty of Environment and Natural Resources, University of Freiburg, Freiburg, Germany
 3 Institute of Sustainable and Environmental Chemistry, Leuphana University Lüneburg, Lüneburg, Germany

*Correspondence to:* Elena Fernández-Pascual (elena.fernandez@hydrology.uni-freiburg.de)

**Answers to editor**
* * *
Dear Prof. Zehe,

We are very pleased to have the opportunity to review and resubmit our manuscript. We truly appreciate all the insightful observations. We have carefully considered all the comments and recommendations, and the manuscript has been accordingly revised.

We provide below a point-by-point response with the editor and reviewers' comments and our responses in blue. We have also indicated how the observations and corrections have been made (in the new revised version) detailing the corresponding page and line number where applicable.

We hope that these revisions improve the paper such that you and the reviewers now deem it worthy of publication in the Hydrology and Earth System Sciences journal.

Once again, thank you for taking the time and energy to help us improve the paper.

We look forward to the outcome of your assessment.

On behalf of all the authors,

Elena Fernández-Pascual

**Editor**

**-----**

**Comment:**

I am happy to let you know that both reviewers recommend publication of your manuscript in HESS. Reviewer 2 suggested a long list of technical improvements, which need to be addressed to improve the presentation quality of your work to the level that we expect for HESS.

**Response:** We want to thank both reviewers for the positive feedback. The corresponding technical improvements have been addressed in the manuscript. We hope our revision meets with your approval.

Answers to referee #1

**\_\_\_\_\_**

**Comment:**

After careful reading of the comments of the reviewers, the answers of authors and the new manuscript, I am fully convinced that the authors did a great job to improve their manuscript. To my opinion, all important issues regarding the generic character of the study, the lack of replicate experiments, difficulties to close mass balances and relationship between reaction and residence time have been thoroughly taken into account. The authors clearly did a great job and the manuscript is in a very good shape. My only comment is that now the manuscript could be again polished a bit for English language. Please, avoid long sentences whenever possible.

**Response:** We wish to acknowledge the constructive and thoughtful comments of the reviewer. Following your recommendation, we have asked some colleagues to correct the whole manuscript.

Answers to referee #2

**Comment 1:**

The authors have made some efforts to rewrite and clarify the questions raised in the first reviews. Some issues, however, still need to be resolved. The authors should again check their line of argument in critical appraisal of their data and the peculiarities of their setup, and the manuscript could be streamlined and condensed. The resulting paper would surely deserve publication as a valuable source for further studies.

**Response 1:** We thank the reviewer for the positive feedback, thoughtful comments and constructive suggestions that have helped us improve the manuscript. We next detail our

responses to the reviewer's comments and the explanation of how the observations and corrections have been addressed in the revised manuscript.

**Comment 2:**

Perhaps the authors might want to revisit the objectives stated in the introduction, also in light of the comments made by the other referee. Maybe objectives i) and ii) are enough and can also be formulated as three bullet points (investigation of transport, investigation of dissipation, comparison of tracers and pesticides). The design of the experiment does include vegetation and alternating saturation but was perhaps not adequate to investigate the influence of vegetation and alternating saturation in great details, due to the lack of replica and controls. Of course, these things can still be discussed, and answering the remaining questions is still challenging enough.

**Response 2:** We appreciate the reviewer's suggestion and we agree that we may have formulated too many questions in our study. As a result, the interpretation of the findings has turned out to be quite challenging. However, if we remove the study of the influence of vegetation and alternating saturation conditions from the objectives, the whole approach will make no sense because the operation and design of the experiment was intended to answer those questions, and so, it will be very difficult to justify the choice of our particular setup. On the other hand, our aim was not to investigate the influence of the aforementioned factors in great detail, but in a very general way. Besides, the inclusion of vegetation and alternating saturation conditions also sought to promote the processes under study.

**Comment 3:**

The authors refrain from including modeling in the paper, as they point out in their responses to the reviews. Without a quantitative analysis, however, large parts of the conclusions on transport, retardation, sorption and transformation remain rather speculative. A comprehensive numerical simulation would indeed be out of scope of the paper, but perhaps a first assessment of the results using solute transport theory (transfer functions) would have provided a more adequate means to analyze the transport behavior. Instead, the analysis of the data relies heavily on correlations between the breakthrough curves, but correlation ignores any shifts due to retardation. The analysis of lag correlation (which neglects any dispersion) was newly introduced in the revised version, but it is limited to the bromide BTC. Comparing the different tracers and assessing possible retardation and transformation is difficult, if not impossible, with this approach. Looking at lag correlations between the different BTCs (and reporting the significant findings) could possibly improve this weakness a bit, if further analyses should really be out of scope.

**Response 3:** We are aware of the limitations of the correlation analysis, but it has served as a first comparison between the behavior of the different compounds. As we have commented in the previous review, the evaluation of the results using solute transport theory has not been made due to the complexity and particular approach of our experiment. In any case, we have followed your suggestion and we have looked at more lag-correlations. The results are discussed in the next comments.

**Comment 4:**

Any quantitative assessment requires getting a good description of the flow behavior. The presented perceptual model of the flow field is that tracers enter at the inlet, percolate downwards to the bottom gravel layer, from where they are transported to the upper layers by upwelling. The observations show that the middle layers are not participating in this transport, which is attributed to preferential flow. That water should have passed by all observation points in this layer is unlikely, but if all other explanations can be ruled out, this surprising finding would have to be considered in the interpretation. In my opinion, it is much more likely that lateral transport also occurred at or near the surface, perhaps on top of the saturated, "unconnected" middle layers. This would much better explain the higher concentrations near the surface compared to the bottom. Preferential flow could still be present in the second run, when solutes also appear in the middle layers. I encourage the authors to revisit this point.

**Response 4:** We thank the reviewer for the valuable opinion. We fully agree and also consider that some type of lateral transport at or near the surface may have occurred in our experiment. This hypothesis was contemplated from the beginning, but due to the type of injection we considered it unlikely. In any case, we cannot ignore this possibility because, as the reviewer has pointed out, it would much better explain part of the behavior of the solutes in our system. Other possible explanations, such as the influence of the roots, have also been taken into account. All these points are now included in the discussion of the results in the revised version.

**Comment 5:**

The discussion and interpretation of the results should be checked critically, streamlined and condensed. The structure with 5 sub-sections is fine, but perhaps it can be rearranged to separate more clearly the interpretation of the experimental results from the inferences on wetlands and the referencing to the literature. For example, last paragraph of 3.2 and last half of 3.3 could go to the end; 3.3 and 3.4 could be combined, etc.

**Response 5:** Thanks for the suggestion. We agree with your assessment. As such, we have rearranged some of the paragraphs to separate the interpretation of the results from the implications of our findings for wetland systems. Only the subsections "Recoveries of solutes at the outlet" and "Final mass balance" have not been merged because we consider that they discuss two different points that must be treated separately. Also, if we combine them the resulting subsection will be too long.

**Specific Comments**

**Comment 6:**

*p* 2, *l* 29: Strictly speaking, it is not a control, because the difference is not only that on part is being planted, but also is located at the inlet.

**Response 6:** We thank the reviewer for pointing this out. Indeed, the definition of control is not very clear in this particular experiment since it is a single unit. Therefore, we will omit the term "control" from the text.

**Comment 7:**

p 4, ll 12 to 15: The tracer injection is not fully clear. It is stated that solutes were contained in 40 L of water. This solution was pumped into the system until saturation. According to Fig. 2, saturation was reached in 1-2 days. On page 3, l 41, it is stated that saturation with target substances was one week. In Tab. 2 (B) the pumping rate is given with 21.6 L h-1. Was the tracer solution injected in two hours or two days? Was other water used to further saturate the system after the 40 L were injected?

**Response 7:** We apologize for the confusion. The water level in Fig. 2 is only schematic and does not correspond to real water level measurements (as indicated in the caption). This means that saturation was not reached in 2 days but in about 2 hours. What we meant on p. 3, L 41 was that the system remained saturated for about a week, but no other water was used to further saturate the system after the 40 L were injected. We have clarified this issue on p. 4 L. 1-2 and on p. 4 L. 14-16 as follows:

 $\rightarrow$  During the first saturation phase, only one injection was made, whereas in the second saturation phase, the system was kept saturated by constant injections of tap water.

 $\rightarrow$  The inflow was held constant for about two hours, until the system became saturated and the upward flow formed a surface ponding of approximately two centimeters.

**Comment 8:**

*p* 6, *l* 11: Due to its conservative and non-sorbing character, Br– can hardly serve for investigating retardation, which is commonly caused by reversible adsorption/desorption.

**Response 8:** Thanks for raising this important point. The cross-correlations between  $Br^-$  time series were intended to provide information about the arrival of the solutes to the different layers. Indeed, given the conservative character of  $Br^-$  the delay could not be related to sorption processes but rather to other causes, such as low pore connectivity, as has been discussed on p. 7 L. 25-27:

 $\rightarrow$  In contrast, the delayed peak of the uppermost layer was associated with possible low pore connectivity in the vicinity of the sampling ports.

**Comment 9:**

**p* 6, *l* 18: Correlating BTC between the vegetated and non-vegetated zones could also be interesting, especially for comparison of the zones**

**Response 9:** Thank you for this suggestion. We have performed additional correlations between the time series of the solutes of the vegetated and non-vegetated zone distinguishing between the first and second run. These results are presented now in a new table (Table. 5, see below) and they are discussed in subsection 3.1, on p. 7 L. 29-32:

 $\rightarrow$  If we compare the performance of the vegetated and the non-vegetated zone by means of correlations (Table 5), we observe stronger correlations in the lower layers (at 27 and 39 cm depth) than in the upper layers (at 15 and 3 cm depth), especially in the uppermost during the first run. These results suggested a greater influence of the plants and/or other causes (e.g. transport along the surface) on solute transport in upper layers.

| Depth |                                              | First run |            | Second ru | n          |
|-------|----------------------------------------------|-----------|------------|-----------|------------|
| (cm)  |                                              | rho       | p-value    | rho       | p-value    |
| 3     | Br nv : Br v           | 0.43      | 0.09       | 0.53      | 0.05*      |
|       | UR nv : UR v           | 0.30      | 0.26       | 0.33      | 0.25       |
|       | SRB nv : SRB v         | 0.79      | 0.26       | 0.32      | 0.26       |
|       | Bos nv : Bos v         | 0.79      | 0.06       | 0.76      | 0.03*      |
|       | $Pen_{nv}$ : $Pen_{v}$                       | 0.46      | 0.35       | -         | -          |
|       | Met nv : Met v         | 0.59      | 0.22       | 0.58      | 0.13       |
|       | Met-ESA nv : Met-ESA v | 0.40      | 0.43       | -         | -          |
|       | Met-OA nv : Met-OA v   | -         | -          | -         | -          |
| 15    | Br nv : Br v           | 0.49      | 0.06       | 0.63      | 0.02*      |
|       | UR nv : UR v           | 0.53      | 0.04*      | 0.05      | 0.86       |
|       | SRB nv : SRB v         | 0.51      | 0.04*      | -0.01     | 0.98       |
|       | Bos nv : Bos v         | -         | -          | -         | -          |
|       | Pen nv : Pen v         | -         | -          | -         | -          |
|       | Met nv : Met v         | -         | -          | -         | -          |
|       | Met-ESA nv : Met-ESA v | -         | -          | -         | -          |
|       | Met-OA nv : Met-OA v   |           |            | -         | -          |
| 27    | Br nv : Br v           | 0.58      | 0.02*      | 0.97      | < 0.001*** |
|       | $UR_{nv}$ : $UR_v$                           | 0.85      | < 0.001*** | 0.78      | < 0.001*** |
|       | SRB nv : SRB v         | 0.67      | 0.004 **   | 0.64      | 0.01**     |
|       | Bos nv : Bos v         | -         | -          | -         | -          |
|       | Pen nv : Pen v         | -         | -          | -         | -          |
|       | Met nv : Met v         | -         | -          | 0.76      | 0.03*      |
|       | Met-ESA nv : Met-ESA v | -         | -          | -         | -          |
|       | Met-OA nv : Met-OA v   | -         | -          | -         | -          |
| 39    | Br nv : Br v           | 0.53      | 0.03*      | 0.27      | 0.35       |
|       | $UR_{nv}$ : $UR_v$                           | 0.84      | < 0.001*** | 0.95      | < 0.001*** |
|       | SRB nv : SRB v         | -0.06     | 0.83       | 0.73      | 0.003**    |
|       | $Bos_{nv}$ : $Bos_v$                         | -         | -          | -         | -          |
|       | $Pen_{nv}$ : $Pen_v$                         | -         | -          | -         | -          |
|       | Met nv : Met v         | 0.40      | 0.44       | 0.76      | 0.03*      |
|       | Met-ESA nv : Met-ESA v | -         | -          | -         | -          |
|       | Met-OA nv : Met-OA v   | -         | -          | -         | -          |

**Table 5.** Spearman rank correlation of the breakthrough curves between the vegetated and non-vegetrated zones for the different solutes and the first and second run.

Signif. Codes: 0.001 '\*\*\*'; 0.01 '\*\*'; 0.05 '\*'

nv = non-vegetated; v = vegetated

**Comment 10:**

**p* 6, *l* 24: As stated in the next paragraph, an overall mass balance for pesticides was not possible.**

**Response 10:** You raise a very valid point here. We have corrected the sentence on p. 6 L. 20-22 as follows:

 $\rightarrow$  The fate of the tracers and their main dissipation pathways were examined with a final overall mass balance that accounted for five different compartments (pore water, outlet water, sediments, stems + leaves and roots).

**Comment 11:**

p 7, ll 8 to 18: Why can these observations not be due to flow along the surface? Higher values in the upper layers than in the bottom are hardly possible if solute flow was first to the bottom and then up - or you missed important parts of the breakthrough with your sampling frequency

**Response 11:** We thank the reviewer for pointing this out. As already stated in comment 4, lateral transport at or near the surface is now included as a possible explanation of the results of our study. Yet, we consider that the presence of plants may have also favored the transport of solutes towards the vegetated surface. This point is discussed in section 3.1, on p. 7 L. 10-15 as follows:

 $\rightarrow$  Hence, it can be speculated that lateral transport at or near the surface may have occurred during the injections causing augmented transport of solutes towards the vegetated surface. However, other possible explanations could not be ruled out. These include the likely influence of the plants by means of water uptake and the possible contribution of the roots to the formation of channels through which preferential flow took place. Other mechanisms, not necessarily related to plants (e.g. fingering), may have been involved in the transport of solutes to the vegetated area, too.

**Comment 12:**

p 7, ll 19 to 24: What are the time units of the lag correlation? If it is days, why are only lags up to 7 days tested/displayed? From the BTC it appears that lag could be in the range of weeks – months. Why were only Br-BTC analysed?

**Response 12:** The time units of the lags are days but referring to the observation dates, and this may vary between several days to weeks. This information has been specified in Fig. 6.

We computed the cross-correlation of two univariate series. The number of lags displayed in the plot are calculated by default as  $10*\log_{10}(N/m)$  where N is the number of observations and m the number of series. In our case, given that we had from 8 to 16 number of observations and 2 series, the number of lags displayed were from 8 to 5.

We have analyzed Br- because due to its more conservative character, it serves for investigating transport. Unfortunately, the lag analysis performed to the other BTCs did not yield conclusive information.

**Comment 13:**

p7, ll 24/25: That is difficult to understand. Do you argue that transport was from the bottom to the top, but delayed due to low connectivity? Why was there a transport then at all? How fast would a transport have been that was not delayed? How do you rule out transport along the surface? Isn't that much more likely in view of the higher concentrations at the top compared to the bottom?

**Response 13:** Once again, thank you for pointing this out. As we have explained in the response to comments 4 and 11, transport along the surface is now included in our explanation of transport in the experiment.

What we argue here is that the system was filled from the bottom to the top and that due to the conditions previous to the injection in the sediments, the solutes distributed heterogeneously and reached the sampling ports at different times. It should be noted that the values obtained at the different sampling ports were only representative for the nearest areas. Considering this, we hypothesized that the delayed measurement of solutes in the uppermost layer was due to low pore connectivity in the vicinity of the sampling ports. Probably the solutes arrived to this layer earlier, but we couldn't measure them at the time of the arrival because of the presence of water-filled pores in the surroundings of the sampling ports. We apologize because we have not made this point clear enough. Therefore, we have better explained this question in the text on p. 7 L. 25-28 as follows:

 $\rightarrow$  In contrast, the delayed peak of the uppermost layer was associated with possible low pore connectivity in the vicinity of the sampling ports. Here the solutes probably arrived earlier, but we could not measure them at the time of the arrival because of the presence of water-filled pores in the surroundings of the sampling ports.

**Comment 14:**

p 7, l 29: The contrast between "strong" correlation and "not any" correlation sounds greater than it actually is (0.77 vs. 0.55). More important, that does not tell much about the transport characteristics other than that they are different. How about lag correlation here?

**Response 14:** We thank the reviewer for this correction. While it is true that some of the values indicated no correlation (e.g. 0.14 or 0.26) other values pointed out that the correlations were not significant (e.g. 0.55 with p-value 0.15). The sentence has been corrected on p. 7 L. 33-36 as follows:

 $\rightarrow$  With the exception of the uppermost layer, the non-vegetated zone showed strong correlation between the two runs regardless of the layer, whereas in the vegetated zone some layers did not show any correlation (layers at 3 and 27 cm depth) or displayed correlations that were statistically non-significant (layers at 15 and 39 cm depth).

As suggested, we have performed the lag correlation analysis to the Br- breakthrough curves between the first and second run. The results have been included in the supplementary material (Fig. S3, see below). This new information has been added to the text on p. 7 L. 38-40 as follows:

 $\rightarrow$  Lag correlations between the first and second run were also analyzed (Fig. S3 of the supplementary material). A significant value (at time t=-3) was found in the vegetated zone at 15 cm depth.

**Figure S3.**: Lag analysis performed to the Br- breakthrough curves between the first and second run for the vegetated and non-vegetated zones and the sampling depths: 1) 3cm; 2) 15cm; 3) 27cm and 4) 39cm.

**Comment 15:**

p 7, ll 30 to 33: Why not mentioning that Br also has experienced plant uptake? For example, S Xu et al., Environ. Sci. Technol. 2004, 38 (21), 5642-5648

**Response 15:** Thanks for raising this important point. We completely agree, and the possible influence of Br- uptake by the plants has been detailed on p. 8 L. 1-2:

 $\rightarrow$  However, other causes, such as the influence of flushing between runs and Br- uptake by the plants (Xu et al., 2004) could not be ruled out.

**Comment 16:**

p 8, ll 7/8: These parts are more or less the beginning and the end of the transport regime through the tank, right?

**Response 16:** That is right. The information has been incorporated on p. 8 L. 17-19 as follows:

 $\rightarrow$  According to the correlations performed to the solute time series (Fig. 8), two spots exhibited the strongest relationships: the non-vegetated part of the lowermost layer and the vegetated part of the uppermost layer. These spots coincided with the beginning and end of the transport regime through the system.

**Comment 17:**

*p* 8 *l* 17: "confirmed creation of preferential flow paths" - Or maybe it was only high conductivity and low sorption in the gravel layer, and overland/near-surface flow?

**Response 17:** Thank you for pointing this out. We have rewritten the sentence on p. 8 L. 27-30 as follows:

 $\rightarrow$  This was not the case for SRB, which only displayed strong positive correlation with Br- in the lower- and uppermost layers. The former was explained by possible high conductivity and low sorption in the gravel. The results from the uppermost layer were associated with the likely promotion of transport towards the vegetated surface, given the strong sorptive character of SRB.

**Comment 18:**

p 9, l 40: "evidencing their great mobility and persistence" – But the total recovery of metazachlor and TPs was low, < 20 %. This is a contradiction.

**Response 18:** Here, we wanted to emphasize that the TPs were still coming out from the system at the end of the experiment when we flashed it. The fact that we recovered TPs at the outlet, even if it was a small amount, was an indication that they were not further degraded and/or retained in the system. We consider that this is an important finding. However, since the sentence can lead to confusion, we have rewritten it on p. 9 L. 41-42 and p.10 L. 1-2 as follows:

 $\rightarrow$  Even though small amounts of TPs were obtained, it was an indication that they were not further degraded or retained in the system, which was in agreement with the findings of other studies (Mamy et al., 2005; European Food Safety Authority (EFSA), 2008). Higher amounts of met-ESA were recovered compared to met-OA.

**Comment 19:**

**p 10, ll 9 to 14: These sentences are directly contradicting each other: Either DT50 values let expect higher recoveries, or DT50 values lead to lower recoveries**

**Response 19:** We thank the reviewer for raising this point. We realize that there is a contradiction in our argument regarding the leaching potential and therefore it will be removed from the text. From the DT50 values, we would have expected higher recoveries, because the duration of the experiment was below those values. Since the recoveries were low, we think that both, the high sorption potential of these solutes and the possible incidence of plant uptake were the cause. We have corrected the sentence on p. 10 L. 15-16 as follows:

 $\rightarrow$  Hence, we have hypothesized that the cause of the low recoveries of boscalid and penconazole could have been their high sorption potential and possible plant uptake.

**Comment 20:**

p 10, l 24 to p 11, l 2: This part not only related to recovery, perhaps put it towards the end (discussion of meaning of results for pesticide removal in wetlands in general).

**Response 20:** We appreciate this suggestion. The indicated part has been moved to the end of the discussion to the new subsection called: "Implications for pesticide mitigation in wetland systems"

**Comment 21:**

p 11, 135 to p 12, 16: No plant uptake, no transformation, no recovery, no mineralization, no volatilization - where did it go? Is it really plausible that the bulk of the substances was sorbed on sand and gravel particles? What would that imply for future applications of these substances – would sorbing capacities be depleted?

**Response 21:** We are not arguing here that plant uptake or transformation did not take place at all. They probably occurred but to a lesser extent compared to sorption. The retention of these substances has definitely played a fundamental role (at least for boscalid and penconazole) and it should be considered when assessing the mitigation capacities of these pesticides in wetlands. In these systems, the depletion of the sorption capacities will depend on both, the concentration of the adsorbing substances and the number of sorption places. These factors should be taken into account for future applications of strongly sorbing pesticides such as boscalid and penconazole.

This conclusion has been included in the manuscript on p. 11 L. 27-31:

 $\rightarrow$  With this in mind, it can be concluded that retention has played a fundamental role in our study (at least for boscalid and penconazole). Therefore, special attention should be given to retention processes when assessing the mitigation capacities of strongly sorbing pesticides such as boscalid and penconazole in wetlands. In these systems the depletion of the sorption capacities will depend on both, the concentration of the adsorbing substances and the number of sorption places.

**Comment 22:**

p 12, ll 7/8: Is this difference between vegetated and non-vegetated for Br and UR really significant? Can it only be attributed to the vegetation? How does the distance to the inlet influence this finding?

**Response 22:** We thank the reviewer for underscoring these questions. We have checked the differences in the values between the vegetated and the non-vegetated area and in fact they are not significant. In view of the results, we don't think that the distance to the inlet have had any influence on this finding. We have corrected the sentence on p. 11 L. 32-34 and now it can be read as follows:

 $\rightarrow$  Lower amounts of UR and Br- were recovered from the pore water of the vegetated compared to the non-vegetated zone (Fig. 10-A), however, the differences were not significant.

**Comment 23:**

p 12, ll 10/11: How do you rule other possibilities? Couldn't preferential flow paths also be due to the flushing in between runs that contributed to a change in pore space? Which effect would a stronger hydraulic gradient invoked by root water uptake in the vegetated zone have?

**Response 23:** Indeed, it is possible that transport in the second run may have been influenced by the flushings performed at the end of the first run. This important information has been added to the discussion about the different performances of the system between runs on p. 8 L. 1-2:

 $\rightarrow$  However, other causes, such as the influence of flushing between runs and Br- uptake by the plants (Xu et al., 2004) could not be ruled out.

Root water uptake could have also facilitated transport of solutes towards the vegetated surface, although probably to a lesser extent in view of the results of the second run. Nevertheless, this possibility has also been included in the explanation of transport on p. 7 L. 12-14:

 $\rightarrow$  However, other possible explanations could not be ruled out. These include the likely influence of the plants by means of water uptake and the possible contribution of the roots to the formation of channels through which preferential flow took place.

**Comment 24:**

**p12, ll 26/27: From Fig 8 and 9 (A1 and B1) I would rather argue that SRB and the two pesticides are not behaving that similar.**

**Response 24:** We thank the reviewer for pointing this out. Obviously, we have not made our point clear enough. We have not stated that SRB and the other two pesticides (boscalid and penconazole) have behaved similarly in the pore water, as their breakthrough curves were different. Instead, we have argued that compared to the other solutes (Br, UR and metazachlor), they have shown a greater tendency to sorption. And this hypothesis has been based on their similar rapid decrease in their concentrations after the injections and gradual increase in accumulated mass recovery during the flushing phase. Regarding the recovery curves, what we wanted to express was that they still exhibited increases even after the systems was repeatedly flashed. To avoid misunderstandings, we have removed the sentence "Analogous recovery curves for SRB, boscalid and penconazole were observed" from the text.

**Comment 25:**

p12, ll 33/34: I do not see where you have shown unequivocally that UR was transformed bio-chemically - you assume that, because there is a large part missing in the mass balance (cf. p 10, l19/20).

**Response 25:** Yes, that is correct. Unfortunately, biological degradation of UR has not been demonstrated unequivocally in this study. Rather, the results suggested that this process may have also occurred. In a recent study (Lange et al., 2018) biochemical

transformation of UR has been shown in arable soils, if this process happens also in saturated wetlands still needs to be shown.

**Comment 26:**

p12, l 38: How realistic / relevant is that long time period? Elsewhere, you have reported mean residence times of 6 days for typical wetland systems.

**Response 26:** Obviously in natural wetland systems the period that we have simulated is probably not realistic. But we have shown that the promotion of long periods of stagnation may be beneficial to eliminate pesticides. This parameter could be taken into account when constructing engineered systems designed to remove contaminants from the water. This conclusion has been included in the revised manuscript on p. 12 L. 18-20 as follows:

 $\rightarrow$  According to the findings, promoting solute contact with the medium through long periods of stagnation should be taken into account when constructing engineered systems designed to remove contaminants from the water.

**Comment 27:**

**p13, l 12 to 14: Absence in middle layers because of the flow regime (local effect of this particular setup)? (related to comments above )**

**Response 27:** We wanted to highlight that boscalid and penconazole were not detected in the middle layers, while all the other compounds were detected (particularly in the nonvegetated zone). Therefore, we do not think that their absence was due to a local effect because, if that had been the case, it would have affected all solutes.

Nevertheless, it is true that the particular setup has had an effect on the results, especially the type of injection which has conditioned the way the solutes entered the system. This has been discussed in the manuscript, for instance on p. 8 L. 34-35:

 $\rightarrow$  Spatial and temporal variability of transport and dissipation processes were associated with the conditions prior to injection, the way the solutes entered the system, the presence of plants and the promotion of aeration during the drying phase.

**Comment 28:**

**p 19, Table 2(A): Mean initial organic carbon content - how is zero error achieved?**

**Response 28:** This is because we rounded to one decimal place. The actual value is:  $0.2 \pm 0.02$ . We have added a second decimal place to avoid misunderstandings. The same has been done for the mean initial dithionite-extractable Fe (Fed).

**Comment 29:**

*p* 19, Table 2(A): Conductivity: Is Br breakthrough detectable in the conductivity measurements with the 5 TE sensors?

**Response 29:** Yes, some of the sensors captured the trend of Br- but not a complete breakthrough. Strong correlations between conductivity and Br- have been observed,

particularly in the non-vegetated area at 15 cm depth for the first run and the vegetated area at 15 and 27 cm depth for the second run. The information has been added to the text on p. 7 L. 19-22 as follows:

 $\rightarrow$  The delayed peak of Br- observed at 15 cm depth in the non-vegetated area for the first run was also detected by the conductivity probe located at the same depth. The complete breakthrough curve could not be capture by the sensor, but a strong correlation (Spearman's rho=0.83 and p-value<0.001) between Br and the conductivity values was found (see Table S2 of the supplementary material).

**Table S2.** Spearman rank correlation between the breakthrough curves of Br- and the conductivity values of the probes located at the same depth for the first and second experimental run and the different zones.

| Zonas         | First run                                                                                                        |                                                                                                                                   | Second run                                                                                                                                                                                                                                                                                                                                       |                                                                                                                                                                                                                                                                                                                                                                                                                                                                                                                                                                                                                                                                                                                                                                                                                                                                                                                                                                                                                                                                                                                                                                                                                                                                                                                                                                                                                                                                                                                                                                                                                                                                                                                                                                                                                                                                                                                                                                                                                                                                                                                                     |
|---------------|------------------------------------------------------------------------------------------------------------------|-----------------------------------------------------------------------------------------------------------------------------------|--------------------------------------------------------------------------------------------------------------------------------------------------------------------------------------------------------------------------------------------------------------------------------------------------------------------------------------------------|-------------------------------------------------------------------------------------------------------------------------------------------------------------------------------------------------------------------------------------------------------------------------------------------------------------------------------------------------------------------------------------------------------------------------------------------------------------------------------------------------------------------------------------------------------------------------------------------------------------------------------------------------------------------------------------------------------------------------------------------------------------------------------------------------------------------------------------------------------------------------------------------------------------------------------------------------------------------------------------------------------------------------------------------------------------------------------------------------------------------------------------------------------------------------------------------------------------------------------------------------------------------------------------------------------------------------------------------------------------------------------------------------------------------------------------------------------------------------------------------------------------------------------------------------------------------------------------------------------------------------------------------------------------------------------------------------------------------------------------------------------------------------------------------------------------------------------------------------------------------------------------------------------------------------------------------------------------------------------------------------------------------------------------------------------------------------------------------------------------------------------------|
| Zones         | rho                                                                                                              | p-value                                                                                                                           | rho                                                                                                                                                                                                                                                                                                                                              | p-value                                                                                                                                                                                                                                                                                                                                                                                                                                                                                                                                                                                                                                                                                                                                                                                                                                                                                                                                                                                                                                                                                                                                                                                                                                                                                                                                                                                                                                                                                                                                                                                                                                                                                                                                                                                                                                                                                                                                                                                                                                                                                                                             |
| Non-vegetated | -0.34                                                                                                            | 0.19                                                                                                                              | 0.60                                                                                                                                                                                                                                                                                                                                             | 0.02 *                                                                                                                                                                                                                                                                                                                                                                                                                                                                                                                                                                                                                                                                                                                                                                                                                                                                                                                                                                                                                                                                                                                                                                                                                                                                                                                                                                                                                                                                                                                                                                                                                                                                                                                                                                                                                                                                                                                                                                                                                                                                                                                              |
| Vegetated     | 0.07                                                                                                             | 0.80                                                                                                                              | -0.2                                                                                                                                                                                                                                                                                                                                             | 0.49                                                                                                                                                                                                                                                                                                                                                                                                                                                                                                                                                                                                                                                                                                                                                                                                                                                                                                                                                                                                                                                                                                                                                                                                                                                                                                                                                                                                                                                                                                                                                                                                                                                                                                                                                                                                                                                                                                                                                                                                                                                                                                                                |
| Non-vegetated | 0.83                                                                                                             | <0.001 ***                                                                                                                        | -                                                                                                                                                                                                                                                                                                                                                | -                                                                                                                                                                                                                                                                                                                                                                                                                                                                                                                                                                                                                                                                                                                                                                                                                                                                                                                                                                                                                                                                                                                                                                                                                                                                                                                                                                                                                                                                                                                                                                                                                                                                                                                                                                                                                                                                                                                                                                                                                                                                                                                                   |
| Vegetated     | -0.39                                                                                                            | 0.13                                                                                                                              | 0.84                                                                                                                                                                                                                                                                                                                                             | <0.001 ***                                                                                                                                                                                                                                                                                                                                                                                                                                                                                                                                                                                                                                                                                                                                                                                                                                                                                                                                                                                                                                                                                                                                                                                                                                                                                                                                                                                                                                                                                                                                                                                                                                                                                                                                                                                                                                                                                                                                                                                                                                                                                                                          |
| Non-vegetated | -                                                                                                                | -                                                                                                                                 | -                                                                                                                                                                                                                                                                                                                                                | -                                                                                                                                                                                                                                                                                                                                                                                                                                                                                                                                                                                                                                                                                                                                                                                                                                                                                                                                                                                                                                                                                                                                                                                                                                                                                                                                                                                                                                                                                                                                                                                                                                                                                                                                                                                                                                                                                                                                                                                                                                                                                                                                   |
| Vegetated     | -0.18                                                                                                            | 0.51                                                                                                                              | 0.82                                                                                                                                                                                                                                                                                                                                             | <0.001 ***                                                                                                                                                                                                                                                                                                                                                                                                                                                                                                                                                                                                                                                                                                                                                                                                                                                                                                                                                                                                                                                                                                                                                                                                                                                                                                                                                                                                                                                                                                                                                                                                                                                                                                                                                                                                                                                                                                                                                                                                                                                                                                                          |
| Non-vegetated | -0.07                                                                                                            | 0.81                                                                                                                              | 0.05                                                                                                                                                                                                                                                                                                                                             | 0.86                                                                                                                                                                                                                                                                                                                                                                                                                                                                                                                                                                                                                                                                                                                                                                                                                                                                                                                                                                                                                                                                                                                                                                                                                                                                                                                                                                                                                                                                                                                                                                                                                                                                                                                                                                                                                                                                                                                                                                                                                                                                                                                                |
| Vegetated     | 0.32                                                                                                             | 0.23                                                                                                                              | 0.54                                                                                                                                                                                                                                                                                                                                             | 0.04 *                                                                                                                                                                                                                                                                                                                                                                                                                                                                                                                                                                                                                                                                                                                                                                                                                                                                                                                                                                                                                                                                                                                                                                                                                                                                                                                                                                                                                                                                                                                                                                                                                                                                                                                                                                                                                                                                                                                                                                                                                                                                                                                              |
|               | Zones
Non-vegetated
Vegetated
Non-vegetated
Non-vegetated
Vegetated
Non-vegetated
Vegetated | ZonesFirst run
rhoNon-vegetated-0.34Vegetated0.07Non-vegetated-0.39Non-vegetated-Vegetated-0.18Non-vegetated-0.07Vegetated0.32 | $\begin{tabular}{ c c c } \hline First run \\ \hline rho & p-value \\ \hline rho & 0.19 \\ \hline rho & 0.19 \\ \hline rho & 0.80 \\ \hline rho & 0.83 &